

5 # Reconciling the signal and noise of atmospheric warming on decadal timescales

Roger N Jones* and James H Ricketts

Victoria Institute of Strategic Economic Studies, Victoria University, Melbourne, Victoria 8001, Australia

40 *Correspondence to*: Roger N. Jones (roger.jones@vu.edu.au)





**Abstract**

Interactions between externally-forced and internally-generated climate variations on decadal timescales is a major determinant of changing climate risk. Severe testing is applied to observed global and regional surface and satellite temperatures and modelled surface temperatures to determine whether these interactions are independent, as in the traditional signal-to-noise model, or whether they interact, resulting in steplike warming. The multi-step bivariate test is used to detect step changes in temperature data. The resulting data are then subject to six tests designed to show strong differences between the two statistical hypotheses, $h_{step}$ and $h_{trend}$: (1) Since the mid-20th century, most of the observed warming has taken place in four events: in 1979/80 and 1997/98 at the global scale, 1988/89 in the northern hemisphere and 1968/70 in the southern hemisphere. Temperature is more steplike than trend-like on a regional basis. Satellite temperature is more steplike than surface temperature. Warming from internal trends is less than 40% of the total for four of five global records tested (1880–2013/14). (2) Correlations between step-change frequency in models and observations (1880–2005), are 0.32 (CMIP3) and 0.34 (CMIP5). For the period 1950–2005, grouping selected events (1963/64, 1968–70, 1976/77, 1979/80, 1987/88 and 1996–98), correlation increases to 0.78. (3) Steps and shifts (steps minus internal trends) from a 107-member climate model ensemble 2006–2095 explain total warming and equilibrium climate sensitivity better than internal trends. (4) In three regions tested, the change between stationary and non-stationary temperatures is steplike and attributable to external forcing. (5) Steplike changes are also present in tide gauge observations, rainfall, ocean heat content, forest fire danger index and related variables. (6) Across a selection of tests, a simple stepladder model better represents the internal structures of warming than a simple trend – strong evidence that the climate system is exhibiting complex system behaviour on decadal timescales. This model indicates that *in situ* warming of the atmosphere does not occur; instead, a store-and-release mechanism from the ocean to the atmosphere is proposed. It is physically plausible and theoretically sound. The presence of steplike – rather than gradual – warming is important information for characterising and managing future climate risk.

**Key words:** global warming, climate change, decadal variability, step change, severe testing, statistical induction, signal to noise, complex trends

# 1 Introduction

The climate research community conducts two separate narratives describing how the atmosphere warms under the influence of increasing greenhouse gases: one focused on methods and the other on theory (Jones, 2015b, a). The method-focused narrative describes how model and observational data should be analysed and used in detection and attribution, projection and forecasting. The theory-focused narrative describes how the climate system changes over multiple timescales. These narratives are usually articulated separately and are often at cross purposes with each other. Although they both recognise the climate as having linear and nonlinear components, one treats them as being separate, whereas the other explores the possibility that the two interact.

The two main hypotheses that describe the interactions between climate change and variability over decadal timescales are (Corti et al., 1999;Hasselmann, 2002):





1. Externally-forced climate change and internally-generated natural variability change independently of each other.
2. They interact, where patterns of the response project principally onto modes of climate variability (Corti et al., 1999) or form a two-way relationship (Branstator and Selten, 2009).

These two hypotheses have very different outcomes for the characterisation of climate-related risk (Jones et al., 2013). The

methods-focused narrative centres on the use of a signal-to-noise model using ordinary least-squares trend analysis (e.g., North et al., 1995;Hegerl and Zwiers, 2011;Santer et al., 2011). This dominates climate practice and has led to the construction of the gradualist adaptation narrative, which describes adaptation as an incremental series of adjustments over time (Jones et al., 2013). However, if the internally and externally forced components of climate interact, producing steps, shifts or jumps, adaptation planning based on gradual change will lead to risk being underdetermined. *H1* is the default assumption, but

according to the latest Intergovernmental Panel on Climate Change report, the choice between the two remains unresolved (Kirtman et al., 2013).

This paper explores the potential for *H2* to be true by applying severe testing principles (Mayo and Spanos, 2010) to detect and analyse step changes in temperature data. The bivariate test of Maronna and Yohai (1978) is used to test regional and global surface air temperature, global satellite temperature of the lower troposphere and global mean temperature from the

CMIP3 and CMIP5 climate model archives.

## 2    Analytic Framework
### 2.1    Reasoning by statistical inference
The gradualist narrative is a product of induction – if warming is accepted as following a trend, induction leads to the assumption that warming is gradual. Induction also suggests that gradual increases in radiative forcing will lead to gradual

warming. As an analytic process, induction is reasoning by inference, which can be by analogy, statistical inference (often using probabilities) or induction to a particular (if all A have been B, then the next A will be B, Curd and Cover, 1998). The application of linear trend analysis to atmospheric warming is invariably justified as inference to the best explanation. However, the latter stance has been criticised, because using parsimony and likelihood tests to select a structural model is inferior to tests that apply experimental reasoning and are statistically suited for the problem in question (Mayo, 1996;Mayo

and Spanos, 2010;Spanos, 2010). This is particularly relevant for complex systems exhibiting intrinsic nonlinear behaviour. These authors argue that conditions for severe testing should be probative, rather than relying on a particular probability threshold. If test *T* has no likelihood of finding flaws in *H*, then it is not a good test. Mayo (1996) calls this the fallacy of acceptance: no evidence against the null is interpreted as evidence for it, and evidence against the null is interpreted as evidence for an alternative.

Consistent with this, Mayo and Spanos (2011) advise care in distinguishing between the error statistic and the probability of confirmation – likelihood tests passing criteria such as *H* if $pH_0<0.05$ run the risk of being interpreted as addressing scientific hypotheses with the same level of confidence. This is a weakness of ordinary trend analysis; although a trend may register a low probability of meeting the null hypothesis, it does not infer that the data forms a smooth trend, only that the residuals are





normally distributed when the trend is removed. Trend statistics assume no history, whereas in physical systems, process is often one way, which should be considered in any testing environment.

Inference to the best explanation can be rescued if it passes the test: "no available competing hypothesis explains a fact as well as *H* does" (Musgrave, 2010). Mayo (2005) provides criteria for severe testing that is even stricter: Data *x* in test *T* provide

good evidence for inferring *H* to the extent that hypothesis *H* has passed a severe test with *x*. The severe testing of not-*H* is part of this severity, meaning that all other possibilities must be exhausted before *H* can be accepted.

Cox and Mayo (2010) distinguish between probabilistic and behavioural reasoning – the first explores which levels of probability are appropriate for making a specific inference – when do data provide good evidence for *H*? Behavioural reasoning will prompt a particular decision based on a given probabilistic position being met. For example, if a test achieves $pH_0 < 0.05$

a hypothesis may be considered 'proven'. Probabilistic reasoning is suitable for complex situations where there is no single cause or effect and there is no easy way to distinguish different types of error when analysing these. For example, a test might provide a particular *p* value but might not represent all of the phenomena of interest. This is relevant to the analysis of temperature records.

### 2.2    Development of severe testing

A hypothesis *H* passes a severe test *T* with data *x* if (Mayo and Spanos, 2010):

1.   *x* agrees with *H* and,

2.   with very high probability, test *T* would have produced a result that accords less well with *H* than does *x*, if *H* were false or incorrect.

Here, we are using the substantive null of model adequacy approach described by Mayo and Cox (2010) – specific results are

used to provide theoretical evidence that can relate statistical tests with a scientific null $H_0$. Rather than *H* and $H_0$, rival hypotheses *H1* and *H2* are represented by statistically distinct models. This structure implies two stages of data and hypothesis development. The first stage considers the original temperature data from observations or models and its link with hypotheses *H1* and *H2*. and the second stage is that produced by various statistical tests. The two hypothesis stages are scientific hypotheses (*H*; Mayo calls these theories, but here they are applied as hypotheses) and statistical hypotheses (*h*). The latter are subject to

Type I and Type II errors. Sometimes these error types are used to justify scientific hypotheses but one is linked to statistical error and the other is probative, so they should be kept distinct.

Scientific hypotheses need to be examined closely in order to identify the probative conditions required to be built into one or more statistical tests for severe testing. These are then subject to error-based statistical testing to ensure that both *H* and -*H* and *h* and -*h* have sound and testable bases. This very quickly becomes complicated:

• Type I statistical errors – incorrectly rejecting the null (i.e., wrongly accepting a statistical hypothesis).

• Type II statistical errors – failing to reject the null (i.e., failing to accept a statistical hypothesis).

• *H1* hypothesis – The climate signal is gradual and variability is noise, therefore the signal is gradual and follows a (probably monotonic) trend. Non-linear changes in climate data are due to internal climate variability and are random. The hypothesis is confirmed once a trend is judged as significant.



- *H1* null hypothesis – a trend has not emerged from the variability.
- *H2* hypothesis – externally forced and internally generated climate processes interact with each other producing a nonlinear climate signal on decadal timescales.
- *H2* null hypothesis – the climate change signal is gradual and any nonlinearity is of internal origin (*H1*).

This structure is asymmetric and points to the fact that *H1* has never really been severely tested – the usual *H1* null hypothesis is that a trend has not emerged beyond a level of significance. Although alternative structures have been explored including steps and segmented trends (Seidel and Lanzante, 2004), the results have been inconclusive, and the exact nature of change on decadal timescales is still an open question (Trenberth, 2015).

Slingo (2013) describes it thus:

*Statistical model comparisons ... do not allow the identification of a 'true' model, only the computation of the relative likelihood of the models considered. Additionally, these statistical models do not take into account the physics of the climate or any of the known external influences that affect global temperatures. In an analysis ... if all the tested models are poorly specified then even the most likely of the tested set of models will be a poor representation of the behaviour of the real climate. In such cases the relative likelihood of the models considered*

*is of little scientific value.*

Another complication is that nonlinear change has been used to challenge global warming theory on the basis that if observed change is not gradual, therefore *H1* is false, and climate change is either disproven or overstated (Jones and Ricketts, 2016). Evidence of nonlinear change, such as step change, is therefore associated with challenges to global warming theory (e.g., see Skeptical Science, 2015).

This asymmetry in null hypotheses means that if *H2* was to be accepted or rejected on the basis of severe testing, severe testing is needed for both *H1* and *H2*.

## 2.3    Development of key statistical tests

Understanding how the global atmosphere warms over decadal timescales is an inverse, ill-posed mathematical problem, similar to that for borehole or satellite temperatures (e.g., Şerban and Jacobsen, 2001; Aires et al., 2001). Atmospheric warming
is the product of external forcing and internal variability subject to some undetermined delay between forcing and response. Global mean temperature also integrates regional signals, so is spatially complex – if responses to forcing are regional occurring in different places at different times, then the global signal will be a product of these changes. Global average warming is a complex timeseries that exhibits both linear and nonlinear characteristics.

External forcing is boundary-limited which implies long-term linear behaviour (Lorenz, 1975), and internal variability is
nonlinear – both points are agreed to under *H1* and *H2*. The major question concerns how these combine. The long-term signal is invariably measured using trend analysis, however, the cut-off between what constitutes short- and long-term timescales is subjective, being a product of the test method and the question being posed (Jones and Ricketts, 2016).



The accumulation of heat energy through radiative forcing is additive, whereas its entrainment into the hydrothermal system is nonlinear (Ozawa et al., 2003;Lucarini and Ragone, 2011;Tsonis and Swanson, 2012), favouring *H2*. The presence and nature of change points in the warming record and in climate model output will indicate whether this could be the case. Change points in climate data are increasingly being located by a variety of different techniques (Rodionov, 2005;Reeves et al.,

2007;Tsonis et al., 2007;Overland et al., 2008;Hope et al., 2010;Killick et al., 2010;Beaulieu et al., 2012;Fischer et al., 2012;Jones, 2012;Jandhyala et al., 2013;Ruggieri, 2013;Cahill et al., 2015).

The hypotheses *H1* and *H2*, imply at least two structural models. For *H1*, close adherence to a warming trend implies that the atmosphere warms gradually. If so, this must occur via either of the following two processes or a combination:

1.  Radiatively-forced warming occurs in the atmosphere and the heat generated is retained there (*in situ* warming).
Statistically, this would manifest as gradually increasing temperatures, especially over land. It would also imply a trend in lower troposphere satellite temperatures as the airmass warms gradually.

2.  Most of the heat generated by added greenhouse gas forcing goes into the ocean and is gradually released into the atmosphere. Again, this would imply gradual warming, especially over the oceans, with the land following suit, but with greater variation if decadal changes in shallow and deep-ocean mixing of heat are taken into account.

Discussions in the literature are not clear as to whether the oceanic component is due to varied take-up or release of heat from the ocean.

3.  If both 1 and 2 are operating, then the warming rate in the atmospheric component would be monotonic (proportional to forcing) and the contribution from the ocean governed by interannual and decadal variability. This would be best represented by a segmented trend if decadal-scale deep and shallow ocean mixing of heat is a key

factor.

For *H2*, we build on the hypothesis that patterns of forced response may either project onto modes of climate variability or that they interact. These manifest as step changes in temperature data. Currently, if such changes are detected, they are routinely attributed to climate variability. The possibility that they may be a response to external forcing has not been severely tested (Jones and Ricketts, 2016).

If warming is mediated by the hydrothermal ocean-atmosphere system, it could be entrained by the nonlinear processes involved (Lucarini and Ragone, 2011). Lucarini and Ragone (2011) describe this as the generation of entropy, as moist static energy is transformed into mechanical energy like a heat engine. This could flip between different states, modulated by Lorenzian 'strange attractors' as described by Palmer (1993). Increased forcing may increase entropy, making the climate system more energetic and less efficient, speeding up these processes over time (Lucarini and Ragone, 2011), much like a

bubbling pot.

Given that this entrainment would be into nonlinear modes of climate variability, we characterise it as rapid changes or steps often associated with regime change (see also Tsonis and Swanson, 2012). This is essentially a store and release process, where heat stored in the ocean is released when the ocean-atmosphere relationship becomes unstable, precipitating regime change.





These ideas are discussed in more detail in Jones and Ricketts (2016). To date, we have identified widespread step changes in a range of climate variables, most notably temperature (Jones et al., 2013), and carried out attribution studies for one region, south-east Australia (Jones, 2012). The exploration of *H2* is to therefore detect step changes more broadly and to contrast this with trend-like behaviour.

Given that this entrainment would be into nonlinear modes of climate variability, we characterise it as rapid changes or steps often associated with regime change (see also Tsonis and Swanson, 2012). This is essentially a store and release process, where heat stored in the ocean is released when the ocean-atmosphere relationship becomes unstable, precipitating regime change. These ideas are discussed in more detail in Jones and Ricketts (2016). To date, we have identified widespread step changes in a range of climate variables, most notably temperature (Jones et al., 2013), and carried out attribution studies for one region,

south-east Australia (Jones, 2012). The exploration of *H2* is to therefore detect step changes more broadly and to contrast this with trend-like behaviour.

The challenge for severe testing is to create tests that can distinguish between *H1*, *-H1*, *H2* and *-H2* with sufficient confidence to select one over the other. The following six probative tests have been identified to test the relationship between linear and nonlinear behaviour and their responses to external forcing:

1. Stratified analysis of change points: the timing and distribution of change points and their relationship with known regime changes and with each other. Change points aligning with known nonlinear processes indicate a causal link.

2. Identification of similar patterns of steps between observations and physical models indicates a physically coherent origin, rather than random stochasticity.

3. Using internal trends and shifts (steps minus internal trends) to estimate the gradual and rapid warming components
in a record, and testing each of these against criteria such as total warming and equilibrium climate sensitivity (ECS) in observations and models separately.

4. Tests to diagnose stationarity – using a simple linear inverse model to measure the emergence of an anomalous signal from the background noise of variability, and whether it is gradual or steplike.

5. Testing of other variables including rainfall, sea surface temperatures, sea level rise, and air pressure, to see whether
they undergo similar changes.

6. Statistical models applied to test underlying step- and trend-like structures in the data.

### 2.4    Method

A multi-step and rule-based application of the Maronna-Yohai bivariate test (MYBT, Maronna and Yohai, 1978) has been developed to expand the original test beyond the detection of single steps (Ricketts and Jones, 2016, see Supplementary
Information for details). Previously, the bivariate test has been used to detect inhomogeneities in climate variables (Potter, 1981a;Bücher and Dessens, 1991a;Kirono and Jones, 2007;Sahin and Cigizoglu, 2010), decadal regime shifts in climate-related data and step changes in a wide range of climatic timeseries (Buishand, 1984;Gan, 1995;Vivès and Jones, 2005;Boucharel et al., 2011;Jones, 2012;Jones et al., 2013). The main purpose of automating the test is to improve its objectivity and robustness by using a predefined set of rules.





The test adapts the formulation of Bücher and Dessens (1991a) and tests a single serially-independent variate ($x_i$) against a reference variate ($y_i$) using a random timeseries following Vivès and Jones (2005). The important outputs of the test in a timeseries of length $N$ are, (1) The $T_i$ statistic which is defined for times $i < N$, (2) the $T_{i0}$ value which is the maximum $T_i$ value, (3) $i_0$, the time associated with $T_{i0}$, (4) shift at that time, and (5) $p$, the probability of zero shift. Note that $i_0$ is the last year prior

to the change. In this paper, we routinely give the year of change.

A single timeseries analysis consists of a *screening pass*, followed by a *convergent pass*. In both passes, we apply a *resampling test* to each segment being examined, where the test is repeated 100 times, resampling the random number reference series. The screening pass starts from the most significant shift in a timeseries, determined using the resampling test and, if $p < 0.01$, the series is divided into shorter timeseries either side of the step and these are tested until all steps have been detected. As

such, it is a recursive procedure whereby the first steps detected may be influenced by as-yet-unlocated steps. The convergent pass then serially refines these segments to provide a causal sequence. The convergent process is repeated until a stable set of step changes is produced.

The above analysis is run 100 times. Ricketts and Jones (2016) show that this procedure may produce several different but related solutions (solution = set of change dates); the most common solution is returned as the best estimate. Alternatives often

indicate the presence of localised events embedded in larger scale areally-averaged data. The majority solution is selected for further analysis. Most historical temperature records analysed contain one or two stable configurations for surface temperature and zero or one for satellite temperature. Climate model data may produce a larger number of stable solutions, especially the higher forcing scenarios.

Mean annual data for observations is considered serially independent – and in most cases applied in the paper, the MYBT is

reliable. Deseasonalised quarterly and monthly data can be used to locate a shift within a year, but is not serially independent, so is used here in combination with the t-test either side of the change date to assess significance. A resampling test that shuffles data either side of a shift will also indicate whether a change point is abrupt, or the timeseries is trend-like. Twenty-first century model data is not serially independent under high rates of forcing, an issue discussed in Sect 4.3. For error testing using statistical hypotheses, we routinely use behavioural reasoning at levels of $p < 0.01$ for the bivariate test (exceptions are noted),

and non-significant (NS, $p > 0.05$), $p < 0.05$ and $p < 0.01$ for trend analysis and the t-test.

Local attribution of step changes uses a technique detailed in Jones (2012). The basic methodology is suitable for continental mid-latitude areas where annual average maximum temperature (*Tmax*) is correlated with total rainfall (*P*), and minimum temperature (*Tmin*) is correlated with *Tmax* (Power et al., 1998;Nicholls et al., 2004;Karoly and Braganza, 2005). For Central England Temperature, a largely maritime climate, diurnal temperature is assessed against precipitation instead of *Tmax*. The

method uses the following steps:

1.   Homogenous regional average data is obtained for *Tmax*, *Tmin* and *P*.
2.   A period of stationary climate is calculated by testing when the relationship between *Tmin* and *Tmax* undergoes a statistically significant step change. The relationship between *Tmax* and *P* will change at the same, or later date.
3.   Linear regressions are calculated between each pair (*Tmax/P* and *Tmin/Tmax*) for the stationary period.





4. Externally forced warming is estimated for the non-stationary period using these regressions.

5. The results are tested for step changes.

Timeseries tested here are mean annual global air temperature anomalies from five groups (GISS, Hansen et al., 2010;NCDC, Peterson and Vose, 1997;HadCRU, Morice et al., 2012;BEST, Rohde et al., 2012;C&W, Cowtan and Way, 2014), hemispheric temperatures from three groups (HadCRU, NCDC and GISS) and zonal temperatures from two groups (NCDC and GISS) to see how prevalent step changes are, whether they coincide across different records and to investigate the relationship between step changes and trends. Tropospheric satellite temperatures from two groups (UAH, Christy et al., 2003;Christy et al., 2007; RSS, Mears and Wentz, 2009) are also tested. The specific records used are detailed in the Supplementary Information. Simulated mean global surface temperature from the CMIP3 and CMIP5 climate model archives is also tested. The analysis is carried out in two parts. The first part investigates simulated 20th century temperatures to determine how well the models reproduce the pattern of step changes in the observed data. The second part analyses how step changes evolve over the 21st century under the different Radiative Concentration Pathways (RCPs).

Comparing steps and trends creates two statistical hypotheses, $h_{step}$ and $h_{trend}$. Conventional trend analysis will extract a monotonic signal from the data. Measurement of change where nonlinear behaviour is present is not an exact process. The bivariate test measures total change between segments of a timeseries, ignoring any trend that may be present. These we refer to as steps. However, trends may also be present within timeseries registering steps, so internal trends are calculated between steps. The distance between the end of one trend and the start of the next is referred to as a shift. The process of calculating steps then trends, we call the step and trend model. Steps, internal trends and shifts all provide data for severe testing.

Shifts and internal trends are not strictly additive – summed over a number of steps they can add up to more or less than the change in temperature measured between the beginning and end of a series. These differences are largest in records containing reversals and negative trends.

The main phenomena analysed are:

- Steps – measurement of the whole change between two timeseries assuming stationarity as produced by the bivariate test. This assumes no trend either side of the step.

- Internal trends – measurement of the trends between steps.

- Shifts – measure of the internal step between the end of a preceding trend and the beginning of the next trend.

- Trend/step ratio – the ratio between total internal trends and total steps in a multi-step timeseries.

- Trend/shift ratio – the ratio between total internal trends and internal shifts (steps minus trends).

### 3 Results – observations
#### 3.1 Global and zonal temperatures
This section undertakes global, hemispheric and zonal analysis to determine temporal and spatial patterns of step changes in observed temperature, consistent with tests 1 and 2.



Step changes meeting the *p*<0.01 threshold in global and zonal temperatures show a great deal of structure over the 1880–2014 time period. All series were tested from their earliest recorded date (1850 and 1880) and results from 1880–2014 are shown. Downward steps occur in the late 19$^{th}$ and early 20$^{th}$ century, upward steps between 1912 and 1938 with one downward step in 1964. From 1968, upward steps dominate, with one exception in the high southern hemisphere (SH) latitudes in 2007 (Fig.

5    1).

**Figure 1: Dates of statistically significant step changes (*p*<0.01) 1880–2014, for a range of mean annual temperature records. Downward steps are blue and upward red. Records are sourced from Goddard Institute of Space Studies (GISS), the Hadley Centre and Climate Research Unit: HadCRU (land and ocean), HadSST (ocean), CRUtem (land), National Climatic Data Center: NCDC**

**(land, land and ocean), ERSST (ocean), Berkeley Earth Surface Temperature (BEST) and Cowtan and Way (C&W). See Supplementary Information for details.**

The 1997 step change is global, with some regional steps occurring in 1996 and 1998. A global step change occurs 1979/80; also registering in many regions, except the northern hemisphere mid and high latitudes. All other step changes occur across more limited regions, with some being confined solely to land or to ocean. The 1997 step is the largest at 0.31±0.01 °C. The

1979/80 step is the next largest at 0.22±0.03 °C. The greater variation in size of 1979/80 is affected by the timing and size of previous steps and trends. In the first half of the 20$^{th}$ century, three global records show positive steps in 1920/21 and in 1937, and two in 1930 (Fig. 1). The GISS record also shows a downward step in 1902, coinciding with the northern hemisphere (NH) ocean, tropics and southern hemisphere. The two groups are based on the early 20$^{th}$ century differences: GISS, BEST, C&W in one group and HadCRU and NCDC in the other. The anomaly averaged from all five records shows upward step

changes in 1930, 1979 and 1997, coinciding with the HadCRU and NCDC records.

Differences emerge between ocean and land records. The global HadSST (HadCRU) record shifts in 1937, 1979 and 1997, whereas the ERSST (NCDC) record shifts in 1890, 1930, 1977, 1987 and 1997. Global land records from both CRU and NCDC shift in 1920/21, 1980 and 1997. Northern hemisphere land and ocean step changes are consistent across three records: in 1924/25, 1987 and 1997. The NH ocean shows a downward step in 1902/03 and is less consistent between the two records

tested for subsequent upward steps. The SH is consistent across 1937, 1979 and 1997, with two records showing a downward step in 1890 and an upward step in 1969.

The tropics show a downward step in 1902/03, and upward steps in 1926, 1979 and 1997. Three NH mid-latitude records step upwards in 1920, 1921 or 1930, in 1987/88 and 1997/98. One zonal record also shows a downward step in 1964. The two NH high latitude records show a single downward step in 1902 and in 2005, both step upwards in 1921 and 1994 and a single step

upwards in 2005. The three SH mid-latitude records show a downward step in 1887 and one in 1902, and upward steps in 1933 or 1937, 1968 or 1970, 1977/1978 or 1984, and 1997 or 1998. SH high latitude data is not very reliable, being absent for NCDC 60°S–90°S. The GISS 64°S–90°S average anomaly steps downward in 1912 and an upward in 1955.



Fig. 2 shows the internal trends and their error significance for the five global mean temperature records. Steps and trends are consistent for the last two periods 1979/80 to 1996 and 1997 to 2013/14, but diverge in the middle of the record, due to differences in the timing and magnitude of steps and accompanying internal trends. Data quality may be an issue in the earlier parts of the record. For example, the version of GISS data used here shows five steps in 1902, 1920, 1937, 1980 and 1997,

whereas a previous version to 2013 stabilised on steps in 1930, 1979 and 1997, consistent with the average anomaly of all five records. This indicates that the timing and magnitude of steps in the early 20th century can be influenced by adjustments made to improve data quality. However, all global step change dates coincide with regional steps, showing that while the relative importance of dates associated with step changes may be different, the dates themselves are quite stable. This gives us added confidence we are not detecting false positives.

Internal trends are mainly non-significant in the early record, the exception being the GISS 1920–37 period. The 1979/80 to 1996 trend is significant at the $p<0.01$ level in two records (HadCRU and NCDC) and $p<0.05$ in the other three records. The NH step change in 1987 seen in all three records tested strongly influences this trend, which is examined further in the next section. The post-1997 period is non-significant in two records and trends at $p<0.05$ in three records.

There is no objective way to partition shifts and internal trends. Giving the first preference to internal trends in calculating

ratios provides the criteria for the severe testing of non-linear responses. As some of the internal trends show $p>0.05$, this is a conservative stance preferencing the methodological status quo. Expressed as a ratio between internal trends and steps, four global records range between 0.32 and 0.38 with the GISS record yielding a ratio of 0.62 due to the cool reversal in the early 20th century. For trends and shifts, the ratio ranges between 0.44 and 0.58 with the GISS record an outlier at 1.38.

Test 2 aims to determine whether at the regional level, trends or steps are more prominent than at the global scale. The global

trend/step ratio for the HadCRU record, for example, is 0.55 (0.30 °C/0.55 °C), for the NH is 0.31, the SH 0.28 and the tropics (30°N–30°S) is 0.33; close to the average of the two hemispheres. When divided into land and ocean, the HadCRU and NCDC records, show 0.90 and 1.15 for land, and 0.16 and 0.26 for ocean, respectively, showing the oceans to be more steplike and the land having roughly equal measure. SH ocean is very steplike (0.16) and SH land, less so (0.39). The mid-latitudes are also very steplike as is the tropical ocean. High ratios (>1) often involve a temporary cool reversal around the early 20th century.

**Figure 2: Mean global anomalies of surface temperature with internal trends. The annual anomalies (dotted lines) from five records (HadCRU, C&W, BEST, NCDC, GISS) are taken from a 1880–1899 baseline. Internal trends (dashed lines) are separated by step changes detected by the bivariate test at the $p<0.01$ error level. The size of each step (in red) and change in temperature of each internal trend (in black) is shown in the figure table along with its significance, where NS is $p>0.05$, * is $p>0.01<0.05$, ** is $p<0.01$.**

**Totals of trends, steps, shifts (change from one trend to the next) and ratios are also shown.**

This is also the case for single steps on a regional basis. In 1997/87 the global shift was 0.16±0.01 °C, a ratio of about 50% compared to the step change of 0.32 °C. For the northern hemisphere, this ratio varied between 57% and 68% for three land and three ocean data sets. For the northern hemisphere mid-latitudes, land and ocean from two data sets (NCDC 30°N–60°N, GISS 24°N–44°N), steps/shifts measure 0.43 °C/0.44 °C, close to a 1:1 ratio, indicating no trend.



The more steplike character of both the oceans and the mid-latitudes is consistent with those areas being the loci of change in terms of decadal regimes and nonlinear equator-to-pole transport. This is inconsistent with the hypothesis of gradual warming Varying shift dates and rates of change will contribute to the global record being more trend-like than individual regions.

### 3.2    Satellite era records

A comparison of surface and satellite temperatures stratifies records according to altitude and source of measurement. Satellite records of lower troposphere temperatures sourced from the RSS and UAH records beginning in December 1978, were analysed for step changes (1979–2014). Anomalies were investigated annual and seasonally. Annual mean global and zonal temperatures show 1995 and 1998 as the two main shift dates, with 1995 more prominent at the global scale (Table 1). For individual seasons, steps in 1995 are dominated by the NH JJA and SON periods, especially on land. This can be traced back

to warm El Niño conditions in 1994/5. For the quarterly timeseries (4 seasons x 36 years), the JJA and SON quarters of 1997 dominate the UAH global record, less so for the RSS record.

Quarterly anomalies for the RSS and UAH satellite and HadCRU and GISS surface mean global temperature were compared for similarities and differences. Quarterly timeseries are affected by autocorrelation due to the El Niño-Southern Oscillation (ENSO), for the bivariate test making results robust for timing but not significance. Student's t-test (two sided, unequal

variance), was used as a back-up.

**Table 1 about here**

RSS shifts in DJF 1987/88 by 0.11 °C ($p<0.05$ MYBT and $p<0.1$ t-test) and UAH shifts in DJF 1987/88 and 0.09 °C (NS MYBT and $p<0.05$ t-test). For surface temperature, HadCRU and GISS shift in JJA 1987 by 0.14 °C and 0.15 °C, respectively ($p<0.01$, both tests). On an annual basis, the bivariate test registers 1987/88 at the $p<0.05$ level. The lower significance in the

satellite records is due to the slightly lower shift size and higher variance. RSS shifts in JJA 1997 by 0.23 °C, UAH shifts in DJF 1997/98 by 0.26 °C, HadCRU in JJA 1997 by 0.26 °C and GISS in SON 1997 by 0.25 °C (all $p<0.01$, both tests). These four data sets show consistent shift dates in 1997 and similar shift dates in 1986/7, showing that the significant step change in the NH is present at the global scale. This suggests that the period of accelerated trend noted by many for 1976–1998 (e.g., Trenberth, 2015) is actually a period containing two step changes, one global (1979/80) and one regional (1987/88).

When all four records are plotted on a common baseline of 1979–1998, the surface and satellite temperatures display similar shifts but different internal trends (Fig. 3). Shown this way, the supposed differences between surface and satellite trends are largely removed. The satellite data contain 'significant' negative internal trends over 1979–1986 (RSS $p<0.01$, UAH $p<0.05$), otherwise are $p>0.05$. The surface data show significant positive internal trends over 1997–2014 (GISS $p<0.01$, HadCRU $p<0.05$), otherwise are $p>0.05$. The decline post 1981 and lower trends in the early 1990s in the satellite data are likely due to

volcanic eruptions, which amplify cooling at altitude (Free and Lanzante, 2009). The differences in internal trends post 1996 may be due to orbital decay that has not been fully allowed for in the satellite record, cooling from above affecting the satellite data and heating from below affecting the surface data, or a combination of these.

Unless substantially contaminated by artefacts, these changes do not reflect gradual warming in the atmosphere, but instead may reflect regime-like change controlled from the surface. The capacity for the oceans to emit sufficient heat during El Niño





events and absorb it during La Niña to cause large warming anomalies at this scale events suggests that available heat energy is not a limiting factor to abrupt changes.

**Figure 3: Quarterly mean satellite (RSS, UAH) and surface (HadCRU, GISS) temperature anomalies on a common baseline 1979–**
**2014. Annual anomalies (dotted lines) and internal trends (dashed lines) are separated by step changes.**

At this timescale, both surface and satellite temperature records are very steplike. The trend/shift ratios for the HadCRU and GISS records are 0.19 and 0.27 respectively and for the RSS and UAH records are -0.55 and -0.40, respectively, showing the effect of the negative internal trends. Shifts are consequently higher than steps in the satellite data. These are clearly due to the presence of the ENSO cycle within the data and, given the coincidence of step dates with some El Niño events, there is no
clear way to allow for these, so the data is analysed and presented as is.

### 3.3    Regional attribution

This section on regional attribution covers the issue of stationarity and the character of change over regional areas. Regional attribution of step changes in annual temperature has previously been carried for south-eastern Australia (SEA, Jones, 2012)
and is repeated here for Texas and central England. The methodology is suitable for continental mid-latitude areas where annual average minimum temperature ($Tmin$) is correlated with maximum temperature ($Tmin/Tmax$), and $Tmax$ is correlated with total annual rainfall ($Tmax/P$) (Nicholls et al., 2004;Power et al., 1998;Karoly and Braganza, 2005). For maritime areas such as central England, diurnal temperature range ($DTR$) is used ($DTR/P$) instead of $Tmax/P$. The method uses the bivariate method to test the dependent variable against the reference variable. A shift in the dependent variable denotes a regime change.
SEA climate was stationary until 1967 when a step change increased $Tmin$ by 0.6 °C with respect to $Tmax$ (Jones, 2012). Six independent climate model simulations for the same region become non-stationary by the same means between 1964 and 2003, showing steps of 0.4 to 0.7 °C (Jones, 2012). Texas becomes non-stationary in 1990 with an increase in $Tmin/Tmax$ of 0.5 °C. $Tmax$ increases by 0.8 °C against $P$ in 1998. For Central England, $Tmin$ increases against $DTR$ by 0.3 °C and $Tmax$ against P by 0.9 °C in 1989. $Tmax$ also increases against $P$ in 1911 by 0.5 °C (Table 2).
**Table 2 about here**

The stationary period is used to established regression relationships that calculate $Tmax$ and $Tmin$ from $P$ and $Tmax$, respectively. These regressions are used to estimate how $Tmax$ and $Tmin$ would have evolved during the non-stationary period. The residual is then attributed to anthropogenic regional warming and is tested using the bivariate test. Here the residuals for $Tmax$ and $Tmin$ are averaged to estimate externally-forced warming ($Tav_{ARW}$).
In SEA, $Tav_{ARW}$ shifts up by 0.5 °C in 1973 (Fig. 4). Similar patterns were found for 11 climate model simulations for SEA, undergoing a series of step changes to 2100 (Jones, 2012). For Texas, $Tav_{ARW}$ shifts by 0.8 °C in 1990. Central England temperature shifts up by 0.7 °C in 1989 and by 0.5 °C in 1911. Using the full record for Central England average temperature from 1659, a significant step change was found in 1920, whereas using a starting date of 1878 identifies 1911. Given that the



second mode identified in the longer test is 1911, we conclude the 1911 date is an artefact of the starting date in 1878 and a step change in 1920, consistent with NH data, would register if earlier data were available.

**Figure 4: Anomalies of annual mean temperature attributed to nonlinear changes where the influences of interannual variability have been removed for (a) Central England, (b) Texas, and (c) south-eastern Australia. Internal trends (dashed lines) are separated by step changes ($p<0.01$).**

None of the internal trends in Fig. 4 exceed the $p<0.05$ threshold. The trend/shift ratios for $Tav$ (not shown in Fig. 4) and attributed to external forcing ($Tav_{ARW}$) are 0.23 and 0.88, respectively for SEA, 0.45 and -0.53 for Texas and -0.01 and 0.33 for Central England (1878–2014). The lower ratio in SEA $Tav_{ARW}$ is because reduced rainfall post 1997 produces lower attributed $Tmax_{ARW}$ but if that rainfall reduction is also a response to external forcing (Timbal et al., 2010), $Tmax_{ARW}$ will be underestimated. The negative ratio for Texas is because $Tav_{ARW}$ contains negative internal trends, mostly after 1990 (largely a rainfall effect on $Tmax$). For Central England, the ratio for $Tav$ has been calculated from the long-term record from 1659, which shows no step changes or trends between 1701 and 1920. Late 20[th] century warming in both Central England and continental US elsewhere has also been analysed as nonlinear (Franzke, 2012;Capparelli et al., 2013).

These results show that the transition from stationarity to non-stationarity is abrupt for regional temperature at three locations on three continents, and for six independent climate model simulations for one of those locations (SE Australia). The close association of the observed transition in SEA in 1968 with the widespread shift date over the southern hemisphere mid-latitudes indicates that the onset of the warming signal in these broader regions is abrupt (Jones, 2012). The changes in central England in 1989 and Texas in 1990 may also be associated with a widespread step change in the northern hemisphere mid latitudes in 1987/88 (Overland et al., 2008;Boucharel et al., 2009;Lo and Hsu, 2010;Reid and Beaugrand, 2012;North et al., 2013;Menberg et al., 2014;Reid et al., 2015).

The low trend/shift ratios shown for ocean and some zonal areas also occur over the three land areas analysed. This suggests that shifts may be more distinct at regional scales, integrating into a more trend-like global average. This is the case for sea level rise data, where individual tide gauge records exhibit step ladder-like behaviour at individual locations and global mean sea level follows a curve (Jones et al., 2013).

## 4 Results – models
### 4.1 20[th] century simulations (1861–2014)

These sections report on the multi-step analysis of 102 simulations of global mean surface warming from the CMIP3 archive, and 295 simulations from the CMIP5 archive. Further information on the archives is in the SI. The relevant test for models is to identify similar phenomena to observations. Here we describe analyses of the timing of change points and their relationship with known regime changes and the measurement of the relative contributions of steps, shifts and internal trends in the temperature record (part of tests 1 and 3 listed above).





Starting with observations, the percentage of annual steps ($p<0.01$) in the 45 timeseries of mean annual surface temperature from Fig. 1, are shown in Fig. 5a. Two-thirds of all historical records shift in 1997 and one-third in 1980 and 1937. Lesser peaks of 10–15% occur in 1920, 1921, 1926, 1930, 1968–69, 1987 and 1988. The three shifts in 1979/80, 1987/88 and 1997/98 are the main contributors to the higher rate of trend noted from around 1970. Because these peaks measure how strongly steps

5       occur globally and regionally, percentages denote how pervasive a step is. The models register a significant step at the global scale only, so will only pick up the most extensive step changes – any steps occurring below the assigned level of probability ($p<0.01$) will show up as part of a trend, as is the case for 1987/88 in the observations.

Fig. 5b shows step changes from the CMIP3 combined SRES scenarios A1B and A2 simulations for the 20[th] and 21[st] century: 84 are independent and 18 are ensemble averages. The CMIP3 models were driven by observed forcing including sulphate

aerosols to 1999–2000 and not all contain natural forcings (see Table S2). They do a reasonable job of capturing the three main post-1950 peaks. Figs 5c–f show the CMIP5 RCP2.6, RCP4.5, RCP 6.0 and RCP 8.5 ensemble results, respectively. The models were driven by observed forcing, including natural volcanic and solar forcing, to 2005. Visually, the CMIP5 results illustrate the observed peaks and troughs better than CMIP3. This is presumably due to the improved representation of forcing factors and physical processes, and to improved model resolution (Table S3).

The RCP4.5 result (Fig. 5d) with 107 independent members, is the largest multi-model ensemble (MME). The three major post-1950 step changes are reproduced as follows: 55% (58 of 107) of the runs undergo a step change in 1996–98 (17% step in 1996, 16% in 1997 and 22% in 1998), 40% of the runs peak in 1976–78, just missing the observed peak in 1979/80 and 19% peak in 1986–88. In the mid-1970s, the models may be picking up the observed regime shift 1976–77 in the Pacific Ocean (Ebbesmeyer et al., 1991;Miller et al., 1994;Mantua et al., 1997;Hare and Mantua, 2000) as a contemporaneous increase in

warming. With weak El Niños affecting observations during 1977–1980 (Wolter and Timlin, 2011), this step change may have been delayed in the observed temperature record until 1979–80.

Of the pre-1950 peaks, the models peak around 1916, rather than 1920, and 1936–37 forms a minor peak, less prominent than in the observations. The volcanic eruptions of Krakatoa (1883) and Mt Agung (1963) both feature in the model simulations but less so in the observations. The mid-20[th] century period of little change is also reasonably well reproduced.

**Figure 5: Step changes in observed and simulated surface air temperatures. Frequency in percent of statistically significant step changes from (a) global, hemispheric and zonal averages (45, 1880–2014); (b) global mean warming from 102 model simulations from the CMIP3 archive for SRESA1b and A2 emission scenarios; (c–f) global mean warming 1961–2100 from the CMIP5 archive for the (c) RCP2.6 pathway (61), (d) RCP4.5 pathway (107), (e) RCP6.0 pathway (47) and (f) RCP8.5 pathway (80).**

Correlations over the full period 1880–2005 between observations and the CMIP3 and CMIP5 models, are 0.32 and 0.34, respectively (p<0.01). For the period 1950–2005, the correlations rise to 0.45 and 0.40, respectively. If specific events: 1963/64, 1968–70, 1976/77, 1979/80, 1987/88 and 1996–98 are grouped, and all other years analysed individually, then the correlation increases to 0.78 for both CMIP3 and CMIP5 records (note that this treats the simulated and observed peaks in the 1970s separately). We consider this a reasonable test, because all these dates have been linked to regime changes or break





points in temperature in the literature. Finessing the exact years involved around these events makes little difference to the result, so the correlation is robust.

Although collectively, the model ensembles reproduce the observed peaks, single models do not fare as well. We experimented with a skill score that worked on scoring matched steps between models and observations, but the resulting scores did not

correlate with any other factor. The only event reproduced widely by the models was the 1996–8 step change, peaking in 1997, where 58 of the 107 MME (55%) undergo a step change, although 40% of the MME produces a step in 1976–78.

### 4.2    Relationship between steps and trends over time

Here, we report on the relationships between steps, shifts and trends, the magnitude of warming and ECS to estimate the proportion of signal in each warming component. Total warming over time can be represented by straightforward differencing,

change measured from a simple trend and the sum of various components, such as the sum of steps, and of shifts and trends. All come up with slightly different answers, but describe a process that over many decades largely conforms to a trend.

Warming components measured here are steps, the internal trends between steps, and the shifts from one trend to the next. Counting shifts as the remainder between internal trends, preferences trends over shifts. When each is contrasted with an independent variable, such as ECS, this poses a strong test for shifts because internal trends estimate $-H_{step}$ in each timeseries.

The hindcast (1861–2005) and projection (2006–2095) components of the RCP4.5 107-member ensemble were analysed separately.

For the hindcasts (1861–2005), total warming (the 2000–05 average minus the 1861–99 average) is positively correlated with total steps (0.93, $p<0.01$). Their means are 0.97 °C and 0.94 °C, respectively. The correlation between total warming and internal trends is 0.36 ($p<0.01$) and shifts is 0.58 ($p<0.01$). Shifts therefore explain 2.5 times the variance explained by internal

trends in estimating total warming (Fig. 6a). A simple linear trend measured over the entire period has the same correlation with steps (0.93, $p<0.01$) but averages 0.76 °C, so underestimates total warming by 0.18 °C. Total warming, total steps, total shifts and total internal trends correlate poorly with ECS (-0.01, -0.01, 0.07 and -0.09, all NS, Table 4, Fig. 6b).

The ratio of total internal trends to total steps slightly favours shifts (mean 0.44), ranging between -0.09 and 1.22. A low ratio means that trends either cancel each other out or are negligible. A high ratio usually indicates the timeseries contains one or

more negative shifts and/or a number of positive trends. Observations fit comfortably within this distribution with ratios of 0.32 to 0.38, except the GISS timeseries, which has a ratio of 0.62 because of a downward shift and upward trends in the early part of the record (Fig. 6c). The MME ratios are slightly negative with respect to total warming (-0.14, NS), suggesting that the mix of shifts and trends is largely unrelated to the amount of hindcast warming (1861–2005).

For the historical period, total warming and its various components – steps, shifts or trends – are unrelated to ECS. The

relationship between total shifts and total internal trends is negative (0.47, $p<0.01$), which is to be expected, but the lack of a relationship between the shift\trend ratios and warming or ECS, suggests that this uncertainty is stochastic.

**Figure 6: Multi-model ensemble (RCP4.5, 107 members) characteristics of hindcast (1861–2005) and projected (2006–2095) periods. (a) relationship between total warming and steps, trends and shifts (1861–2005); (b) relationship between ECS and steps, trends and**



shifts (1861–2005); (c) total shifts and total trends 1961–2005 with observed points from five warming records; (d) relationship between total warming and steps, trends and shifts (2006–2095); (e) relationship between ECS and steps, trends and shifts (2006–2095); (f) total shifts and total trends 2005–2095 from individual climate models.

For the projection period, total warming over 2006–95 is based on the difference between five-year averages centred on 2006
and 2095. Total warming averages 1.55 °C, total steps average 1.57 °C and they are highly correlated (0.98, $p<0.01$). The correlation between shifts and internal trends with total warming is 0.70 and 0.74, respectively, trends having a slightly higher correlation (Fig. 6d). However, correlations between ECS, and total steps, shifts and trends, are 0.81, 0.72 and 0.43, respectively (all $p<0.01$, Fig. 6e). This shows that the timeseries are becoming more trend-like at higher rates of forcing, when compared to the hindcast period. Shifts have 2.9 times more explanatory power than trends with respect to ECS, but 0.9 times
the explanatory power with respect to total warming over 2006–2095. We take this as meaning that shifts (steps minus internal trends) carry most of the signal and that trends are more random, affected by short-term (interannual) stochastic behaviour. Some of the signal embedded in trends could also be due to shifts occurring at regional scales, which are too small to register statistically as steps at the global scale.

The ratio of trends to steps is 0.51, ranging from 0.14 to 0.88. The ratio of trends to shifts favours trend (1.22) but has a large
range (3.25 to 0.15). The correlations of both ratios with warming are very low (0.07, 0.03, respectively, NS). This seeming paradox where there is no correlation with the amount of warming but there is with ECS, when both ECS and warming are correlated, can be viewed by plotting the different modelling groups according to the relationship between shifts and trends. Individual models plot along linear pathways as was the case for the hindcast ensemble (Fig. 6c). The high sensitivity models plot towards the upper right and lower sensitivity models to the lower left. The ratios for these individual groups vary widely
– the CSIRO eight-model ensemble has trend/shift ratios of 0.25 to 0.56 and the GISS-E2-R seventeen-member ensemble ranges from 0.17 to 0.72. The potential for the same model to produce very different shift/trend ratios shows high stochastic uncertainty, probably generated by ocean-atmosphere interactions. The timing of these interactions appears to be largely unrelated to climate sensitivity, although the warming response to shifts when they do occur is related to sensitivity.

Interestingly, the GISS models form two groups, the main difference being the ocean configuration (see Schmidt et al., 2014a),
where the Russell ocean model produces more steplike outcomes and the HYCOM ocean model produces more trend-like outcomes.

For each individual decade from 1876–1875 to 2086–2095, correlations were performed between step size and ECS (Table 3). The late 19th century produces downward steps in response to the Mt Krakatoa eruption in 1883 and is negatively correlated with ECS. Positive steps dominate from 1886 through to 1945 and are positively correlated at levels of low or no significance.
The period 1946 to 1965 is negatively correlated with ECS; in 1956–65, corresponding with the 1963 Mt Agung eruption, downward steps result in a negative correlation of -0.52 ($p<0.05$). Correlations between ECS and step size become positive after 1965, being 0.41 for 1976–85 and 0.49 for 1986–95 (both $p<0.01$). For the decade 1996–2005, 101 of the 107 member MME undergo an upward step, but the correlation with ECS is only 0.19 (NS). This low correlation may partly be due to a





rebound from the negative forcing of the 1991 Mt Pinatubo eruption in the models, which has been over-estimated by about one third (Schmidt et al., 2014b). Correlations for the forcing period (2006–2095) rise to 0.68 in 2006–15 and vary between 0.57 and 0.82 for subsequent decades to 2095.

The lack of predictability in the hindcasts is a result of negative aerosol forcing due to volcanic eruptions and anthropogenic
sources occurring after 1950. The more sensitive models produce strong positive and negative responses depending on the direction of forcing, whereas in the less sensitive models this effect is reduced. This effect cancels out any consistent relationship between ECS and step size over the historical period. The implication of this finding is that the magnitude of 20[th] century warming in the models has little predictive skill and is not a reliable guide to potential future risk.

The hindcast results are also uncorrelated with the 21[st]-century projections. Total warming (1861–2005) is negatively
correlated with 21[st] century warming (2006–95, -0.25, $p\sim0.01$) and uncorrelated with respect to ECS (-0.01). Total steps from the hindcast and forecast periods show similar negative correlations. Internal trends 1861–2005 are also uncorrelated with future total warming, steps or trends. This strongly indicates that 20[th] century warming may not be a good guide to future warming, if observations are being affected in a similar way.

A final analysis looks at the explanatory power of different change models with respect to ECS over time. Linear and quadratic
trends, steps and warming to date are calculated for successive decades for each ensemble member and the results correlated with ECS. Both trends and warming difference respond to negative forcing in the first part of the record. Step changes are less volatile, remaining close to zero until increasing from 1995 and remain higher than the other models until the end of the century (Fig. 7a). The standard error measured from total accrued warming was also least out of the three statistical models. Although it would be possible to derive a closer fit for some of those models with a greater number of factors, step changes clearly carry
the greatest signal with respect to ECS over time. The analysis repeated from 1965 produces a similar result (Fig. 7b).

**Figure 7: Correlation with successive calculations of change over time using a linear trend, step changes, warming-to-date and quadratic trend; (a) to the nominated date or decadal average subtracted from 1861–99 for warming-to-date and (b) as for (a) and decadal average subtracted from 1861–1959. Dotted lines mark p<0.01.**

This result is further evidence for step changes carrying the signal. Warming-to-date assesses any warming irrespective of its cause, whereas if step changes are part of a direct response to forcing they would be a direct part of the response. This certainly seems to be case for climate models, so seems a realistic assumption to also link to observations. The advantage for using warming to date as a measure is that it has roughly a decade's advantage over statistical tests, which require hindsight, so unless the physical mechanism(s) become known, both have roughly equivalent skill at the present time.

## 4.3    21[st] Century forcing profiles

If increased forcing raises the rate of entropy production, we would expect to see steplike behaviour becoming more trend-like over time. Such behaviour would involve either:

- increase the frequency and distribution of regional step changes that integrate to become more trend-like at the global scale, or



- see an increase in the rate of diffuse warming, producing widespread trend-like behaviour.

If either is the case, then simulations for the four different emissions pathways, RCP2.6, 4.5, 6.0 and 8.5, should show this. Figs 5c–f shows the percentage of step changes in any given year for the multi-model ensemble for each of these pathways. For RCP2.6, peaks occur to about 2050, after which the ensemble stabilises. Some models step downward, the earliest of these

in 2051. Individual members stabilise between 2018 and 2092, with 48 of the final shifts being positive and 13 negative. This timing is weakly correlated with ECS (0.18, NS). ECS is uncorrelated with the size of the final shift, or to the gradient of the following trend. The RCP4.5 ensemble produces frequent steps that peak around 2025 and decline towards the end of the century. RCP6 produces a fairly constant rate of steps and RCP8.5 produces sustained steps throughout the century, peaking in the 2080s at a higher rate than 1996–98.

This evolution shows a step-ladder like process in the 20th century that changes in to an elevator-like process in the 21st becoming more trend-like with increasing forcing. Depending on the subsequent rate of forcing trend-like processes can either recede back to a steplike process or even stabilise. The HadGEM2-ES single model ensemble is used to illustrate this (Fig. 8a).

This ensemble shares the same historical forcing to 2005. It warms by less than observations to 2010, with a reversal 1964–

1980, then warms substantially in a series of steps over the next few decades. It undergoes a step change of 0.37 °C and shift of 0.18 °C in 1998, one year after the observed shift. The next step occurs in 2012, 2013, 2014 and 2015 in the four simulations, ranging from 0.40 °C to 0.49 °C in absolute terms and 0.19 °C to 0.27 °C as the shift from the pre-step trend to the post-step trend. The first half of the 21st century shows the influence of decadal variability on mediating step changes. In 2021, the RCP2.6 simulation undergoes a step change and is higher than the others for most of that decade. The RCP6.0 simulation is

lower than the others from 2025–45 before accelerating under a sustained step-and-trend process. The relative proportion of internal trends to total warming under the four scenarios is 0.34, 0.60, 0.57 and 0.79, for warming of 1.9 °C, 2.9 °C, 3.7 °C and 5.3 °C, respectively. The RCP4.5 has a higher trend ratio, showing the stochastic uncertainty inherent in the simulations.

**Figure 8: Global mean surface temperature as analysed by the multi-step bivariate test; (a) Step and trend breakdown of global**

**means surface temperature in the RCP2.6, 4.5, 6.5 and 8.0 simulations from the HadGEM-ES model, run 3; (b–e) Ti0 results from a 40-year moving window for the RCP2.6, 4.5, 6.5 and 8.0 simulations, respectively.**

Like most statistical tests that detect change points, the bivariate test is considerably weakened under autocorrelated data, where its timing is fairly robust but $p(H_0)$ is sensitive. Such autocorrelations may be caused by simple trends, lag-1 or longer lag processes influencing the complex nature of warming. Removing these without assuming an underlying process is difficult,

so one way of assessing the influence is to pass a moving window through a timeseries. If the data is steplike and largely free of autocorrelation, a distinct step will produce a line of horizontal $Ti_0$ statistics on a single date as it passes through the window. If there are no steps within a window period and autocorrelation is low, background $Ti_0$ values will return to low values (single



digits). With autocorrelation, background $Ti_0$ values remain above the p<0.01 threshold and form a 'cloud', rather than steps producing horizontal lines.

In Fig. 8b–e, successive horizontal lines extending right from low $Ti_0$ values indicate step-ladder-like behaviour in the 20[th] century. Horizontal lines that stay on the right without returning to low $T_{i0}$ values indicate both steplike and trending behaviour.

A cloud to the far right, as in Fig. 8e, shows a trend-dominated process. Summarising 21[st] century behaviour under increasing emissions, RCP2.6 shows a return to steplike changes, stabilising around 2050, RCP4.5 shows a return to steplike change late century, RCP6.0 shows increasing trend-like behaviour over the century and RCP8.5 shows a consistent trend to the end of the century, with few steps.

An indication of change at the regional scale and how it may relate to global change is illustrated by using selected CMIP3

models for SE Australia as described in Jones (2012). For example, for the CSIRO Mark3.5 A1B simulation, for global mean warming, internal trends comprise 52% of total warming 2006–2095, whereas for SEA *Tmax* the ratio is 13% and *Tmin* 47%. These were consistent for A1B and A2 forced simulations, which are roughly equivalent to RCP4.5 and 6.0. The number of step changes is also notable: four and five at the local scale and twelve at the global scale (Fig. 9). The higher ratio for *Tmin* compared to *Tmax* may be due to *Tmin* being related to large-scale sea surface temperature patterns and *Tmax* being related to

more local soil moisture patterns as is the case for the central and western United States (Alfaro et al., 2006). Jones et al. (2013) showed that such changes at the local scale produce significant increases in impact risks.

**Figure 9: Anomalies of annual mean temperature showing internal trends separated by step changes from the CSIRO Mk3.5 A1B simulation; (a) maximum temperature southeastern Australia; (b) maximum temperature southeastern Australia; (c) global mean**

**surface temperature Internal trends (dashed lines) are separated by step changes (*p*<0.01).**

These analyses do not support increasing trend-like behaviour at the local scale, and therefore favours the first alternative above, but further work across more regions is required to confirm this.

## 5  Severe testing of steps and trends

Earlier sections have identified steps and trends in temperature and tested how trend, step and trend-shift relationships relate

to total warming the independent variable ECS. This section examines how well trend, step and step-trend models reproduce the temperature records examined throughout the paper. This tests $h_{trend}$ against $h_{step}$. The error value assigning $p<h_0$ is not the principal measure being sought. Instead, the statistical model that combines low error with unstructured residuals while sustaining physically plausible assumptions is preferred. Another aim is, if possible, to provide likelihoods for severe testing. Four statistical models are tested: ordinary least squares trend, LOWESS, step, and step and trend. The LOWESS model was

applied with a bandwidth of 0.5 to assess sensitivity to fluctuations in the data, contrasting those with both the trend and step model. It is not considered a valid statistical rival because it is fitted without regard to physical process. Likewise, although the step and trend model will fit well to the data, the step model is the one used for severe testing, being a straightforward measure of $h_{step}$. The trend model represents $h_{trend}$.





With the data produced, we look at goodness of fit ($r^2$), the residual sum of squares (RSS), cumulative ($\sum R$) residuals and cumulative residuals squared ($\sum R^2$). Residuals (R) show how much variance is explained by the model, cumulative residuals will show whether residuals are showing structure not explained by the model and cumulative residuals squared show accumulating error, including rapid changes not accounted for. To these have been added four more tests: F-tests for

autocorrelation (F-auto) and heteroscedasticity (F-hetero) of the residuals over the whole record and percentage of exceedance over moving 40-year windows. White's test (White, 1980) is used for heteroscedasticity. The first four of these tests use absolute error, or the amount of a timeseries not explained by the statistical models and the second four show patterns, working on accuracy and precision, respectively. The statistical models that fail a combination of both are therefore the weakest. Results are shown in Fig. 10 and Tables 4 and 5. The data and statistical models for HadCRU record 1880–2014 are shown in

Fig. 10a. Cumulative residuals that track close to zero (Fig. 10b) show the model mimicking the data closely and sustained departures show significant deviation. Here, the trend model deviates substantially and the LOWESS model less so, while the step and step and trend models deviate least. This follows through to the cumulative residuals squared. The less change the better; whereas upward kinks show rapid changes or large outliers (positive or negative) not incorporated into the model (Fig. 10c). Trend analysis produces an $r^2$ value of 0.76 and residual sum of squares of 0.87, and the other three statistical models

have an $r^2$ of 0.87 and RSS of 0.8. For $\sum R^2$ the trend model behaves more poorly than the other three.

**Figure 10: Testing three models to mean global anomalies of surface temperature from the HadCRU record, 1880–2014 (a–c) and 1965–2014 (d–f); (a) and (d) mean annual anomalies and linear, step change and shift and trend models; (b) and (e) cumulative residuals for each model, where success is measured as tracking close to zero; (c) and (f) cumulative sum of residuals squared, where**

**upward steps show nonlinearity not explained by each model.**

With the autocorrelation and heteroscedasticity tests, the LOWESS test performs less well than the other two for the 40-year tests. Although the LOWESS model performs well over the whole record, it is subject to deviations within the record that cancel each other out – akin to cutting corners. The step and trend model does worst for F-hetero over the whole record, but the best over 40-year windows. This is due to high variance within the early part of the record and is an issue of precision, as

standard error of this relationship is almost half that of the trend model (not shown, but is similar to the $\sum R^2$ relationship). The step model is clearly superior to the trend model for the moving window tests. The results for the other four long-term global warming records: BEST, C&W, GISS and NCDC, are not shown but have similar results. These tests, omitting LOWESS, were carried out for HadCRU 1965–2014, a period with a sustained radiative forcing signal (Fig. 10d). The results for the different statistical models are similar, with $r^2$ values of 0.85, 0.86 and 0.89, respectively. The

step and trend model is still the best performed, but the step model only slightly better than the trend model – this is due to the northern hemisphere shift in 1987/88 being incorporated into the into the trend at global scale, where each of the models are statistically identical., Dividing this timeseries into quarters will bring 1987/88 into the picture but also make both the MYBT and t-test test more sensitive.





**Table 4 about here**

Also shown in Table 4, are the zonal temperatures from NCDC 30°N–60°N (1880–2014) where total internal trends are slightly
negative (-0.04 °C) and shifts are positive (1.13 °C or 106% of steps). The pattern of results is similar to those for the global
HadCRU record but the residuals are slightly more than double and the cumulative residuals almost double. The step model is
clearly superior to the trend model, which fails White's test for the whole record, fails the 40-year F-auto at a level of 51% and
has an RSS double that for steps. This record is entirely made up of steps, showing the lack of trend occurring within some
regions.

The quarterly record of HadCRU from Fig. 3 (1965–2014) is more fine-grained, incorporating the 1987/88 shift (Table 4). If
warming is gradual, the results for trends should be scalable, however, they perform less well at this timescale. The respective
$r^2$ results are 0.69, 0.72, 0.75 and 0.76, whereas the differences in the cumulative residuals are 2.0, 0.5, 0.7 and 0.2, where zero
is a perfect score. Here, the LOWESS model performs similarly to the step model because it closely follows the data. The step
model performs better than the trend model for HadCRU quarterly data, and almost as well as the step and trend model. For
the GISS quarterly data, the results are similar.

The satellite records are more steplike than surface temperature when measured using cumulative residuals. The step and trend
model for the 40-step window heteroscedasticity tests for satellite data fails for both RSS and UAH. This is due to two instances
of short-term departures on an otherwise stable background that measures heteroscedasticity as significant with the F-test: 1)
a warm period during 1998, which is represented as a single step but lasts four quarters and 2) a small warming event associated
with an El Niño event in 2010 lasting two quarters. Removing this short-term warming from these sequences removes the
heteroscedasticity. So although not all deviations are removed by representing the satellite record as being stepwise, it still
provides a better explanation of change than the trend model.

Simulated global annual mean surface temperatures from climate models show results consistent with observations (Table 5).
The data from Fig. 7 were analysed in the same way, except that quadratic (RCP4.5, RCP6.0), cubic (RCP8.5) and quartic
(RCP2.6) polynomial functions were used instead of a linear trend. The LOWESS model used here at 0.5 record length is
relatively low resolution providing 120-year smoothing. The step model outperforms both the trend and the LOWESS model
in all simulations, with the exception of the RSS in the RCP8.5 simulation. The RCP2.6 simulation is the most steplike. In the
RCP4.5 simulation, the step model does slightly worse than in the RCP6.0 simulation, which is actually more steplike. This
shows the role of stochastic uncertainty in the warming process as portrayed in Fig. 6f. The RCP8.5 simulation is the most
trend-like; the step model fails in the final decades of the 2st century because the bivariate test detects no steps, but the climate
continues to warm. This is what we would expect if shifts became more local and more frequent, integrating into a curve at
the global level, much like sea level rise does today.

**Table 5 about here**



### 5.1    Severe testing summary

A range of statistical tests have been used to examine $h_{step}$ and $h_{trend}$ as representatives of scientific hypotheses $H1$ and $H2$. This is consistent with the substantive null described by Mayo and Cox (2010) where we have applied tests designed to provide a clear choice between $H1$ and $H2$. The focus is on whether atmospheric warming is gradual, forming a monotonic trend, or is stepwise and periodic, forming a complex trend over time. To paraphrase Mayo and Cox (2010): hypothesis $H1$ predicts that $h_{trend}$ is at least a very close approximation to the true situation; rival hypothesis $H2$ predicts a specified discrepancy from $h_{trend}$, and the test has a high probability of detecting such a discrepancy from $H1$ were $H2$ correct. Detecting no discrepancy is evidence for its absence.

As stated in the introductory sections, no single test can undertake that task. We rely on the multi-step Maronna-Yohai bivariate test to identify step changes in the input data but beyond that make as few assumptions as possible. A total of six probative tests with links to the two substantive hypotheses were proposed earlier in the paper – these are designed to pinpoint discrepancies between $H1$ and $H2$ by analysing the global warming data they seek to explain. The data generated consists of steps, trends and shifts applying the multi-step MYBT model and trends applying least squares trend analysis. The use of models such as LOWESS are for sensitivity testing and not part of the probative assessment.

The results of the probative tests are summarised through the following findings:

**Stratified analysis of change points**

- Global and regional analyses of steps show a highly coherent pattern of change points, where warming in the second half of the 20th century aligns with known regime changes associated with changes in decadal variability (Table 6). These events comprise the major proportion of historical warming to 2014.

- Analysis of steps, internal trends and shifts in observations attributes higher proportions of warming to shifts at the zonal scale (up to 100%), moving to lower proportions at the global scale. Three regional assessments also contain high shift/step ratios, with trends playing a lesser role.

- This effect is larger in the mid-latitude regions and with SST, indicating the role of equator-to-pole hydrothermal transport of energy in the ocean-atmosphere system. Their timing shows a strong role is being played by decadal variability.

- Surface and satellite temperatures undergo contemporaneous shifts at the global scale, largely removing the discrepancy between trends within the two data sets. Both surface and satellite temperature records are very steplike, with surface trend/shift ratios of 0.19 and 0.27 and satellite ratios of -0.55 and -0.40 showing the effect of internal trends. Shifts are consequently higher than steps in the satellite data.

**Similar patterns of change in observations and physical models**

- Correlations between step change frequency in the observed 44-member group of global and regional data and the CMIP3 and CMIP5 MMEs analysed (1880–2005), are 0.32 and 0.34, respectively ($p<0.01$). For the period 1950–2005, correlations rise to 0.45 and 0.40, respectively. Grouping specific events (1963/64, 1968–70, 1976/77, 1979/80,



1987/88 and 1996–98) and analysing other years individually, correlation increases to 0.78 for both CMIP3 and CMIP5 records. Variations in forcing, especially volcanoes may affect the timing and direction of step changes, but they are not their sole cause, given that 21st century simulations produce step changes from smoothly varying changes in forcing.

• Fifty-eight members of a 107-member MME (CMIP5 RCP4.5) show a step change in 1996–98 reproducing the observed change in 1997 within ±1 year.

### Nonlinear components of warming carry more of the signal than linear components

• Analysis of steps and trends in observed and model data shows that steps explain change better than trends when the structure of the residuals area assessed using goodness of fit, residual sum of squares, cumulative residuals,
cumulative residuals squared, autoregressive residuals testing and White's test for heteroscedasticity.

• For simulated historical warming 1861–2005, the $r^2$ values for steps, shifts and trends in explaining total warming are 0.87, 0.43 and 0.13, respectively. Simulated warming for this period is not correlated with ECS.

• For the 21st century (2006–2095) the $r^2$ values for steps, shifts and trends in explaining total warming are 0.96, 0.54 and 0.49, respectively. The $r^2$ values for steps, shifts and trends in explaining ECS are 0.65, 0.52 and 0.18,
respectively.

### Stationary and non-stationary periods are separated by step changes

• In all three locations on three continents tested, and for six independent climate model simulations for SE Australia, warming commenced with a step change in *Tmin* and sometimes *Tmax*. Warming is not slowly emergent in any of this data as would be expected if warming is gradual. The coincident timing of shifts in SE Australia with southern
hemisphere step changes and those in the UK and USA with northern hemisphere changes, suggest that warming has commenced abruptly in different areas of the globe at different times, and that the separation between stationarity and non-stationarity in the temperature record is abrupt.

### Other variables show similar step changes

• Step changes exhibiting similar timing have been shown for tide gauge observations, rainfall, ocean heat content,
forest fire danger index and a range of other climate variables, in addition to many impact variables. These are overwhelmingly attributed to random climate variability, including abrupt changes identified as part of decadal regime change.

### The best representations of underlying step- and trend-like structures in the data.

• For observations and selected model data the simple step-ladder model performs better than the monotonic trend
model for goodness of fit ($r^2$), the residual sum of squares (RSS), cumulative ($\sum R$) residuals and cumulative residuals squared ($\sum R^2$), White's test for heteroscedasticity, a moving 40-year window regression of the residuals and a moving 40-year window White's test.





Table 6 summarises the major tests undertaken with expected outcomes for $h_{trend}$ and $h_{step}$. While objections could be made to each of these on an individual basis, collectively they show that for externally-forced warming on decadal scales, $h_{step}$ is better supported than $h_{trend}$. However, long-term warming (greater than ~50 years) is largely trend-like, and is proportional to the amount of forcing.

In summary, these tests show that $h_{step}$ is a close approximation of the data when analysing decadal-scale warming. Over the long term, this warming conforms to a complex trend that can be simplified as a monotonic curve, but the actual pathway is steplike. As outlined in Section 3.3, this rules out gradual warming, either in situ in the atmosphere or as gradual release from the ocean, in favour of a more abrupt process of storage and release. The precise mechanisms by which this occurs remain to be determined. This conclusion supports the substantive hypothesis *H2* over *H1*, where the climate change and variability

interact, rather than vary independently.

**Table 6 about here**

## 6    Discussion

It would be reasonable to ask the question – if shifts in temperature and other variables are ubiquitous within the climate

system, why have they not been recognised earlier? This question has been explored at length in related papers that cover the following points:

- The history and philosophy of gradualisms and trend analysis as its key tool for understanding how the world works has its origins in the scientific enlightenment and since then has defined H1 as the dominant paradigm of climate change (Jones, 2015b). This has been reinforced by the success of methods for long-term trend analysis (Jones,

20          2015a). Phenomena that do not fit this model are labelled as noise and considered to be random.

- The value-laden framing of the signal-to-noise model in defining what information is useful for decision making and what is not (Koutsoyiannis, 2010;Jones, 2015b).

- The great success of ordinary least squares and related tests that use linear statistical methods in explaining climate phenomena, covering methods such as timeseries analysis, pattern matching (Santer et al., 1990;Mitchell,

25          2003a;Hasselmann, 1993), vector analysis and its application to understanding climate processes (North et al., 1995;Hasselmann, 1979), detection and attribution (Hegerl et al., 2007;Stott et al., 2010) and development of climate projections (Whetton et al., 2005;Hulme and Mearns, 2001;Mitchell, 2003b).

- The difficulty in analysing change points in complex data and achieving a clear error judgement using Neyman-Pearson testing (Type I and II errors using $pH_0$<threshold), and linking that to specific hypotheses (see Mayo, 2010).

- The climate wars, specifically the role of steps and trends, where trends are associated with climate change theory and steps with opposition to the theory (Skeptical Science, 2015;Foster and Abraham, 2015;Cahill et al., 2015;Lewandowsky et al., 2015a;Lewandowsky et al., 2015b). This has become a situation where methods are being held as representative of particular theoretical positions (Jones and Ricketts, 2016).





- The cognitive values attached to parsimony or Occam's razor, where a phenomenon should be described in the simplest terms possible. Applied to statistics, its main aim is to avoid over-fitting. However, in a complex physical system, a statistically simple relationship may be energetically complex (Jones and Ricketts, 2016). This has not been a factor under consideration to date.

- The link between model skill and predictive capacity is defined by the analytic framework applied. For example, seamless links between weather and climate forecasting over a range of timescales are a key scientific target (Palmer et al., 2008;Hoskins, 2013). The Global Framework for Climate Services (World Meteorological Organization, 2011), reflects this: *Weather and climate research are closely intertwined; progress in our understanding of climate processes and their numerical representation is common to both. Seamless prediction (on timescales from a few hours*

   *to centuries) needs to be further developed and extended to aspects across multiple disciplines relevant to climate processes* (World Meteorological Organization, 2010). Solomon et al. (2011) state that *"Long experience in weather and climate forecasting has shown that forecasts are of little utility without a priori assessment of forecast skill and reliability"*. The assumption that the processes involved are timescale invariant indicate that the meaning of seamless has not really been thought through. For the moment, seamless means a concentration on mean change and other

   variables that show skill in climate models. However, skill is measured according to the H1 signal to noise construct and would like quite different if analysed in H2 mode (Jones, 2015b). This framing also overlooks the considerable literature on scenarios that has arisen because long-term predictions under considerable uncertainty tend to fail (Wack, 1985a, b;Börjeson et al., 2006).

- The different areas of scientific knowledge and expertise required to understand the climate system. In particular, the

   relative roles of radiative physics largely understood as being linear and hydrometeorology, with its substantial nonlinear behaviour, remain largely unreconciled.

If mean global atmospheric warming is accepted as an ill-posed mathematical problem, a single test passing a $pH_0$ threshold cannot adequately represent the various influences present, and a more applied approach is required. This involves undertaking severe statistical testing informed by a process-based understanding of how the climate may change.

25   Technically, trend analysis and the Maronna-Yohai bivariate test face similar limitations, with respect to the serial independence and normal distribution of the input data, but the former has widespread acceptance whereas the latter is unfamiliar, creating different degrees of trust. Objectively, if the data they analyse is subject to lagging or unit-root processes then the likelihoods expressed by either test will be compromised (e.g., Cohn and Lins, 2005;Koutsoyiannis, 2010). We have been quite open about this with respect to the bivariate test, and it has informed how these tests are applied here.

30   Most challenges to trend-like behaviour in surface warming are associated with contrarian positions that either seek to repudiate the theory of greenhouse gas driven climate change entirely, or maintain that the risk described by groups such as the IPCC is overstated (McKitrick, 2014, 2015;Tisdale, 2015). In particular, this controversy has surrounded the question of whether



warming paused or entered a hiatus from about 1998 or continued unabated. Much of the recent statistical analysis of global atmospheric warming has concerned this issue.

For example, Cahill et al. (2015) recently published a segmented trend model for global mean surface temperature with change points around 1912 (1907–1920, $p<0.05$ limits), 1940 (1934–1948) and 1970 (1963–1979). If this model is subject to the same

tests as in Tables 4 and 5 for the five mean global surface warming records in Fig. 2, the results are similar to the step and trend model used here, so it does produce very low residual error. Using likelihood ratios or any similar measure does not distinguish between the segmented trend and step and trend models, therefore a focus on the probative aspects of severe testing is required, linking $h_{trend}$ with *H1*.

Cahill et al. (2015) used a Bayesian belief approach, stating that step changes are physically implausible: "Isolated pieces of

trend line with sudden temperature changes between them (i.e. a 'stairway model') would not provide a physically plausible model for global temperature given the thermal inertia of the system". Although they do not specify the exact physical process, this presumably refers to the thermal inertia of the ocean. This conforms with the description of the ocean-driven component of *H1* described in Sect 2.3: most of the heat generated by added greenhouse gas forcing goes into the ocean and is gradually released into the atmosphere, mediated by the rate of shallow and deep-ocean mixing. It is not clear whether Cahill et al. (2015)

refer to a process whereby heat absorbed into the ocean at varying rates, or is released by the ocean at varying rates, both alternatives potentially being mediated by the relationship between shallow and deep-ocean mixing.

In any case, Cahill et al. (2015) explicitly reject $h_{step}$. In doing so they are implicitly claiming to meet Slingo's (2013) caution that a statistical model be well specified. In their conclusions they restate *H1* – "recent variations in short-term trends are fully consistent with an ongoing steady global warming trend superimposed by short-term stochastic variations" (Cahill et al., 2015).

Foster and Abraham (2015) reject discontinuous change for the same reason and extensively test trend-related models to reach a similar conclusion. Such claims also implicitly reject a host of studies that have detected step changes in temperature data (Table 6). Presumably if those studies have all committed Type I errors, the models involved in detecting such changes – the bivariate test, Rodionov's (2006) STARS test and others, would also be invalidated for homogeneity testing of climate data, their other main use. If step changes cannot physically exist in the data, the tests that have detected them are invalid and the

homogeneity adjustments to climate records made on the basis of such tests are likewise invalid. This suggests that such arguments have a very narrow focus and are inconsistent with the bigger picture.

Many studies have applied statistical techniques to extract the noise from temperature data to diagnose the signal. For example, Foster and Rahmstorf (2011) remove solar, volcanic and ENSO influences through multilinear regression on a monthly basis from 1979–2010, concluding that the remaining data more closely follow a single trend. However, if ENSO is coupled with

regime changes and steplike warming, the regression relationships for ENSO will contain part of the signal. Due to constraints limiting those observations to the satellite-observation period, the fitting data is also the test data. Zhou and Tung (2013) undertook a similar analysis (1856–2010) using non-satellite data for ENSO, adding the Atlantic Meridional Oscillation (AMO), which results in a lower trend for the latter period analysed by Foster and Rahmstorf (2011). However, the AMO is likewise potentially involved in nonlinear changes on decadal timescales, involving rapid shifts in temperature.





A series of studies has explicitly examined climate shifts in oscillatory modes of climate variability on decadal time scales (Swanson et al., 2009;Swanson and Tsonis, 2009;Tsonis and Swanson, 2012;Tsonis and Swanson, 2011;Wang et al., 2009;Wang et al., 2012). If these signals are extracted, a monotonic accelerating curve during the 20th century remains (Swanson et al., 2009). Although they describe these as climate shifts that have the same timing as those in this paper, according

to Tsonis and Swanson (2012), these shifts manifest as a change in global temperature trend. This frames shifts as modulating trends, whereas our analysis suggests that the shifts are primary and trends are secondary.

As we discuss in a related paper where *H2* is examined in greater detail, the *H1* hypothesis comes from a radiative-centric view of the climate system, which if it considers hydrodynamics at all, regards it as an independent process (Jones and Ricketts, 2016). In a coupled climate system, this makes little sense. Radiative processes are additive (Ozawa et al., 2003), so cannot

supply heat energy in bursts unless directly forced.

However, hydrodynamic processes are nonlinear and are quite capable of supplying the energy required (Ozawa et al., 2003;Lucarini and Ragone, 2011;Ghil, 2012). To suggest the step-wise release of that heat energy is physically implausible overlooks the energetics of the ocean-atmosphere system. The atmosphere contains as much heat energy as the top 3.2 m of ocean (Bureau of Meteorology, 2003). About 93% of historically added heat currently resides in the ocean (Roemmich et al.,

2015), thirty times that of the atmosphere. Between 1955 and 2010, the amount of heat added to the atmosphere was about $0.8 \times 10^{22}$ Joules, compared with the $24.0 \times 10^{22}$ Joules added to the top 2000 m of the ocean (Levitus et al., 2012). A physical re-organisation of the ocean-atmosphere system, as part of a regime shift, is large enough to provide the relatively small amount of energy required to cause abrupt sea surface and atmospheric warming.

For example, Reid et al. (2015) in describing the late 1980s regime shift, show it was associated with a large scale shift in

temperature and multiple impacts across terrestrial and marine systems, mainly in the northern hemisphere. Changes in the North Pacific in 1977 were considered even more extensive (Hare and Mantua, 2000) as were those in 1997–98 that involved both the Pacific and Atlantic Oceans (Chikamoto et al., 2012a;Chikamoto et al., 2012b).

For example, Reid et al. (2015) in describing the late 1980s regime shift, show it was associated with a large scale shift in temperature and multiple impacts across terrestrial and marine systems, mainly in the northern hemisphere. Changes in the

North Pacific in 1977 were considered even more extensive (Hare and Mantua, 2000) as were those in 1997–98 that involved both the Pacific and Atlantic Oceans (Chikamoto et al., 2012a;Chikamoto et al., 2012b).

One important test for a hypothesis is whether it can offer explanations and/or novel predictions not contained within the original scope. This has not been the aim of much recent work, which is focused on whether the period after 1998 was a hiatus, pause or uninterrupted trend. The aim of papers like Cahill et al. (2015) and Foster and Abraham (2015) was to show that the

year 1998 was unexceptional and that the so-called 'pause' was part of a longer term trend. Underpinning this claim is that there are no steplike changes within the last part of the record from 1970–2014. The most extraordinary claim of this type was made by Rajaratnam et al. (2015) who examined the segmented trends either side of 1998, concluding they were statistically identical. In doing so, they illustrated those trends as being separated by a gap of almost 0.2 °C as shown in Fig. 2, which they



completely ignored in their analysis. Elsewhere, we examine the 1997/98 paper in greater detail, asking whether the climate system is currently undergoing another shift in warming (Jones and Ricketts, 2016).

Meehl (2015) and Slingo (2013) emphasise the importance of having a process-based understanding of how the temperature changes. Temperature change on periods of less than fifty years has not been severely tested in this regard, much of the recent

work being defensive rather than innovative.

### 7 Conclusions

Here, we have adapted and applied severe testing principles proposed by Mayo and Spanos (2010) to determine the role step changes play in decadal-scale warming. This involves the linking of scientific hypotheses $H$ with statistical hypotheses $h$, in order to test $h/H$ and $-h/-H$ to the point where the test agrees/disagrees with the hypothesis/null with a very high probability of

distinguishing between the two. Specifically, the scientific hypothesis – that externally-forced and internally-generated climate processes interact with each other (*H2*) instead of acting independently (*H1*) – is shown by the statistical hypothesis $h_{step}$ passing a series of steps in better shape than $h_{trend}$.

This finding does not invalidate the huge literature that assesses long-term (>50 years) climate change as a relatively linear process, and the warming response as being broadly additive with respect to forcing (e.g., Lucarini et al., 2010;Marvel et al.,

2015). However, on decadal scales, this is not the case – warming appears to be largely governed by a storage and release process, where heat is stored in the ocean and released in bursts projecting onto modes of climate variability as suggested by Corti et al. (1999). We discuss this further in another paper (Jones and Ricketts, 2016).

This has serious implications for how climate change is understood and applied in a whole range of decision-making contexts. The characterisation of changing climate risk as a smooth process will leave climate risk as being seriously underdetermined,

affecting how adaptation is perceived, planned and undertaken (Jones et al., 2013).

The interaction of change and variability is typical of a complex, rather than mechanistic, system. The possibility of Lorenzian attractors in the ocean-atmosphere acting on decadal time scales was raised by Palmer (1993) and, despite later discussions about the potential for nonlinear responses on those timescales (e.g., Lucarini and Ragone, 2011;Tsonis and Swanson, 2012), very little progress has been made in translating this into applied research that can portray a better understanding of changing

climate risk. This may be due in part to science asking the wrong questions.

The signal to noise model of a gradually changing mean surrounded by random climate variability poorly represents warming on decadal timescales. The separation of signal and noise into 'good' and 'bad', likewise, is poor framing for the purposes of understanding and managing risk in fundamentally nonlinear systems (Koutsoyiannis, 2010;Jones, 2015b). However, as we show, the presence of such changes within climate models shows their current potential for investigating nonlinearly changing

climate risks. Investigating step changes in temperature and related variables does not indicate a need to fundamentally change how climate modelling is carried out. It does, however, indicate a need to change how the results are analysed.



**Code availability**

With Supplementary Information as a zip file (Python and R modules)

**Data availability**

With Supplementary Information as Excel files

**Team list**

Roger N. Jones, Victoria University
James H. Ricketts, Victoria University

**Author contributions**

RJ conceived the study, JR coded and tested the multi-step model, RJ developed the severe testing regime for the results and
with JR undertook analyses, JR put together the SI and maintained quality control, RJ led the paper with contributions from
JR

**Competing interests**

The authors declare that they have no conflict of interests.

**Acknowledgements**

JR is the holder of a Victoria University posgraduate research scholarship. Data sources include the Met Office Hadley Centre,
National Aeronautics and Space Administration Goddard Institute for Space Studies and United States National Climatic Data
Center, Berkely Earth, Cowtan and Way and the Australian Bureau of Meteorology. CMIP3 and CMIP5 archives are made
available by the modeling groups, the Program for Climate Model Diagnosis and Intercomparison (PCMDI) and the WCRP's
Working Group on Coupled Modelling (WGCM). The U.S. Department of Energy's Program for Climate Model Diagnosis
and Intercomparison provides coordinating support and led development of software infrastructure in partnership with the
Global Organization for Earth System Science Portals. D Kelly O'Day provided the macro templates, which has been adapted
to provide the step and trend charts.





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



**Table 1. Dates of step changes for lower tropospheric satellite temperature anomalies, with annual timeseries and quarterly breakdowns in parentheses (DJF, MAM, JJA, SON), and quarterly timeseries. Data sources are Remote Sensing Systems (RSS) and University of Alabama, Huntsville (UAH).**

| Region | Annual timeseries (quarterly breakdown) | | Quarterly timeseries | |
|---|---|---|---|---|
| | RSS | UAH | RSS | UAH |
| Global land & ocean | 1995 (98,98,95,95) | 1995 (97,98,94,95) | JJA 1997 | SON 1997 |
| Global land | 1995 (95,98,95,95) | 1998 (98,98,94,95) | SON 1994 | SON 1997 |
| Global ocean | 1998 (98, - ,97,95) | 1995 (97, - , - ,95) | JJA 1997 | SON 1997 |
| NH land & ocean | 1995 (98,98,94,94) | 1998 (98,98,94,94) | JJA1997 | SON 1997 |
| NH land | N/A | 1998 (98,98,98,98) | N/A | JJA 1997 |
| NH ocean | N/A | 1994 ( - , - , - ,94) | N/A | JJA1997 |
| SH land & ocean | 1995 (98, - , - ,95) | 1995 (97, - , 87,95) | SON 1997 | SON 1997 |
| SH land | N/A | 1995 (95, - , 91,95) | N/A | MAM 2002 |
| SH ocean | N/A | 1995 (97, - , - ,95) | N/A | DJF 1998 |
| Tropics land & ocean | 1995 ( - , - , - ,93) | - ( - , - , - ,95) | JJA1997 | JJA1997 |
| Tropics land | 1995 ( - , - , - ,87) | 1995 (98, - , 95,95) | SON 1997 | JJA1997 |
| Tropics ocean | 1995 ( - , - , - ,95) | - ( - , - , - , - ) | JJA 1997 | - |
| NH ex-trop land & ocean | 1998 (95,98,98,94) | 1998 (98,98,98,94) | SON 1997 | DJF 1998 |
| NH ex-trop land | 1998 ( - ,98,94,94) | 1998 ( - ,98,98,98) | MAM 1994 | DJF 1998 |
| NH ex-trop ocean | 1998 (99,98,98,94) | 1994 (02,98, - ,94) | SON 1997 | MAM 1998 |
| SH ex-trop land & ocean | 1998 (96, - , - ,95) | 1996 (97, - , - ,95) | DJF 1998 | DJF 2001 |
| SH ex-trop land | 1995 ( - , - , - , - ) | 2001 (03, - , - ,02) | JJA 1995 | MAM 2002 |
| SH ex-trop ocean | 1998 (96, - , - , - ) | 1996 (97, - , - ,95) | DJF 1998 | DJF 1998 |
| N polar land & ocean | 1995 (03,95,98,95) | 1995 (05,95,98,95) | DJF 2000 | MAM 1998 |
| N polar land | 1995 ( - ,94,98,95) | 1995 ( - ,89,98, - ) | DJF 2005 | MAM 2000 |
| N polar ocean | 1995 (03,05,98,95) | 1995 (05,95,98,95) | MAM 2002 | MAM 1998 |
| S polar land & ocean | - | - | - | - |
| S polar land | - | - | - | - |

**Table 2. Year of non-stationarity in regional temperature for south-eastern Australia, Texas and Central England. Data source, year of first change greater than one standard deviation for *Tmax* against *P* and *Tmin* against *Tmax, or DTR/P* using the bivariate test. The stationary period is also shown.**

| Data source | *Tmax*/*P* | | *Tmin*/*Tmax* | | *DTR*/*P* | | Stationary Period |
|---|---|---|---|---|---|---|---|
| | Year | Change | Year | Change | Year | Change | (SEA) |
| SE Australia | 1999 | 0.7 | 1968 | 0.6 | | | 1910–1967 |
| Texas | 1998 | 0.8 | 1990 | 0.5 | | | 1895–1990 |
| Central UK | 1989 | 0.9 | N/S | | 1989 | 0.3 | 1878–1988 |
| | 1911 | 0.5 | | | | | |





**Table 3. Steps collated for each decade from 1876 to 2195 from the RCP4.5 MME, showing total steps up and down and the correlation between step size and ECS. The second part of the table shows the correlations between total warming, steps and trends over the observed and simulated periods and ECS. Correlations are classified as not significant (NS, $p>0.05$), $p<0.05$ (*) and $p<0.01$ (**). Total correlations with the MME are n=107 and with ECS are n=92.**

| Change and period | Steps up | Steps down | Correlation with ECS | Significance |
|---|---|---|---|---|
| Steps 1876–1885 | 0 | 26 | -0.40 | * |
| Steps 1886–1895 | 13 | 1 | -0.32 | NS |
| Steps 1896–1905 | 7 | 1 | -0.09 | NS |
| Steps 1906–1915 | 31 | 0 | 0.27 | NS |
| Steps 1916–1925 | 65 | 0 | 0.27 | * |
| Steps 1926–1935 | 17 | 1 | 0.09 | NS |
| Steps 1936–1945 | 33 | 0 | 0.20 | NS |
| Steps 1946–1955 | 6 | 1 | -0.85 | * |
| Steps 1956–1965 | 4 | 12 | -0.52 | * |
| Steps 1966–1975 | 29 | 0 | 0.33 | NS |
| Steps 1976–1985 | 56 | 0 | 0.41 | ** |
| Steps 1986–1995 | 34 | 0 | 0.49 | ** |
| Steps 1996–2005 | 101 | 0 | 0.19 | NS |
| Steps 2006–2015 | 83 | 0 | 0.68 | ** |
| Steps 2016–2025 | 82 | 0 | 0.65 | ** |
| Steps 2026–2035 | 70 | 0 | 0.74 | ** |
| Steps 2036–2045 | 82 | 0 | 0.66 | ** |
| Steps 2045–2055 | 75 | 0 | 0.57 | ** |
| Steps 2056–2065 | 65 | 0 | 0.67 | ** |
| Steps 2066–2075 | 61 | 0 | 0.60 | ** |
| Steps 2076–2085 | 51 | 0 | 0.66 | ** |
| Steps 2086–2095 | 27 | 0 | 0.82 | ** |
| | Mean ( °C) | Range ( °C) | | |
| Warming 1861–2005 | 0.9 | 0.4–1.4 | -0.01 | NS |
| Warming 2006–2095 | 1.5 | 0.7–2.4 | 0.81 | ** |
| Steps 1861–2005 | 1.0 | 0.3–1.5 | -0.01 | NS |
| Steps 2006–2095 | 1.6 | 0.7–2.5 | 0.81 | ** |
| Shifts 1861–2005 | 0.6 | 0.0–1.2 | 0.07 | NS |
| Shifts 2006–2095 | 0.8 | 0.3–1.5 | 0.72 | ** |
| Trends 1861–2005 | 0.4 | 0.0–1.0 | -0.09 | NS |
| Trends 2006–2095 | 0.8 | 0.1–1.6 | 0.43 | ** |




**Table 4. Results of eight tests on four statistical models for selected observed global temperature data (except where noted). The statistical models tested are trends (power shown), LOWESS (0.5 total series smoothing), steps and steps and trends. Result include the adjusted $r^2$ value, the residual sum of squares (SS), cumulative residuals and squared cumulative residuals. F-tests for the whole series are shown, with $p<0.05$, $p<0.01$ noted if registered, otherwise $p>0.05$. F-test failure for 40-year period autocorrelation and heteroscedasticity is measured at $p<0.01$.**

| Model | $r^2$ | Residual SS | Cumulative residuals ($\sum R\ y^{-1}$) | Cumulative residuals$^2$ ($\sum R^2\ y^{-1}$) | F-test auto-correlation (F, $pH_0$) | F-test hetero-scedasticity (F, $pH_0$) | 40-y periods fail F-test auto-correlation | 40-y periods fail F-test hetero-scedasticity |
|---|---|---|---|---|---|---|---|---|
| **HadCRU 1861–2014** | | | | | | | | |
| Trend | 0.76 | 2.6 | 1.2 | 1.3 | 0.0 | 3.7 | 58% | 13% |
| LOWESS | 0.87 | 1.4 | 0.7 | 0.8 | 0.3 | 1.0 | 28% | 13% |
| Step | 0.87 | 1.4 | 0.5 | 0.8 | 0.7 | 3.2 | 0% | 0% |
| Step-trend | 0.87 | 1.3 | 0.1 | 0.8 | 0.2 | 5.8, 0.05 | 0% | 0% |
| **HadCRU 1965–2014** | | | | | | | | |
| Trend | 0.85 | 0.43 | 0.20 | 0.24 | 0.0 | 1.2 | 0% | 0% |
| Step | 0.86 | 0.40 | 0.20 | 0.21 | 0.4 | 0.7 | 0% | 0% |
| Step-trend | 0.89 | 0.31 | 0.06 | 0.18 | 0.0 | 1.4 | 0% | 0% |
| **NCDC 30°N–60°N 1880–2014** | | | | | | | | |
| Trend | 0.64 | 6.3 | 1.8 | 2.3 | 0.0 | 10.2, 0.01 | 51% | 9% |
| LOWESS | 0.79 | 3.7 | 0.9 | 1.6 | 0.2 | 3.0 | 19% | 0% |
| Step | 0.83 | 2.9 | 0.3 | 1.4 | 0.0 | 3.0 | 0% | 1% |
| Step-trend | 0.83 | 2.9 | 0.2 | 1.4 | 0.0 | 3.2, 0.05 | 1% | 0% |
| **HadCRU quarterly 1979–2014** | | | | | | | | |
| Trend | 0.69 | 1.7 | 2.0 | 3.5 | 0.0 | 1.1 | 20% | 3% |
| LOWESS | 0.72 | 1.6 | 0.5 | 3.3 | 0.2 | 2.8 | 3% | 5% |
| Step | 0.75 | 1.4 | 0.7 | 2.8 | 0.0 | 0.2 | 0% | 0% |
| Step-trend | 0.76 | 1.3 | 0.2 | 2.7 | 0.0 | 0.4 | 0% | 4% |
| **GISS quarterly 1979–2014** | | | | | | | | |
| Trend | 0.67 | 1.9 | 1.6 | 4.1 | 0.0 | 1.1 | 20% | 0% |
| LOWESS | 0.69 | 1.8 | 0.5 | 3.9 | 0.1 | 2.2 | 6% | 2% |
| Step | 0.71 | 1.6 | 0.9 | 3.4 | 0.0 | 0.0 | 4% | 0% |
| Step-trend | 0.72 | 1.6 | 0.3 | 3.3 | 0.0 | 0.6 | 0% | 0% |
| **RSS quarterly 1979–2014** | | | | | | | | |
| Trend | 0.40 | 3.4 | 4.4 | 6.9 | 0.0 | 1.2 | 11% | 6% |
| LOWESS | 0.46 | 3.1 | 1.1 | 6.4 | 0.3 | 2.3 | 4% | 14% |
| Step | 0.52 | 2.7 | 0.9 | 5.5 | 0.0 | 0.3 | 4% | 8% |
| Step-trend | 0.53 | 2.6 | 0.7 | 5.1 | 0.0 | 1.3 | 0% | 37% |
| **UAH quarterly 1979–2014** | | | | | | | | |
| Trend | 0.35 | 3.6 | 3.1 | 7.4 | 0.0 | 1.8 | 6% | 9% |
| LOWESS | 0.39 | 3.4 | 1.0 | 7.2 | 0.1 | 3.3, 0.05 | 4% | 20% |
| Step | 0.46 | 3.0 | 1.5 | 6.1 | 0.0 | 0.7 | 7% | 12% |
| Step-trend | 0.46 | 2.9 | 0.8 | 5.8 | 0.0 | 1.5 | 4% | 42% |



**Table 5. Results of eight tests on four statistical models for representing global mean warming from HadGEM-ES climate model run3 RCP2.6, 4.5, 6.0 and 8.5, showing the amount of warming for different measures. The statistical models tested are trends (power shown), LOWESS (0.5 total series smoothing), steps and steps and trends. Results include the adjusted $r^2$ value, the residual sum of squares (SS), cumulative residuals and squared cumulative residuals. F-tests for the whole series are shown, with $p<0.05$, $p<0.01$ noted if registered, otherwise $p>0.05$. F-test failure for 40-year period autocorrelation and heteroscedasticity is measured at $p<0.01$.**

| Pathway | Warming (°C) | Steps (°C) | Trends (°C) | Shifts (°C) |
|---|---|---|---|---|
| RCP2.6 | 1.93 | 2.29 | 0.65 | 1.24 |
| RCP4.5 | 2.93 | 3.30 | 1.76 | 1.07 |
| RCP6.0 | 3.65 | 3.86 | 2.09 | 1.75 |
| RCP8.5 | 5.34 | 5.35 | 4.24 | 1.41 |

| Model | $r^2$ | Residual SS | Cumulative residual ($\sum R/y$) | Cumulative residual$^2$ ($\sum R^2/y$) | F-test auto-correlation (F, pH$_0$) | F-test hetero-scedasticity (F, pH$_0$) | 40-y periods fail F-test auto-correlation | 40-y periods fail F-test hetero-scedasticity |
|---|---|---|---|---|---|---|---|---|
| **RCP2.6** | | | | | | | | |
| Trend ($x^4$) | 0.95 | 3.9 | 4.7 | 3.6 | 0.4 | 8.9, 0.01 | 75% | 18% |
| LOWESS | 0.96 | 4.7 | 7.7 | 2.8 | 6.9, 0.01 | 0.4 | 64% | 31% |
| Step | 0.98 | 1.1 | 0.04 | 1.2 | 0.1 | 10.7, 0.01 | 1% | 3% |
| Step-trend | 0.98 | 0.9 | 0.01 | 1.1 | 0.0 | 12.1, 0.01 | 0% | 4% |
| **RCP4.5** | | | | | | | | |
| Trend ($x^2$) | 0.95 | 8.8 | 16.6 | 4.8 | 0.8 | 2.1 | 77% | 73% |
| LOWESS | 0.99 | 3.9 | 13.3 | 2.5 | 2.3 | 4.1, 0.05 | 61% | 45% |
| Step | 0.98 | 2.4 | 0.5 | 1.4 | 0.0 | 5.7, 0.05 | 19% | 14% |
| Step-trend | 0.99 | 1.0 | 0.02 | 1.1 | 0.0 | 13.4, 0.01 | 0% | 2% |
| **RCP6.0** | | | | | | | | |
| Trend ($x^2$) | 0.97 | 4.5 | 51.1 | 5.2 | 3.7 | 23.5, 0.01 | 63% | 56% |
| LOWESS | 0.98 | 2.9 | 24.6 | 2.4 | 0.9 | 8.3, 0.01 | 52% | 31% |
| Step | 0.99 | 1.2 | 0.06 | 1.2 | 0.1 | 9.7, 0.01 | 2% | 5% |
| Step-trend | 0.99 | 0.6 | 0.01 | 1.1 | 0.0 | 17.9, 0.01 | 0% | 20% |
| **RCP8.5** | | | | | | | | |
| Trend ($x^3$) | 0.99 | 4.3 | 4.5 | 3.1 | 0.0 | 11.8, 0.01 | 62% | 39% |
| LOWESS | 0.992 | 3.1 | 66.6 | 2.8 | 2.0 | 4.5, 0.05 | 45% | 22% |
| Step | 0.99 | 8.1 | 2.0 | 1.7 | 0.2 | 106.7, 0.01 | 13% | 18% |
| Step-trend | 0.997 | 0.7 | 0.01 | 1.1 | 0.0 | 12.0, 0.01 | 0% | 3% |





**Table 6. Selected test results that distinguish between h$_{trend}$ and h$_{step}$. The null positions for each are generally not considered diametric. There is no generally accepted null with respect to h$_{trend}$ that references nonlinear change whereas for H$_{step}$ the null is no significant step-wise change points, or if there are they are completely random and do not contain and external forcing signal.**

| Test | Evidence | h$_{trend}$ | h$_{step}$ | Supporting literature |
|---|---|---|---|---|
| Global warming 1895–2014 | Trend/step ratio 0.32–0.38 (4 records), 0.58 (1 record) Trend shift ratio 0.44– 0.58 (4 records), 1.38 (1 record) | Gradual change, fluctuations but no steps | Substantial fraction of record contains steps | (Belolipetsky et al., 2015;Varotsos et al., 2014;Bartsev et al., 2016) |
| Regime changes | 1997 29 in 1997, 37 in 1996–98 of 45 global & regional records | Extreme El Niño 1997/98, stochastic event | Step-wise change points identified in temp and physically-related records | (Reid and Beaugrand, 2012;Chikamoto et al., 2012b;Chikamoto et al., 2012a;Overland et al., 2008;Menberg et al., 2014) |
|  | 1987/88 6 in 1987, 4 in 1988 of 44 regional records. Global ocean NH, NH mid-lat | El Niño, stochastic event | Step-wise change points identified in temp and physically-related records | (Reid and Beaugrand, 2012;Reid et al., 2015;North et al., 2013;Lo and Hsu, 2010;Menberg et al., 2014;Overland et al., 2008;Boucharel et al., 2009) |
|  | 1979 15 in 1979, 7 in 1980, 5 in 1977, 1 in 1976 of 44 global and regional records. Global, tropics, SH | N Pacific regime shift 1976–77, El Niño 1978/79 | Step-wise change points identified in temp and physically-related records | (Overland et al., 2008;Hare and Mantua, 2000;Menberg et al., 2014;Fischer et al., 2012;Meehl et al., 2009;Reid and Beaugrand, 2012) |
|  | 1969 4 in 1969, 8 in 1968–70, southern hemisphere | El Niño, stochastic event | Step-wise change points identified in temp and physically-related records | (Li et al., 2005;Hope et al., 2010;Jones, 2012) |
| Scalability of regional records | Records more steplike at zonal and regional scales and over the oceans. | Regional records would be trend-like if warming is diffuse and gradual | Regional records more steplike, large-scale records more trend-like. | None located |
| Attribution | Step-wise attribution for SE Australia (obs and models), Texas (obs), Central England (obs) | Gradual emergence of signal | Abrupt emergence of signal | (Jones, 2012) |
| Quarterly surface and satellite temperature 1979–2014 | Surface and satellite records share similar shifts but not trends | Significant trend for periods >30 years | Contemporaneous step-wise change points in independently measured records | None located |
| Simulated temperature patterns 1861–2005 | Clustering on runs test highly non-random (p~0.0' runs test) Significant correlations between timing of steps in models and obs CMIP3 0.32, CMIP5 0.34 1880–2005. | No matching patterns, randomicity | Matching step-wise changes between models and observations | None located |
| Simulated temperature quantities 1861–2005 | Trends/steps ratio 0.44±0.22 | Gradual change, deviations but no steps | Substantial fraction of record contains shifts | None located |
| Simulated temperature relationships with independent variable ECS RCP4.5 2006–2095 | Correlation and r² between ECS and total warming 0.81 & 0.65, steps 0.81 & 0.65, shifts 0.72 & 0.52 and internal trends 0.43 & 0.18 | Shifts random with respect to forcing | Shifts and steps more highly correlated with ECS and warming than trends | None located |
| Autocorrelation and heteroscedasticity observations 1880–2014 | Steps better performer than simple trends (Failure rate Trends 58±1% autoc, 10±4% heterosc.; Steps 2±4% autoc, 0% heterosc. 40y window) | Trends serially independent data, variations due to independent processes | Steps perform better than trends to explain autocorrelation and heteroscedasticity | None located |
| Autocorrelation and heteroscedasticity observations 1965–2014 | Trends and steps pass all tests for annual data, steps slightly better correlation than trends (0.86, 0.85 HadCRU) | Trends serially independent data, variations due to independent processes | Steps perform better than trends to explain autocorrelation and heteroscedasticity | None located |





| | | | | |
|---|---|---|---|---|
| Autocorrelation and heteroscedasticity quarterly observations surface temp 1979–2014 | Trends fail 40-y autocorr 20%, steps 0%, accumulated error trends/steps 2.9 Little difference heterosc. | Trends serially independent data, variations due to independent processes data | Steps perform better than trends to explain autocorrelation and heteroscedasticity | None located |
| Autocorrelation and heteroscedasticity quarterly observations satellite temp 1979–2014 | Accumulated error trends/steps 4.4, 0.9 and 3.1, 2.1 RSS & UAH Trends and steps little difference autocorr. and heterosc. (except steps 24% v 8% heterosc.) | Trends serially independent data, variations due to independent processes data | Steps perform better than trends to explain cumulative error, little difference autocorrelation and heteroscedasticity | None located |




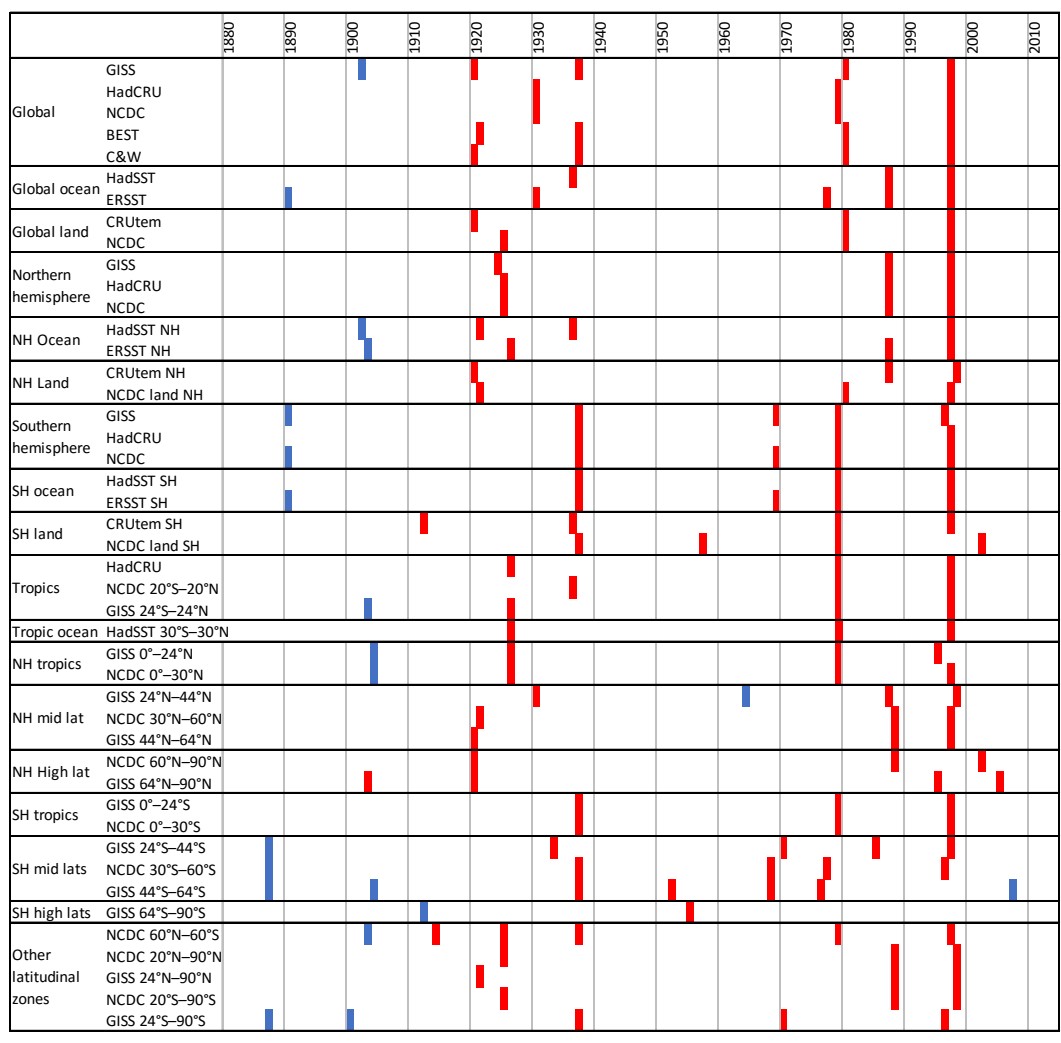

Figure 1: Dates of statistically significant step changes (*p*<0.01) 1880–2014, for a range of mean annual temperature records. Downward steps are blue and upward red. Records are sourced from Goddard Institute of Space Studies (GISS), the Hadley Centre and Climate Research Unit: HadCRU (land and ocean), HadSST (ocean), CRUtem (land), National Climatic Data Center: NCDC (land, land and ocean), ERSST (ocean), Berkeley Earth Surface Temperature (BEST) and Cowtan and Way (C&W). See Supplementary Information for details.




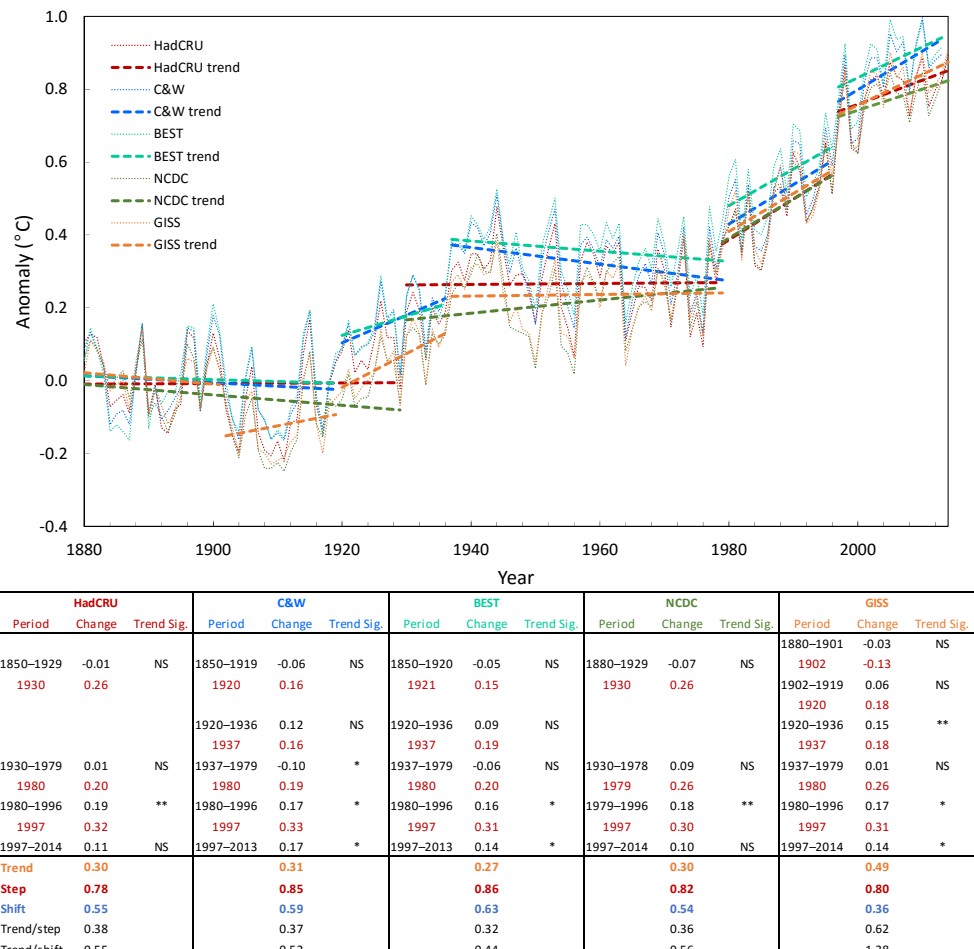

| | HadCRU | | | C&W | | | BEST | | | NCDC | | | GISS | | |
|---|---|---|---|---|---|---|---|---|---|---|---|---|---|---|---|
| | Period | Change | Trend Sig. | Period | Change | Trend Sig. | Period | Change | Trend Sig. | Period | Change | Trend Sig. | Period | Change | Trend Sig. |
| | | | | | | | | | | | | | 1880–1901 | -0.03 | NS |
| | 1850–1929 | -0.01 | NS | 1850–1919 | -0.06 | NS | 1850–1920 | -0.05 | NS | 1880–1929 | -0.07 | NS | 1902 | -0.13 | |
| | 1930 | 0.26 | | 1920 | 0.16 | | 1921 | 0.15 | | 1930 | 0.26 | | 1902–1919 | 0.06 | NS |
| | | | | | | | | | | | | | 1920 | 0.18 | |
| | | | | 1920–1936 | 0.12 | NS | 1920–1936 | 0.09 | NS | | | | 1920–1936 | 0.15 | ** |
| | | | | 1937 | 0.16 | | 1937 | 0.19 | | | | | 1937 | 0.18 | |
| | 1930–1979 | 0.01 | NS | 1937–1979 | -0.10 | * | 1937–1979 | -0.06 | NS | 1930–1978 | 0.09 | NS | 1937–1979 | 0.01 | NS |
| | 1980 | 0.20 | | 1980 | 0.19 | | 1980 | 0.20 | | 1979 | 0.26 | | 1980 | 0.26 | |
| | 1980–1996 | 0.19 | ** | 1980–1996 | 0.17 | * | 1980–1996 | 0.16 | * | 1979–1996 | 0.18 | ** | 1980–1996 | 0.17 | * |
| | 1997 | 0.32 | | 1997 | 0.33 | | 1997 | 0.31 | | 1997 | 0.30 | | 1997 | 0.31 | |
| | 1997–2014 | 0.11 | NS | 1997–2013 | 0.17 | * | 1997–2013 | 0.14 | * | 1997–2014 | 0.10 | NS | 1997–2014 | 0.14 | * |
| Trend | 0.30 | | | 0.31 | | | 0.27 | | | 0.30 | | | 0.49 | | |
| Step | 0.78 | | | 0.85 | | | 0.86 | | | 0.82 | | | 0.80 | | |
| Shift | 0.55 | | | 0.59 | | | 0.63 | | | 0.54 | | | 0.36 | | |
| Trend/step | 0.38 | | | 0.37 | | | 0.32 | | | 0.36 | | | 0.62 | | |
| Trend/shift | 0.55 | | | 0.52 | | | 0.44 | | | 0.56 | | | 1.38 | | |

Figure 2. Mean global anomalies of surface temperature with internal trends. The annual anomalies (dotted
lines) from five records (HadCRU, C&W, BEST, NCDC, GISS) are taken from a 1880–1899 baseline. Internal
trends (dashed lines) are separated by step changes detected by the bivariate test at the $p<0.01$ error level.
The size of each step (in red) and change in temperature of each internal trend (in black) is shown in the
figure table along with its significance, where NS is $p>0.05$, * is $p 0.01<0.05$, ** is $p<0.01$. Totals of trends,
steps, shifts (change from one trend to the next) and ratios are also shown.





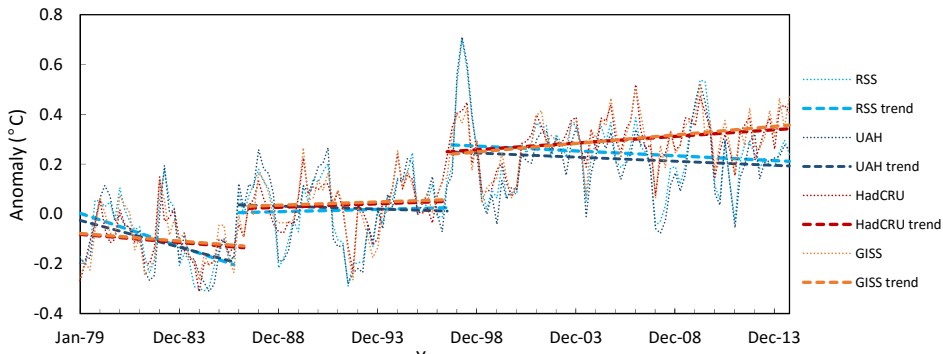

Figure 3. Quarterly mean satellite (RSS, UAH) and surface (HadCRU, GISS) temperature anomalies on a common baseline 1979–2014. Annual anomalies (dotted lines) and internal trends (dashed lines) are separated by step changes.



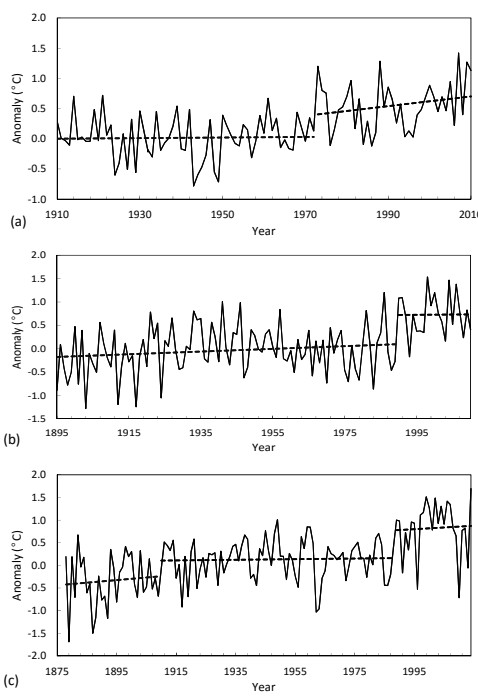

Figure 4. Anomalies of annual mean temperature attributed to nonlinear changes where the influences of interannual variability have been removed for (a) Central England, (b) Texas, and (c) south-eastern Australia. Internal trends (dashed lines) are separated by step changes ($p < 0.01$).



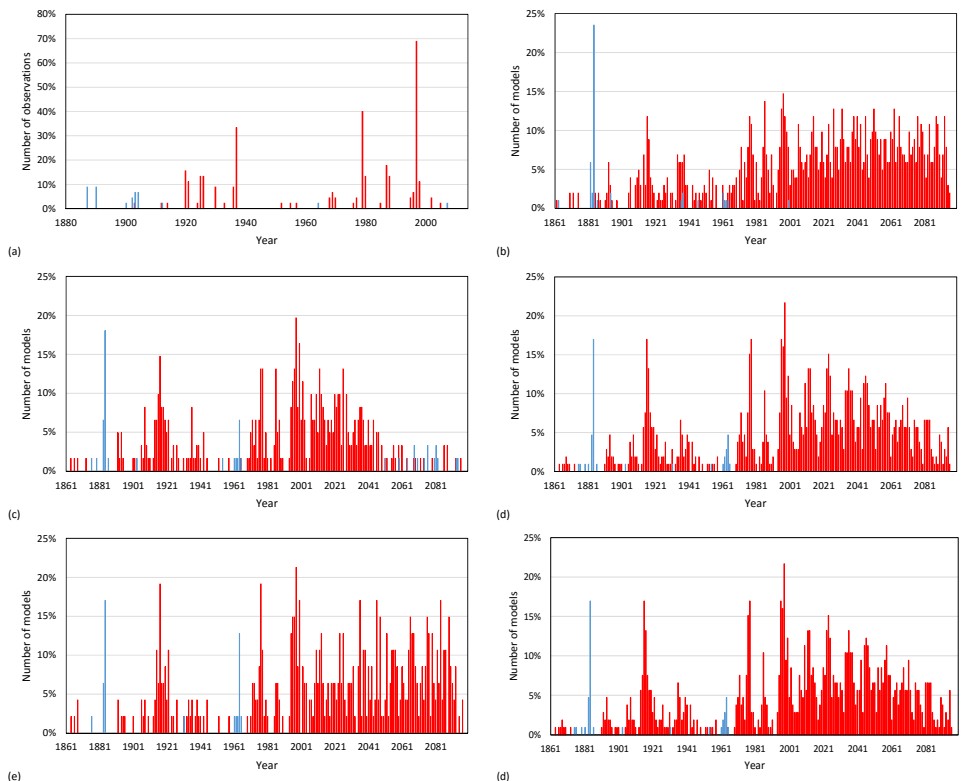

Figure 5. Step changes in observed and simulated surface air temperatures. Frequency in percent of statistically significant step changes from (a) global, hemispheric and zonal averages (45, 1880–2014); (b) global mean warming from 102 model simulations from the CMIP3 archive for SRESA1b and A2 emission scenarios; (c–f) global mean warming 1961–2100 from the CMIP5 archive for the (c) RCP2.6 pathway (61), (d) RCP4.5 pathway (107), (e) RCP6.0 pathway (47) and (f) RCP8.5 pathway (80).




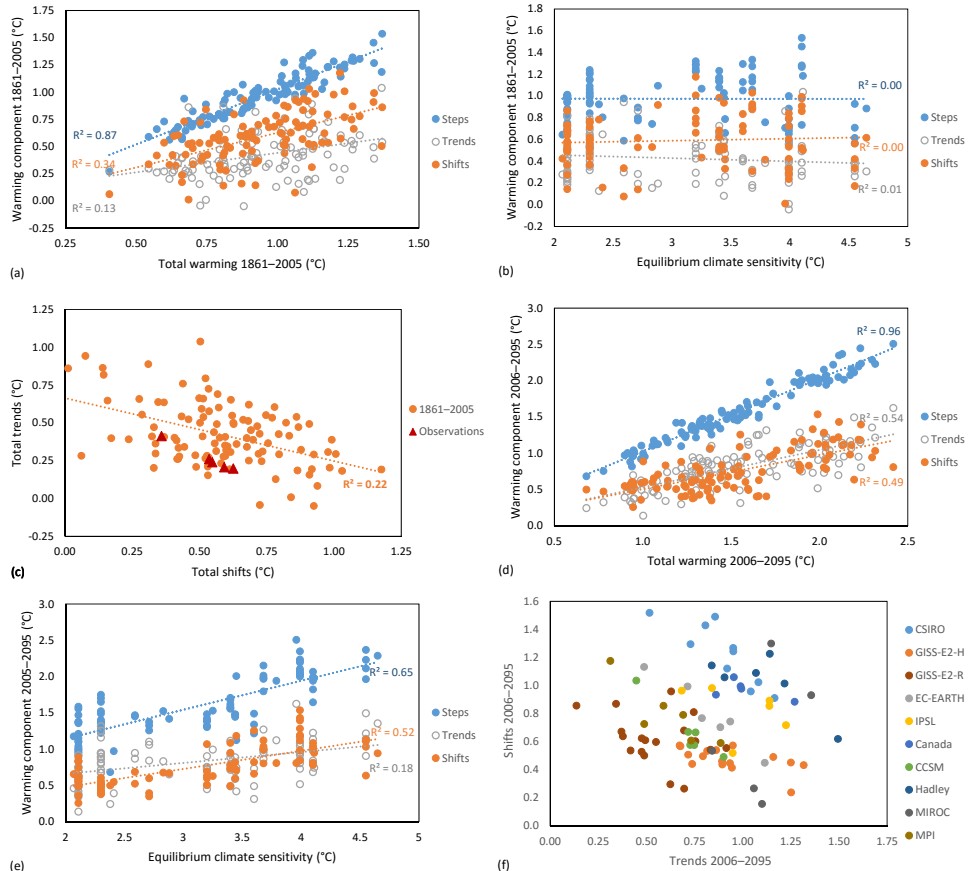

Figure 6. Multi-model ensemble (RCP4.5, 107 members) characteristics of hindcast (1861–2005) and projected (2006–2095) periods. (a) relationship between total warming and steps, trends and shifts (1861–2005); (b) relationship between ECS and steps, trends and shifts (1861–2005); (c) total shifts and total trends 1961–2005 with observed points from five warming records; (d) relationship between total warming and steps, trends and shifts (2006–2095); (e) relationship between ECS and steps, trends and shifts (2006–2095); (f) total shifts and total trends 2005–2095 from individual climate models.



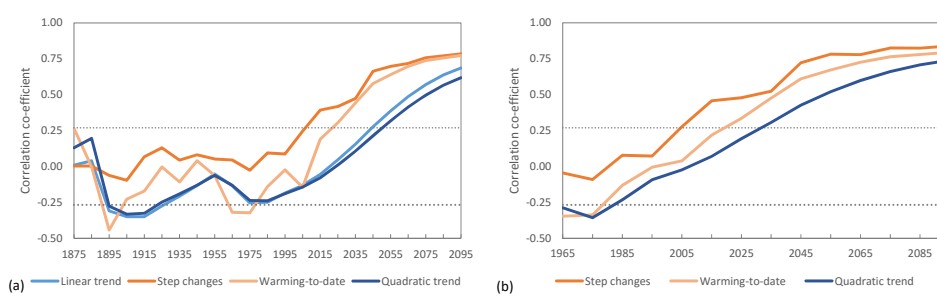

Figure 7. Correlation with successive calculations of change over time using a linear trend, step changes, warming-to-date and quadratic trend; (a) to the nominated date or decadal average subtracted from 1861–99 for warming-to-date and (b) as for (a) and decadal average subtracted from 1861–1959. Dotted lines mark p<0.01.



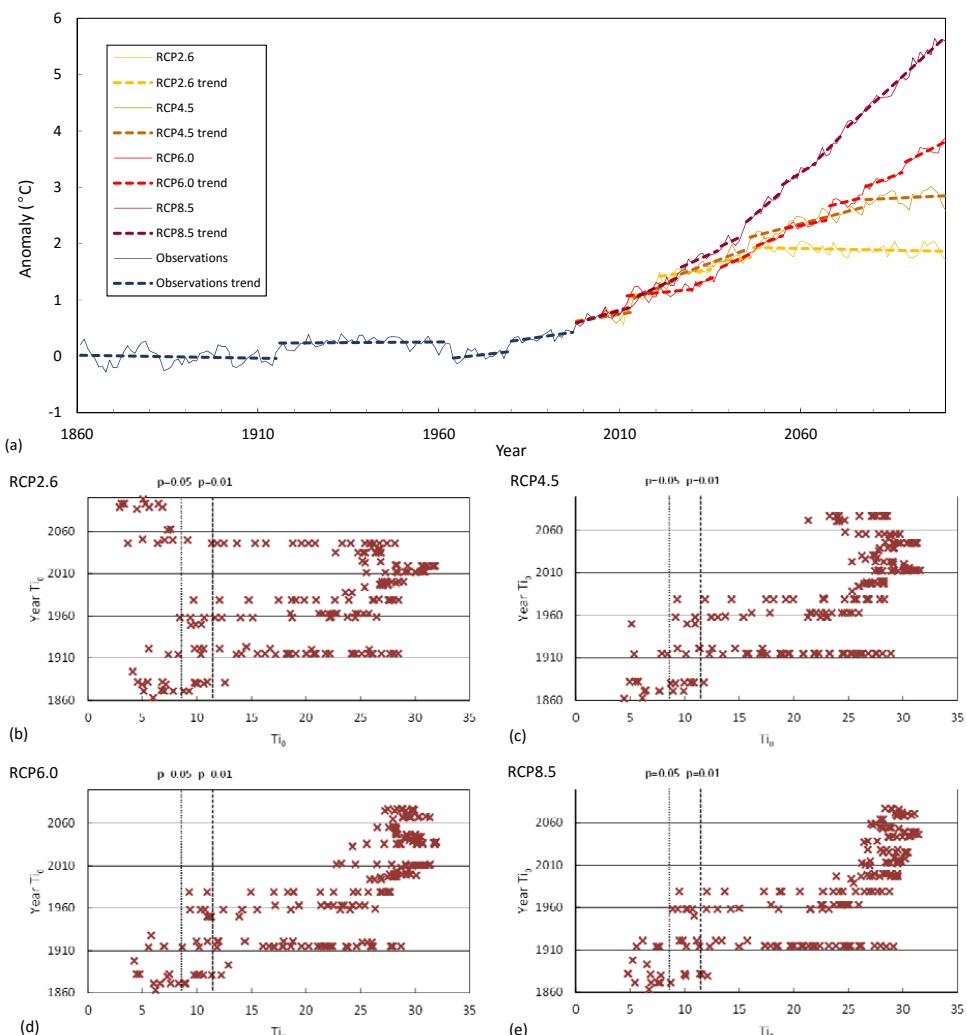

Figure 8. Global mean surface temperature as analysed by the multi-step bivariate test; (a) Step and trend breakdown of global means surface temperature in the RCP2.6, 4.5, 6.5 and 8.0 simulations from the HadGEM-ES model, run 3; (b–e) Ti0 results from a 40-year moving window for the RCP2.6, 4.5, 6.5 and 8.0 simulations, respectively.





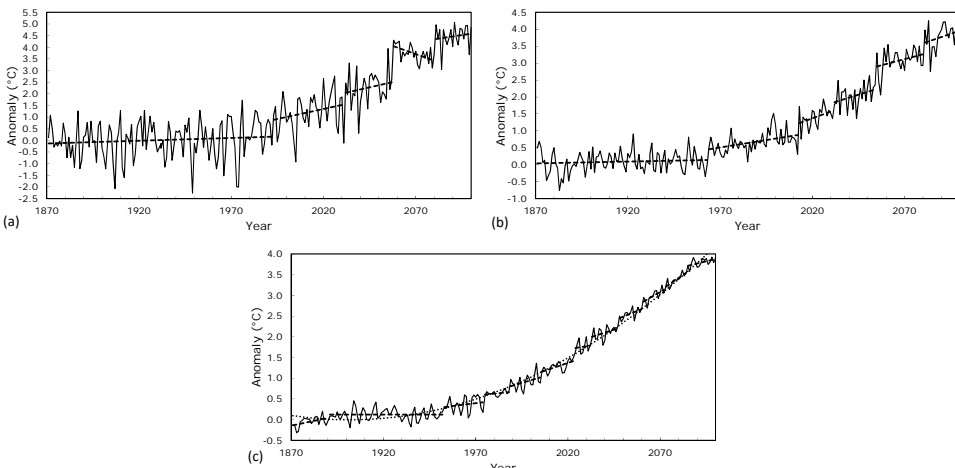

Figure 9. Anomalies of annual mean temperature showing internal trends separated by step changes from the CSIRO Mk3.5 A1B simulation; (a) maximum temperature southeastern Australia; (b) maximum temperature southeastern Australia; (c) global mean surface temperature Internal trends (dashed lines) are separated by step changes ($p<0.01$).





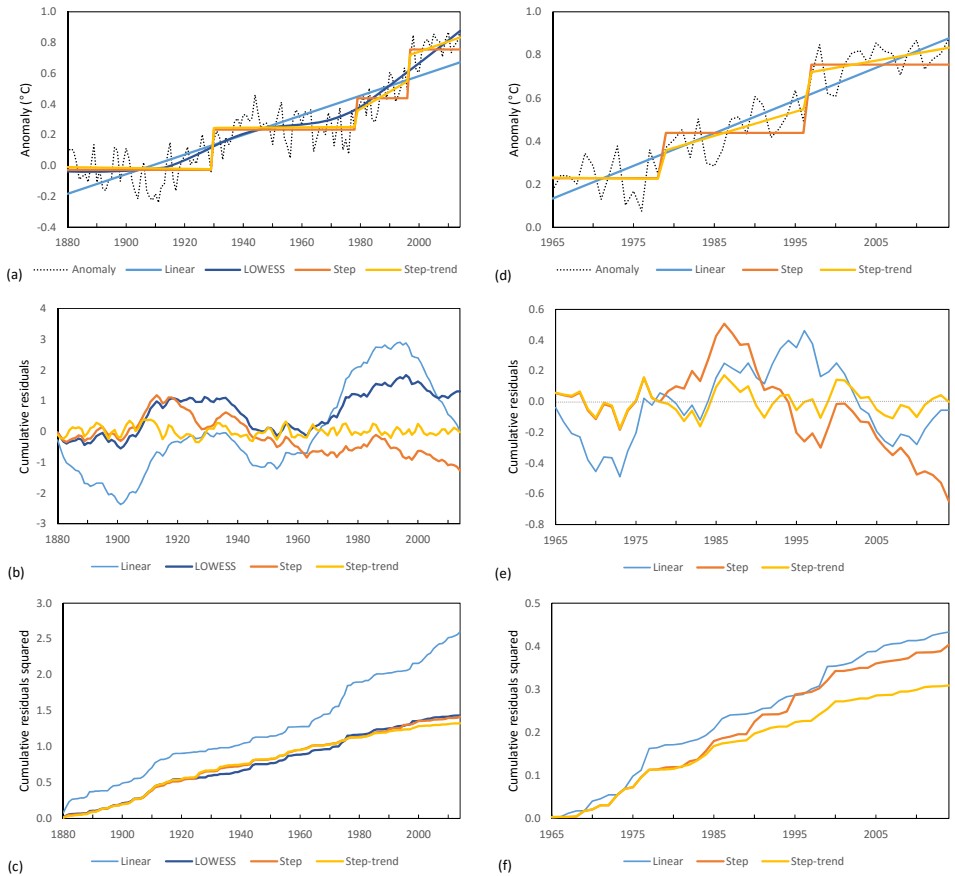

Figure 10. Testing three models to mean global anomalies of surface temperature from the HadCRU record, 1880–2014 (a–c) and 1965–2014 (d–f); (a) and (d) mean annual anomalies and linear, step change and shift and trend models; (b) and (e) cumulative residuals for each model, where success is measured as tracking close to zero; (c) and (f) cumulative sum of residuals squared, where upward steps show nonlinearity not explained by each model.