# Peer review of "Reconciling the signal and noise of atmospheric warming on decadal timescales"

_Earth System Dynamics, 2016_

## Referee Comment (RC1) · Anonymous Referee #1 · 14 Oct 2016

General comments: This paper presents interesting and novel results about the step-like development of many climate variables. There methods that are used are appropriate and it is good that a variety of tests and methods are applied. I still cannot recommend publication in its current form because the discussion of the hypotheses suffers from a lack of clarity (see specific comments). Thus I would recommend the authors to revise and resubmit the paper.

Specific comments: POINT 1: My main concern is that it is not clear enough what exactly the hypotheses are that are tested. Sometimes the authors say that what is tested is whether (i) internally and externally forced components of the climate system are independent or not (page 3, page 29); sometimes they say that what is tested is whether

(ii) the development of climate variables follows a trend or is step-like (abstract, Section 5). It is not the case that independence of internally and externally forced components of the climate system implies that there is a trend; this is possible, but there could also be independence and at the same time step-like behaviour. Also, it is not the case that dependence of internally and externally forced components implies that there is necessarily step-wise behavior. This could be the case, but there could also be dependence and a trend at the same time. As a result, it remains unclear what exactly is tested: (A) (Only) whether the internally and externally forced components are independent. (B) (Only) whether the climate variables follow a trend or not. (C) Whether the internally and externally forced components are independent AND whether there is a trend. (D) Whether the internally and externally forced components are dependent AND there is step-like behavior of the climate variables. Throughout the paper, the authors need to be clearer what exactly is tested.

POINT 2: Related to this, if what is tested is (C) and (D) (as is often suggested; cf. in particular the hypotheses on page 5), then it is important to see that (C) and (D) are not exhaustive (because there are also the possibilities that there is independence and a step-wise development; or that there is dependence and a trend). The authors want to test an exhaustive set of hypotheses, but (C) and (D) are not exhaustive.

POINT 3: Throughout the paper the assumption seems to be that "trend-like" and gradual as opposed to step-wise and non-gradual means that there is a linear relationship (e.g. on page 7). It is unclear why gradual implies that there is a linear relationship. There can be gradual behavior with various kinds of relationships (a quadratic relationship etc).

POINT 4: On page 7 the hypothesis states that there is a "(probably monotonic)" trend. The brackets are confusing. Is it now tested that the trend is monotonic or is it allowed that the trend is not monotonic?

POINT 5: on page 5 six tests are described. It should be clearly stated which tests test

which hypotheses (becomes clear later, but should be stated clearly early on).

POINT 6: The beginning of Section 2: here it is argued that the gradualist thesis is derived from induction. Yet, as the paper later argues, the data actually do not support the gradualist thesis and the gradualist thesis rather seems to be often adopted for no empirical reasons (convenience, simplicity). Hence it seems that the gradualist thesis is not justified by induction after all.

POINT 7: The beginning of Section 2: "The application of linear trend analysis to atmospheric warming is invariably justified as inference to the best explanation". I am puzzled by this sentence. Why is there suddenly a reference to the inference to the best explanation (previously the matter of concern was induction).

Technical corrections: Page 12, line 28 and 29: "otherwise are p>0.05" should be inside the brackets. Page 25, line 32: spaces are missing between the papers that are cited.

---

## Referee Comment (RC2) · Anonymous Referee #2 · 21 Oct 2016

1. Does the paper address relevant scientific questions within the scope of ESD? Yes.

2. Does the paper present novel concepts, ideas, tools, or data? Yes.

3. Are substantial conclusions reached? Yes

4. Are the scientific methods and assumptions valid and clearly outlined? Yes, but some clarification needed in places, due to the complexity of the subject covered and to make the text more readable for a general rather than specialist audience.

5. Are the results sufficient to support the interpretations and conclusions? Yes

6. Is the description of experiments and calculations sufficiently complete and precise

to allow their reproduction by fellow scientists (traceability of results)? Yes

7. Do the authors give proper credit to related work and clearly indicate their own new/original contribution? Yes

8. Does the title clearly reflect the contents of the paper? Yes, but see my comments on 'decadal'.

9. Does the abstract provide a concise and complete summary? Yes

10. Is the overall presentation well structured and clear? Yes, and could be largely maintained in the current format with changes to the last sections including the conclusions. I suggest separating the manuscript into two related papers one that largely addresses the philosophy and statistical approach behind the study and another that reports and discusses the results and their significance.

11. Is the language fluent and precise? Yes mostly.

12. Are mathematical formulae, symbols, abbreviations, and units correctly defined and used? Yes

13. Should any parts of the paper (text, formulae, figures, tables) be clarified, reduced, combined, or eliminated? Yes see comment 10.

14. Are the number and quality of references appropriate? Yes, a broad and comprehensive coverage of the literature.

15. Is the amount and quality of supplementary material appropriate? Yes. I am content with the supplementary material as presented.

General comments

This is an important paper and one of very few that challenges the conventional view that global warming is progressing in a gradual fashion. It proposes instead that warming has a step-like development with at times trends between each step. It backs up

the challenge with significant statistical analyses of many different global and regional temperature time series, plus the same for the time series temperature products of CMIP 3 and 5 models. A large volume of data has been processed in a comprehensive way using a standard and tested procedure.

I have minimal problems with the scientific results and feel that they are the main issue that should be focussed on. My concern in this context is that the current manuscript tries to do two things: 1) expound the philosophy behind the two different approaches to the way global warming has developed in the past and is likely to progress in the future plus how this can be tested statistically; and 2) present the results of the applied statistical analyses of measured regional and global temperature time series plus the same for modelled temperature products from CMIP 3 and 5 to the end of the 21st century. The audiences for these two different themes is likely to be different. Also, the debate at the beginning of the manuscript on the two hypotheses and on the need for severe testing before the reader gets to the meat of the ms might put off some researchers reading the paper. A secondary consequence is that the manuscript is long and needs breaking up with headings into more sub-sections than at present.

I acknowledge the importance of the philosophical debate and for section 2.2. I leave it up to the Editor to decide, but one way of proceeding would be to redraft the current ms into two associated papers, the first outlining the philosophy behind the two opposing views of global warming and its history and why severe testing is needed to address this issue and the second presenting the results with a brief account of the methods (that are backed up by the supplementary information) and the consequences of the analyses presented in the present paper. Some of the early text in the paper might be better placed in the discussion. When I started to read the paper I was keen to see the scientific results and spent much time trying to understand the complexities of the philosophy first. I think many researchers might be turned off from reading the paper by the long debate that is presented at the beginning and thus not discover the key findings that are included.
The current ms outlines a need for severe testing without summarising in more detail what 'severe testing' is, how this differs from other statistical approaches and what its advantages are, in language that is understandable to a general reader. The sentence that includes: "conditions for severe testing should be probative, rather than relying on a particular probability threshold" is relevant here. Severe testing proves or demonstrates that a particular hypothesis is accepted or rejected. The authors back up many of their statements throughout the ms and in the Supplementary Material by citing other papers that are in preparation that are led by one or other of the present authors. Only papers that have been published or are 'in press' should be cited. If more information is needed to back up a point it should be presented in the ms or if two, mss.

Most of my subsequent comments apply whether the Editor and authors decide to proceed with a revised version of the current ms or a revision that splits the contents into two related papers published at the same time, one after the other.

Specific comments

Abstract: I like the Abstract and the way it is organised and numbered against each of the six tests. To make it clearer refer to Test 1 rather than (1) and the same for the remainder unless this breaks the word count. If brevity is needed put the brackets and numbers in bold. It would be helpful if the results section could be organised in the same way. Introduction.

Slightly expand in the statistical analysis ms if the 2 paper route is chosen by incorporating some of the material from Sections 2.1, 2.2 and 2.3, but most of this text would I expect be transferred to the 'Philosophy ms'. I think the latter would be an important paper in its own right as it could include some of the history behind the two views of global warming as well as emphasising that the risks from global warming are possibly greater if the second hypothesis is what is really happening. Include the text on page 5 lines 16 to 19 and perhaps expand on this reason for rejection by the scientific community of a nonlinear pattern to global warming here or in the discussion of the statistical

analysis paper. This discussion should be included in both ms.

Two hypotheses H1 and H2 and even –H1 and –H2 are used throughout the paper, but are not clearly defined early on. Do the authors consider that the two hypotheses outlined at the top of page 3 are equivalent to H1 and H2. If so this should be clearly stated in the Introduction.

2.3 Line 23. I do not understand what you mean by "is an inverse, ill-posed mathematical problem". I realise that the phrase has been quoted from Serban and Jacobsen. Please clarify.for the non-mathematical general reader. Further down I do not see why "external forcing" need always be linear; please clarify. The development of this section comes back to the use of the word decadal in the title. My understanding of decadal is based on data averaged by decade. Most of the data used in the ms is based on annual averages and the step changes take place within a year so I don't think the word decadal should be used; long-term would be better although I realise that decadal is used widely in time series literature. Serban above distinguishes between annual, decadal and centennial averages. Later in the paper I believe the CMIP time series are based on 5 year averages.

Page 6. I can see why radiative forcing and its interactions is additive, but again, I do not see why entrainment of heat energy into the various heat reservoirs of the Earth and especially the hydrothermal system need always be nonlinear. This is likely to vary with the area defined and the time taken, for example heat release from the ocean to the atmosphere. Lines 2 to 6 outline a number of alternative approaches to determine 'shifts', 'change points', 'step changes', but there is no discussion of the advantages/disadvantages of these different approaches and why they were not used in this study. See also: Drijfhout et al. (2015) and Reid et al. (2016) Reid PC, Hari RE, Beaugrand G et al. (2016) Global impacts of the 1980s regime shift. Global Change Biology, 22, 682-703.

Page 7 line 21. First mention of ECS without identifying what the acronym stands for.

There is no summary of the source of the data or how it is derived/measured other than from Table S3 in the Supplementary Information (ECS is taken from Sherwood et al. (2014) unless otherwise noted). Nor is there any explanation of why ECS is compared to the CMIP time series. Page 8 MYBT. Many acronyms are used throughout the paper. It would be helpful to provide these as a list with the full name at the beginning of the paper as searching for them in the text is onerous.

Page 9 lines22 to 28. A diagrammatic representation of the different terms used for the analyses is needed. A descriptive expansion of what is meant by each of the terms would be helpful. The word 'shift' has been used in a different way in previous papers and a different word would be more appropriate here for this characteristic. Is the text in brackets at the end of the last bullet correct? Page 18 line 4. Lack of predictability. How can the author's be so definite that this might be due to aerosols?

Page 23 Section 5.1. This section would be better drafted as the conclusions of the paper rather than as a summary of severe testing. Combine the first two paragraphs with the current conclusions, which are similar in part if worded slightly differently. Ensure that all the observations highlighted in yellow above are covered in the bulleted summarised conclusions.

I don't think it is necessary to repeat the ratio numbers or r2 values from the results in this section. It should instead just give a verbal summary of the main conclusions reached.

Page 25 line 2. Again I do not like the use of the word decadal here. Table 6 does not show that hstep is better at a decadal scale the steps are occurring within a year, but may continue at a new level or develop a trend afterwards for more than a decade. Decadal data is usually averaged and smoothes out any signal from a step change. e.g. Figure SPM 1 in the 2013 IPCC Summary for Policy Makers.. Longer term (> 50 year) warming is also smoothed out and here there may also be issues with the quality of the data, so while it may seem to be proportional to forcing, steps may also have

occurred in this data.

Page 27 line1-2. The hiatus is now thought to be due to an increased storage of heat deeper in the ocean and is not a continuing event considering the warming of the last few years. See Reid 2016 and references included.

Reid PC (2016) Ocean warming: setting the scene. In: Explaining ocean warming:causes, scale, effects and consequences. (eds Laffoley D, Baxter JM) pp Page. Gland, Switzerland, IUCN.

Technical corrections

Page 2 Line 2. Abstract. Change to: 'variations that extend over decadal scales of time'. See later comment on use of decadal.

Page 2 Line 13. 'the correlation'

2.2 Line 21. First mention of H1 and H2 together. They were used separately in the introduction.

Line 7. Start 'For H1. . ..'.on a different line to make it comparable to H2 below. There are no citations to back up the statements made in the H1 section.

Line 18. Decadal again. The transfer from one regime to another is evident at an annual level and not decadal.

Line 17. Should not be numbered 3 or indented.

Lines 25 to 30. This text should be part of a discussion and not here.

Line 33. At the end it is important to note that regime change is precipitated, but to a new level or a trend.

Line 31 to Page 7 line 4 repeated below.

Page 7 line 12. –H1 and –H2 mentioned for the first time. Define what they mean in general language.

Page 7 line 13 onwards. Six tests are identified. It is not clear if the first two are the same as the two tests mentioned on page 9 lines 31-32. Please make this section clearer.

Page 8 line 19/20 "MYBT is considered reliable". Is this remark necessary without some backup? You could refer to page 13 line 2 in the Supplementary Information.

Page 9 line 2. Put in a heading Data and distinguish between the observed and modelled time series by putting them in different paragraphs. It would have been helpful to leave a line space between each paragraph.

Page 9 line 13 Again provide a new sub-heading

Page 9 line 32. Again, does the reference to Test 1 and 2 refer to the first two tests of the six mentioned earlier?

Page 11 below line 13 put in a heading: Shift/Trend Ratios

Page 12 lines 1 and 2. An important result. Missing full stop after warming.

"The more steplike character of both the oceans and the mid-latitudes is consistent with those areas being the loci of change in terms of decadal regimes and nonlinear equator-to-pole transport. This is inconsistent with the hypothesis of gradual warming".

Page 12 line 7. Suggest change to "Annual and seasonal anomalies were investigated". And edit next sentence so not starting with Annual.

Page 12 line 12. Why are quarterly anomalies only examined for the satellite temperatures? This needs explaining.

Page 12 line 23. Confirmation of the results from Reid et al. 2016 that the 1987 regime shift is evident at a global scale and yet on the next line it is said to be only evident at a regional scale.

Page 12 line 25 and 26. An important result.

"When all four records are plotted on a common baseline of 1979–1998, the surface and satellite temperatures display similar shifts but different internal trends (Fig. 3)".

Page 12 lines 333-34. An important observation.

"Unless substantially contaminated by artefacts, these changes do not reflect gradual warming in the atmosphere, but instead may reflect regime-like change controlled from the surface". As is the subsequent comment on heat release from the ocean during El Niño. See commentary in Reid 2016 on this issue.

Page 13 line 5. Which timescale?

Page 13 line 14. Insert 'out' after carried.

Page 14 line 18 An important observation.

"indicates that the onset of the warming signal in these broader regions is abrupt (Jones, 2012)".

Page 14 line 21 Use year (2016) of hard copy publication for Reid et al. (2015).

Page 14 line 23. An important observation. "that shifts may be more distinct at regional scales, integrating into a more trend-like global average".

Page 14 lines 23-25. Sea level steps are said to be ubiquitous in local tide gauge time series, Table 3 in Jones et al. 2013, but were not checked or analysed by Jones et al.

Page 14 line

Page 16 lines 20-21. An important observation.

"A simple linear trend measured over the entire period has the same correlation with steps (0.93, p<0.01) but averages 0.76 °C, so underestimates total warming by 0.18 °C".

Page 17 line 4. Why are 5 year averages used here, the first mention that the data has been treated in this way.

Page 18 lines 7-8 and 12-13. Important observations.

"that the magnitude of 20th century warming in the models has little predictive skill and is not a reliable guide to potential future risk".

"This strongly indicates that 20th century warming may not be a good guide to future warming, if observations are being affected in a similar way".

Page 18 lines19-21. An important observation.

"step changes clearly carry the greatest signal with respect to ECS over time".

Page 18 line 33. Lable bullet A1 and at top of next page the bullet A2.

Page 19 lines8-9. "peaking in the 2080s….." does not fit with the figure 5f. What should be Fig. 5f is a duplicate of Fig. 5d.

Page 20 line 16. Local changes. An important observation.

"significant increases in impact risks". Is this statistically demonstrated. If not don't use the word 'statistically.

Page 20 line 23. Is the first part of Section 5 essential to the paper? Would it be better to label it 'Sensitivity testing'.

Page 20 line 25. Insert 'and' after 'warming'?

Page 21. Line 30 change to: 'performs the best'

Page21 line 31-33. Duplication 'into the' and 'test'. Change to: 'at a global scale when each model is'

Page 22 line 30. '21st'

Page 24 line 9. Spelling 'are' not 'area'

Page 24 line 18-19. Edit sentence beginning: 'Warming is not….'

Page 24 lines24-25. Make sure this statement is backed up by appropriate citations in the results section

Page 24 lines 9-10 and 30-32 repetition. Is this necessary.

Page 25 lines 5 to 10. Delete 'In summary' and draft as the final paragraph of the conclusions.

Page 25 line 7 in situ in italics. And, 'or as a gradual'.

Page 25 lines 9-10. Edit to: 'where climate change and variability interact rather than varying independently.'

Page 25 line 13 Discussion. Include a discussion of how the results of Drijfhout et al. (2015) compare to those presented in this paper.

Drijfhout S, Bathiany S, Beaulieu C et al. (2015) Catalogue of abrupt shifts in Intergovernmental Panel on Climate Change climate models. Proceedings of the National Academy of Sciences, 112, E5777-E5786.

Page 25 line 15 change 'earlier' to 'before'?

Page 25 line 17. 'gradualism' and 'as a key tool to understand how'?

Page 25 line 23 'to explain climate'

Page 25 line 24. Change' 'covering methods' to 'applying procedures'?

Page 25 line 25. Delete 'and its application to understanding climate processes'.

Page 26 line 5. 'analytical'.

Page 26 line 12 a priori Italics

Page 26 lines13-14. Important observation that needs to be included in the conclusions. 'the processes involved are timescale invariant indicate that the meaning of seamless has not really been thought through'.

Page 26 line16. 'would likely be'. And change 'considerable' to 'sizeable' as repeated on the next line.

Page 26 line17. Change 'under' to 'that have'.

Page 26 line 19. First sentence of bullet. Something is missing.

Page 26 line 20-21. 'physics, understood as being primarily linear and hydrometeorology with its substantial nonlinear behaviour; both remain largely unreconciled.

Page 27 line 9. 'stated'

Page 27 line 20. Somewhere in the text above it is worth stating that both Cahill and Foster consider that the hiatus was a non-event.

Page 27 line 26-7. Good point!

Page 28 lines 7-9. Delete: 'As we discussed in a related paper where H2 is examined in greater detail' and the reference to Jones and Ricketts, 2016 as this paper is only 'in preparation'. Edit the sentence without the above text except for H2.

Page 28 lines 11-18. An important paragraph. You might also cite Roemmich's recent papers and Reid 2016 to back up this paragraph.

Page 28 lines 19- 22 repeated below on lines 23-26.

Page 28 line 31, The word 'extraordinary' is perhaps a bit too strong.

Page 28 line 32. 'to either side'

Page 29 lines 1-2. Leave out the sentence: 'Elsewhere . . ...'., but, raise the possibility that we are undergoing another shift at present.

Page 29 lines 3-5. Poor ending to this section. Edit and improve as a statement to round off the discussion.

Page 29 line 13. See earlier comment on >50 year climate change.
Page 29 line 17. Delete sentence beginning: 'We discuss this . . ...'

Page 46. Figure 4. I don't know what the journal policy is for sub-figures, but I prefer the lettering, a, b, c to be in the top left hand corner, inside the enclosing border of each sub-plot. It would also help if the respective sub-plots were labelled: England, Texas and Australia within the enclosing border. Insert at the beginning of the legend: 'Regional temperature change'.

Page 47. Same comment as for Figure 4. Label a, b, c, d, e, f in the top left corner of the subplots and in the top right in order: Add in sequence in the top right corner of a: 'observed', of b: simulated, of c: '2.6', of d: '4.5', of e: '6.0' and of f: '8.5'. In the legend add downward blue and upward red as for Figure 1.

―――――――――――――――――

---

## Editor Comment (EC1) · M. Crucifix (Editor) · 11 Nov 2016

Thank for you having submitted your article to Earth System Dynamics. We now have received two referees reports, which are mostly positive and encourage revision before publication.

The manuscript is slightly unusual in having a fairly developed introduction. It touches on philosophical and technical aspects of inference, hypothesis testing and selection, with even a long citation of climate scientist J. Slingo (taken outside the formal scientific literature). The submitted article also briefly reviews physical reasons as why the warming occurs in steps or gradually, by reference to research on entropy production and / or dynamical systems concepts. Epistemology of statistical inference and com-

plex system dynamics are vast domains indeed. The introduction of Jones and Ricket would easily provide ample opportunity for further discussion and debate (among others there are different competing paradigms of inference). The physical study of the dynamics of complex systems is no less potentially overwhelming. By attempting to embrace such general questions as a prelude to a specific investigation, the authors end up being quite specific in their choices of references (some might say : 'patchy'). I would encourage the authors to better focus the article on the statistical application they are interested in, with only a brief introduction which puts the approach in context. When needed, references to textbooks and reviews are welcome. Extra care also needs to be given to the choice and definitions of words: among others the word 'linear' will be understood differently by a statistician and by a physicist. Remember that main role of the introduction in a research paper is to explain why the study is justified and how it improves on previous attempts and / or addresses unsolved problems.

The discussion is a bit long as well. It brings upfront a list of meta-scientific arguments as to why 'shifts have not been recognized earlier', which include sociological and even psychological elements. Even if the arguments sound plausible, their epistemological status in speculative. Intertwining meta-scientific arguments with a technical investigation about the reality of climate shifts is hazardous. There is certainly a way to express the conclusions of the paper and convey the message about the relevance of climate shifts more soberly and more efficiently. After all, the essence of the argument reads as follows: Dynamical systems display shifts and bifurcations; authors therefore allow themselves to consider as plausible a hypothesis of thermal jumps; their statistical investigation shows that such shifts explain the data better than would a 'gradualist' view.

Referees #2 suggests to split the manuscript. At this stage, I would only consider one manuscript which focuses on the statistical analysis of the observations. Both the introduction and discussion need to be re-written. Once this is done, please proceed as usual and address referees' comments one by one. The broader and perhaps loser

discussion of why gradual trends are usually privileged could be spared for another article where the epistemological status of that discussion can be better framed.

---

## Author Comment (AC1) · 1 Dec 2016

Authors' response

We would like to thank both reviewers for their positive and thoughtful comments, most of which we will accommodate in some way. It is a large paper, so we are grateful for the effort and attention the reviewers have given.

Since this paper was submitted, the introductory sections have expanded into a more philosophically-based paper, which is unfortunately only at the submission stage (although it exists as a working paper). This will be summarised in the revision in a structure that better relates theory to plausible physical mechanisms and statistical tests at

the beginning of the paper to make it quite clear what we are testing and why. This will also help to align the different uses of linear and linearity in the manuscript. The more philosophical passages would be removed from this paper.

If the paper is to be kept as single paper as recommended by the editor, this introduction is very important because we are arguing that a step-based statistical model provides a better explanation of the warming process on decadal timescales than a trend-based model and that physical mechanisms for step-like warming are both theoretically and physically plausible. This is not an argument based on statistical induction alone. We will edit section 2 to focus principally on this point, leaving the broader philosophical questions for other papers.

Response to Reviewer 1

*POINT 1: My main concern is that it is not clear enough what exactly the hypotheses are that are tested. Sometimes the authors say that what is tested is whether (i) internally and externally forced components of the climate system are independent or not (page 3, page 29); sometimes they say that what is tested is whether (ii) the development of climate variables follows a trend or is step-like (abstract, Section 5). It is not the case that independence of internally and externally forced components of the climate system implies that there is a trend; this is possible, but there could also be independence and at the same time step-like behaviour. Also, it is not the case that dependence of internally and externally forced components implies that there is necessarily step-wise behavior. This could be the case, but there could also be dependence and a trend at the same time. As a result, it remains unclear what exactly is tested: (A) (Only) whether the internally and externally forced components are independent. (B) (Only) whether the climate variables follow a trend or not. (C) Whether the internally and externally forced components are independent AND whether there is a trend. (D) Whether the internally and externally forced components are dependent AND there is step-like behavior of the climate variables. Throughout the paper, the authors need to be clearer what exactly is tested. "but there could also be independence and at the*

*same time step-like behaviour"*

The exact hypotheses will be clarified, though the testing of step against trend will be unchanged.

For H1, if the response to external forcing is considered to be independent of variability over shorter timescales (<50 years), the trend model will hold, despite often being obscured by variability. Such variability is generally represented as stochastic behaviour in annual to decadal phenomena, where teleconnections, lagged effects and regime changes all potentially interact. Alternatively, instead of a gradual line or curve, a segmented trend is sometimes proposed.

The potential behaviour of warming under H2 has many possible permutations because the signal may project onto the regime-like structures of decadal climate variability, or may dynamically modify those structures. Here, we deal with one such type of response, in the form of step changes. Step changes have been detected in warming and related climatic variables by several different methods. The purpose of this paper is to detect step changes in a range of temperature records and test these against trends to determine which carries the greater part of the warming signal. The results are used to determine whether H1 or H2 is the more viable hypothesis and, if the signal proves to be nonlinear, to explore the nature of the interaction between external forcing and internal variability.

It may be possible for internally-generated step-like behaviour on a random basis, but not sustained step-like behaviour in one direction (the exception could be a singularity, such as ice-sheet collapse). We will point this out. The H1 and H2 aspects can be redrafted to address the reviewer's points and to focus on the type of testing we undertake and why.

*"it is not the case that dependence of internally and externally forced components implies that there is necessarily step-wise behaviour"*

True, and we will make it clear that we are not claiming this and why we are using steps, given the amount of prior knowledge we have about their occurrence.

*POINT 2: Related to this, if what is tested is (C) and (D) (as is often suggested; cf. in particular the hypotheses on page 5), then it is important to see that (C) and (D) are not exhaustive (because there are also the possibilities that there is independence and a step-wise development; or that there is dependence and a trend). The authors want to test an exhaustive set of hypotheses, but (C) and (D) are not exhaustive.*

We are not testing an exhaustive set of either physical or statistical hypotheses and will make this clear; however, understanding the theoretical background and potential mechanisms that informs the testing environment is important. The existing research identifying step changes as a mechanism for warming change on decadal timescales needs to be tested in an environment that is dominated by an existing paradigm that says the opposite. Because paradigms are partly a sociological construct (though are informed by theory) only a strong case that can cover the theory along with addressing cognitive values has a chance of being accepted. The framework we are using is a theoretical-mechanistic – statistical induction framework where theory is used to distinguish plausible mechanisms that allow a clear choice to be made between alternative hypotheses. This will be described more clearly. Statistical testing provides the means to do this, so statistical hypotheses need to be developed that match the alternative mechanisms. So what we have to do is to make this clear in the light of the A, B, C and D possibilities that Reviewer 1 puts forward.

*POINT 3: Throughout the paper the assumption seems to be that "trend-like" and gradual as opposed to step-wise and non-gradual means that there is a linear relationship (e.g. on page 7). It is unclear why gradual implies that there is a linear relationship. There can be gradual behavior with various kinds of relationships (a quadratic relationship etc).*

Here linear is referring to whether the temperature signal follows a secular trend therefore can be defined by a line and refers to the linear transfer relationship between forcing and temperature. Nonlinear response implies the transfer is nonlinear and is modified by nonlinear physical processes. This will be more tightly defined initially and the different between physically linear response and linear trends, for instance, made clear through-out. The usage in the paper is physical and when statistically linear is used is will be specific, e.g., linear trends.

*POINT 4: On page 7 the hypothesis states that there is a "(probably monotonic)" trend. The brackets are confusing. Is it now tested that the trend is monotonic or is it allowed that the trend is not monotonic?*

Related to the previous point – we will clarify earlier that the underlying signal of atmospheric warming is assumed to be monotonic (while we accept that total global warming is monotonic). This is later clarified on page 6. This section will be rewritten as per the overall response to both reviewers 1 and 2.

*POINT 5: on page 5 six tests are described. It should be clearly stated which tests test which hypotheses (becomes clear later, but should be stated clearly early on).*

It will be simpler if just H1 and H2 are referred to previously – all six tests are designed to assess which has the better explanatory power and more detail will be added to make this clear up front.

*POINT 6: The beginning of Section 2: here it is argued that the gradualist thesis is derived from induction. Yet, as the paper later argues, the data actually do not support the gradualist thesis and the gradualist thesis rather seems to be often adopted for no empirical reasons (convenience, simplicity). Hence it seems that the gradualist thesis is not justified by induction after all. It is probably more accurate to say the gradualist thesis is sustained by statistical inference (as a form of induction).*

We have done some more work around this to inform other papers on the same theme. In the 1970s, various theoretical arguments were put to suggest if a system received

a small internal forcing the response would be proportional (e.g., Leith 1973, 1975). Much of the subsequent work was based on vectorizing the forcing-response relationship, which will produce a linear outcome. This was based on statistical physics.

There are two types of induction therefore: one is largely analogical, based on statistical physics, proposed by scholars like Leith. This is now generally accepted as an approximation that only holds if the residuals are Gaussian (e.g., Palmer, 1999) and they are not in every case. There have been two styles of statistical induction. The first in the 19th century based on, which is truth-centred. The other is modern statistical induction, which is often reduced to a mechanised form based on establishing a sufficiently small probability for the null hypothesis. For some, the trend has again become truth-centred, but this can is only be the case if warming is gradual. Otherwise, it is a statistically-derived approximation of the relationship between forcing and response. This issue is at the core of the paper.

Other cognitive values such as convenience and simplicity are applied, because they are mentioned in the literature frequently, often as an escape clause to bypass some acknowledged but unknown complexity. They are also used as a defence against over-parameterisation and overfitting. Their use is mainly sociological rather than scientific. A full discussion of all of these is not appropriate for this paper, so we will pare it back to the basic issues – we acknowledged these other issues because readers will realise that the story is incomplete but a more comprehensive exploration is not feasible.

This issue has been complicated by the climate wars, where trend analysis has been used as a defence, so if not strictly truth-centred, it had to be defended as scientifically correct. This has hampered the consideration of alternatives. This comment is largely background because most of the philosophical content would be removed in a revision leaving the severe testing component.

*POINT 7: The beginning of Section 2: "The application of linear trend analysis to atmospheric warming is invariably justified as inference to the best explanation". I am*

*puzzled by this sentence. Why is there suddenly a reference to the inference to the best explanation (previously the matter of concern was induction).*

This point is about simplicity as a cognitive value – invariably is also too strong a word. Some of the confusion can arise because there are multiple frames around this issue, as there will be around any long-held view that has evolved over time. The literature may describe the theoretical developments around an issue but it will also reveal sociological aspects of how and why a community of practice believe certain things. These can conceivably be quite different – because the latter will contain assumptions of convenience and workarounds. Most of the discussion about reasoning will be removed except for that which relates directly to the testing approach.

All technical corrections can be addressed.

Response to Reviewer 2

General comments

The single paper option will be chosen as recommended by the editor and the introduction and Section 2 rewritten to focus on the statistical testing. However, it is important to preface the statistical tests with the probative conditions as to why those tests were chosen. It is also possible to simplify the relationship between theoretical propositions of independence and interacting externally-forced and internally-generated change (as per the response to reviewer 1), the mechanisms reflecting those theoretical propositions and why they were selected and the statistical hypotheses. The most important aspect of this is to test between sustained incremental change as would be manifest in gradual warming and episodic change. For that reason, retaining Page 5 lines 6–19 is important because it explains what the statistical hypotheses represent.

Another manuscript that expands on a theoretical-mechanistic and statistical-inductive framework the informs the testing environment and expends greatly on Section 2 has been written, so this section will be tightened to serve as a platform for interpreting the

test results.

Specific comments

Abstract – test numbers can be bracketed

Introduction – see above

2.3 line 23 – ill-posed inverse problem will be given a plain-language description. The upshot of that is that a great deal of statistical testing treats mean global warming as a one-dimensional problem, which it is not. This relates to part of test 1: regional stratification – we can nominate this better in the test description.

External forcing – linear – the explanation referred to in the response to Reviewer 1's comments will clarify the relationship between physical and statistical linearity.

Decadal scale – the most widespread usage is around timescales, not means. We don't see how this can be confusing. CMIP time series are all annual – we're not sure where the 5-year figure comes from. The only place where five-year averages are used is for estimating total warming over a given time period. We do compare steps within set decades to ECS but that is a special case.

*Page 6 – I do not see why entrainment of heat energy into the various heat reservoirs of the Earth and especially the hydrothermal system need always be nonlinear*

Perhaps this is a short-hand argument, but the transport of heat from the equator to the poles is fundamentally nonlinear at the global scale. Some of the transport into the heat reservoirs (e.g., the west Pacific warm pool and the deep ocean is more gradual). However, we would argue that the atmosphere-ocean processes involved in the largescale transport of energy in the system is nonlinear.

Linear behaviour in specific processes or locales is of course possible. Alternatively, there is no reason why the first-order assumption in a nonlinear system should be to assume linearity – it may be, but the assumption it just is, is habit and a social construct.

[Figure]

This section will be rewritten but the nonlinearity of the hydrothermal system will remain a feature.

*Page 6 – Lines 2 to 6 outline a number of alternative approaches to determine 'shifts', 'change points', 'step changes', but there is no discussion of the advantages/disadvantages of these different approaches and why they were not used in this study. See also: Drijfhout et al. (2015) and Reid et al. (2016) Reid PC, Hari RE, Beaugrand G et al. (2016) Global impacts of the 1980s regime shift. Global Change Biology, 22, 682-703.*

Because it was judged that an objective rule-based version of the bivariate test was the best tool we could use, based on our previous use of the test and the effort that has been put into developing the multi-step rule-based model. This has been discussed in past papers and we will be a little more emphatic about it. This discussion will be held in another paper currently being prepared by Ricketts, who does test some of the alternatives.

There is a good summary in Rodionov (2005) and the bivariate test is on a par with the Alexanderssen test (Rodionov does not mention the bivariate test but colleagues at the Australian Bureau of Meteorology tested both in the 1990s when developing homogenization strategies and judged them to have similar performance). The bivariate test has the advantage of being able to use different reference time series. The multi-step procedure was developed to overcome the problems with multiple steps, where the test results do not hold – whereas they will for sequential testing.

*Page 7/8 – acronyms*

We will put a Table for acronyms in the Supplementary Information – ECS is given in the text previous to the acronym on Page 7.

*Page 7 – ECS*

It is explained later, used as an independent variable against which to measure timeseries components through correlation to determine which carries more signal and which carries more noise. We will slightly expand the explanation of the tests as also requested by Reviewer 1

*Page 9 lines 22 to 28. A diagrammatic representation of the different terms used for the analyses is needed. A descriptive expansion of what is meant by each of the terms would be helpful. The word 'shift' has been used in a different way in previous papers and a different word would be more appropriate here for this characteristic. Is the text in brackets at the end of the last bullet correct?*

Finding terminology to go with nonlinear change and defining and measuring it is difficult, whereas there is so much language associated with trend analysis and framing around this that we are used to. We retain steps for the bivariate test because that is what it measures (as is the case for the STARS (Rodionov) test). After much consideration, shift was chosen as the most representative measure of visually what can be seen and measured across the gap produced by displaced trends. A diagram will be produced. Note that the term regime is still being debated (e.g., Overland et al., 2008) – a scientific language for nonlinearity needs to be developed. We cannot do that here. Can do a figure.

*Page 23 Section 5.1. This section would be better drafted as the conclusions of the paper rather than as a summary of severe testing.*

We would prefer to leave this here and focus on this summary in the discussion – it is a long paper and we see a summary and the conclusions as being slightly different.

*Page 25 line 2. Again I do not like the use of the word decadal here. Table 6 does not show that hstep is better at a decadal scale the steps are occurring within a year, but may continue at a new level or develop a trend afterwards for more than a decade.*

Can be changed to decadal timescales, but the use of decadal scale to signify timescales of decades is almost ubiquitous in climatology and if decadal means are signified, decadal mean is the term usually used (Google scholar confirms this if "decadal scales" and climate are search terms). We are quite puzzled with these objections to using decadal in this sense. The only exception might be for ocean sediment coring, where sampling horizons can be decades and centuries.

*Page 27 line 1-2. The hiatus is now thought to be due to an increased storage of heat deeper in the ocean and is not a continuing event considering the warming of the last few years. See Reid 2016 and references included.*

We disagree with this interpretation because we view the hiatus as a regime, and the steady state in between step changes a normal part of climate. In another manuscript ready for submission, we go into this in greater detail interpreting the deep ocean mixing as contributing to the length of the regime because it takes longer for the heat in the shallow ocean to build up to critical levels (Of the 58 CMIP5 models that underwent a step change in 1996–98, the longest interval was 26 years and the shortest 7 years). If heat is being stored in the deep ocean, less is available for the western Pacific warm pool, which acts the heat engine for global climate.

The so-called hiatus and previous mid-century pause (Wally Broecker coined that term) was clearly related to a La Niña phase of the Interdecadal Pacific Oscillation, however, for the 1977 – 1996 period of the El Niño phase, we suggest there were two steps rather than a trend. We are likely to be in the next step change and Peyser et al. (2016) have identified the trigger for this in dynamic changes in sea level in the warm pool, leading to an outburst of heat (They interpret this as variability, but our view is that it is a nonlinear expression of the climate signal).

In the revision, we will reinterpret two recent publications (Peyser et al., 2016; Meehl et al., 2016) that came out after the initial submission to explain the trigger for the recent warming and that which occurred in 1996–98 (Peyser et al., 2016). Meehl et al. (2016) suggest that the IPO may be switching from negative to positive that they interpret as the resumption of an increased warming trend, largely similar to the comment above.

*Page 18 line 4. Lack of predictability. How can the authors be so definite that this might be due to aerosols?*

Because of the negative correlations associated with decades of negative steps responding to volcanic eruptions and sulphate aerosols of the immediate post WWII period and the positive correlations cancelling each other out – as discussed on page 17 – not sure why this is contentious – will look at making it more obvious.

*Page 14 lines 23-25. Sea level steps are said to be ubiquitous in local tide gauge time series, Table 3 in Jones et al. 2013, but were not checked or analysed by Jones et al.*

Misreading of the table therein – model sea level not checked – that statement was for observations. Tide gauge records are illustrated in Figure 36 (Fremantle, San Francisco) of that reference. We are considering putting in a panel of four non-temperature step changes in the revised version to better illustrate Test 5: two tide gauge record (San Francisco, Fremantle), rainfall (northern Australia and south-west Western Australia) and shallow ocean heat content.

Technical corrections – all corrections addressed unless otherwise indicated

*Page 2 Line 2. Abstract. Change to: 'variations that extend over decadal scales of time'. See later comment on use of decadal.*

See responses above

*Page 2 Line 13. 'the correlation'*

*2.2 Line 21. First mention of H1 and H2 together. They were used separately in the introduction.*

Not sure what the point is here – these sections to be rewritten as per both reviewers' comments.

*Page 6 Line 7. Start 'For H1: : :'.on a different line to make it comparable to H2 below. There are no citations to back up the statements made in the H1 section.*

This is because it's our reasoning – will make this clearer in the rewrite – there is another ms that describes this in detail

*Line 18. Decadal again. The transfer from one regime to another is evident at an annual level and not decadal.*

See comments above

*Line 17. Should not be numbered 3 or indented.*

No. We think there a three distinct points, rather than two choices. Wording amended to clarify this.

*Lines 25 to 30. This text should be part of a discussion and not here.*

Ok, but section and whole discussion will be rewritten.

*Line 33. At the end it is important to note that regime change is precipitated, but to a new level or a trend.*

Will consider this, but probably best for discussion – Peyser et al. and Meehl et al. (2016) allow us to better identify the mechanism.

*Line 31 to Page 7 line 4 repeated below. – will remove*

*Page 7 line 12. –H1 and –H2 mentioned for the first time. Define what they mean in general language.*

Section to be rewritten and -H1 and -H2 to be removed to simplify.

*Page 7 line 13 onwards – Six tests are identified. It is not clear if the first two are the same as the two tests mentioned on page 9 lines 31-32. Please make this section clearer.*

They are the same – will expand this section slightly to say what the tests do in more detail.

*Page 8 line 19/20 "MYBT is considered reliable". Is this remark necessary without some backup? You could refer to page 13 line 2 in the Supplementary Information*

We will be more direct about why the bivariate test is being used. Twelve months' development went into the rule-based multi-step component adapting the original test with a great deal of testing to ensure that the results were robust, consistent with known steps and the test could be as reliable as possible with data that contained real trends and lagged autocorrelations (ENSO-like). There is no doubt that redness itself will produce shifts in the data, which is why theory is so important when trying to interpret the results. Some of this is documented in Ricketts (2015), but unfortunately the final paper describing the model has not yet been finalised.

*Page 9 line 2. Put in a heading Data and distinguish between the observed and modelled time series by putting them in different paragraphs. It would have been helpful to leave a line space between each paragraph. Page 9 line 13 Again provide a new sub-heading*

*Page 9 line 32. Again, does the reference to Test 1 and 2 refer to the first two tests of the six mentioned earlier?*

*Page 11 below line 13 put in a heading: Shift/Trend Ratios*

*Page 12 lines 1 and 2. An important result. Missing full stop after warming.*

*Page 12 line 7. Suggest change to "Annual and seasonal anomalies were investigated". And edit next sentence so not starting with Annual.*

*Page 12 line 12. Why are quarterly anomalies only examined for the satellite temperatures? This needs explaining.*

All ok to do

*Page 12 line 23. Confirmation of the results from Reid et al. 2016 that the 1987 regime shift is evident at a global scale and yet on the next line it is said to be only evident at*

*a regional scale.*

Will explain why those results come about – mainly through using different area averages – however, not confident that all the step changes/shifts identified by Reid et al., (2016) should be allocated to 1987/88 (e.g., Australia).

*Page 12 line 25 and 26. An important result. "When all four records are plotted on a common baseline of 1979–1998, the surface and satellite temperatures display similar shifts but different internal trends (Fig. 3)".*

*Page 12 lines 333-34. An important observation. "Unless substantially contaminated by artefacts, these changes do not reflect gradual warming in the atmosphere, but instead may reflect regime-like change controlled from the surface". As is the subsequent comment on heat release from the ocean during El Niño. See commentary in Reid 2016 on this issue.*

We will go into this into the discussion, especially through Peyser et al. (2016)

*Page 13 line 5. Which timescale? – ok Page 13 line 14. Insert 'out' after carried. – ok Page 14 line 18 An important observation. "indicates that the onset of the warming signal in these broader regions is abrupt (Jones, 2012)". Page 14 line 21 Use year (2016) of hard copy publication for Reid et al. (2015). – ok Page 14 lines 23-25. Sea level steps are said to be ubiquitous in local tide gauge time series, Table 3 in Jones et al. 2013, but were not checked or analysed by Jones et al.*

Misreading of the table therein – model sea level not checked – that statement was for observations. Tide gauge records are illustrated in Figure 36 (Fremantle, San Francisco) of that reference.

*Page 17 line 4. Why are 5 year averages used here, the first mention that the data has been treated in this way.*

Not sure why this is an issue – it's a simple difference using 5-year averages to avoid single-year variations. The IPCC often uses 10-year averages in a non-stationary system for simple differences, we use five to minimise the sampling errors with one year but wanted to keep this interval as short as possible. Because it has been done consistently for an ensemble it will provide consistent results.

*Page 18 line 33. Lable bullet A1 and at top of next page the bullet A2.*

Not sure what A1 and A2 signify

*Page 19 lines8-9. "peaking in the 2080s: : :.." does not fit with the figure 5f. What should be Fig. 5f is a duplicate of Fig. 5d.*

Not sure how that happened – will fix

*Page 20 line 23. Is the first part of Section 5 essential to the paper? Would it be better to label it 'Sensitivity testing'.*

Yes, this is essential and is a comprehensive way of testing whether the time series are steplike or trend-like on the timescales of interest. The section will be edited slightly to sharpen it.

*Page 20 line 25. Insert 'and' after 'warming'? Page 21. Line 30 change to: 'performs the best' Page21 line 31-33. Duplication 'into the' and 'test'. Change to: 'at a global scale when each model is' Page 22 line 30. '21st' Page 24 line 9. Spelling 'are' not 'area' Page 24 line 18-19. Edit sentence beginning: 'Warming is not: : :.' Page 24 lines 24-25. Make sure this statement is backed up by appropriate citations in the results section*

ok

*Page 24 lines 9-10 and 30-32 repetition. Is this necessary.*

Section to be edited and this removed

*Page 25 lines 5 to 10. Delete 'In summary' and draft as the final paragraph of the conclusions.*

Will move into the conclusions and edit

*Page 25 line 7 in situ in italics. And, 'or as a gradual'. Page 25 lines 9-10. Edit to: 'where climate change and variability interact rather than varying independently.' Page 25 line 13 Discussion. Include a discussion of how the results of Drijfhout et al. (2015) compare to those presented in this paper. Drijfhout S, Bathiany S, Beaulieu C et al. (2015) Catalogue of abrupt shifts in Intergovernmental Panel on Climate Change climate models. Proceedings of the National Academy of Sciences, 112, E5777-E5786.*

Ok

*Page 25 line 15 change 'earlier' to 'before'? Page 25 line 17. 'gradualism' and 'as a key tool to understand how'? Page 25 line 23 'to explain climate' Page 25 line 24. Change' 'covering methods' to 'applying procedures'? Page 25 line 25. Delete 'and its application to understanding climate processes'. Page 26 line 5. 'analytical'. Page 26 line 12 a priori Italics Page 26 lines13-14. Important observation that needs to be included in the conclusions. 'the processes involved are timescale invariant indicate that the meaning of seamless has not really been thought through'. Page 26 line16. 'would likely be'. And change 'considerable' to 'sizeable' as repeated on the next line. Page 26 line17. Change 'under' to 'that have'. Page 26 line 19. First sentence of bullet. Something is missing. Page 26 line 20-21. 'physics, understood as being primarily linear and hydrometeorology with its substantial nonlinear behaviour; both remain largely unreconciled.*

All the above to be removed except for the bullet point on decadal prediction, retained in discussion

*Page 27 line 9. 'stated'*

To be removed

*Page 27 line 20. Somewhere in the text above it is worth stating that both Cahill and Foster consider that the hiatus was a non-event.*

This passage is to be removed from the discussion

*Page 28 lines 7-9. Delete: 'As we discussed in a related paper where H2 is examined in greater detail' and the reference to Jones and Ricketts, 2016 as this paper is only 'in preparation'. Edit the sentence without the above text except for H2.*

Text to be removed and discussion rewritten

*Page 28 lines 11-18. An important paragraph. You might also cite Roemmich's recent papers and Reid 2016 to back up this paragraph.*

They are cited already

*Page 28 lines 19- 22 repeated below on lines 23-26.*

removed

*Page 28 line 31, The word 'extraordinary' is perhaps a bit too strong.*

Edited

*Page 28 line 32. 'to either side' Page 29 lines 1-2. Leave out the sentence: 'Elsewhere : : :..'., but, raise the possibility that we are undergoing another shift at present.*

Substantially rewritten with new research to suggest that we are undergoing another shift at present

*Page 29 lines 3-5. Poor ending to this section. Edit and improve as a statement to round off the discussion.*

Thank you and will be done

*Page 29 line 13. See earlier comment on >50 year climate change.*

The context for this will be made much clearer – we state that it is a complex trend over the long term – this is physically important as it relates to changing boundary conditions

*Page 29 line 17. Delete sentence beginning: 'We discuss this : : :..'*

Done

*Page 46. Figure 4. I don't know what the journal policy is for sub-figures, but I prefer the lettering, a, b, c to be in the top left hand corner, inside the enclosing border of each sub-plot. It would also help if the respective sub-plots were labelled: England, Texas and Australia within the enclosing border. Insert at the beginning of the legend: 'Regional temperature change'.*

*Page 47. Same comment as for Figure 4. Label a, b, c, d, e, f in the top left corner of the subplots and in the top right in order: Add in sequence in the top right corner of a: 'observed', of b: simulated, of c: '2.6', of d: '4.5', of e: '6.0' and of f: '8.5'. In the legend add downward blue and upward red as for Figure 1.*

Figures can be edited as per suggestions

References

Broecker, W. S.: Global warming: Take action or wait?, Jokull, 1-16, 2005.

Drijfhout, S., Bathiany, S., Beaulieu, C., Brovkin, V., Claussen, M., Huntingford, C., Scheffer, M., Sgubin, G., and Swingedouw, D.: Catalogue of abrupt shifts in Intergovernmental Panel on Climate Change climate models, Proceedings of the National Academy of Sciences, 112, E5777-E5786, 2015.

Jones, R. N., Young, C. K., Handmer, J., Keating, A., Mekala, G. D., and Sheehan, P.: Valuing Adaptation under Rapid Change, National Climate Change Adaptation Research Facility, Gold Coast, Australia, 182 pp., 2013.

Leith, C.: The standard error of time-average estimates of climatic means, Journal of Applied Meteorology, 12, 1066-1069, 1973.

Leith, C.: The design of a statistical-dynamical climate model and statistical constraints on the predictability of climate, in: The Physical Basis of Climate and Climate Modelling, World Meteorological Organisation, Geneva, 137-141, 1975.

Meehl, G. A., Hu, A., and Teng, H.: Initialized decadal prediction for transition to positive phase of the Interdecadal Pacific Oscillation, Nature Communications, 7, 11718, 10.1038/ncomms11718, 2016. Overland, J., Rodionov, S., Minobe, S., and Bond, N.: North Pacific regime shifts: Definitions, issues and recent transitions, Progress In Oceanography, 77, 92-102, 2008.

Palmer, T. N.: A nonlinear dynamical perspective on climate prediction, Journal of Climate, 12, 575-591, 1999.

Peyser, C. E., Yin, J., Landerer, F. W., and Cole, J. E.: Pacific sea level rise patterns and global surface temperature variability, Geophysical Research Letters, n/a-n/a, 10.1002/2016GL069401, 2016.

Reid, P. C., Hari, R. E., Beaugrand, G., Livingstone, D. M., Marty, C., Straile, D., Barichivich, J., Goberville, E., Adrian, R., and Aono, Y.: Global impacts of the 1980s regime shift, Global Change Biology, 22, 703, 10.1111/gcb.13106, 2016.

Ricketts, J. H.: A probabilistic approach to climate regime shift detection based on Maronna's bivariate test, The 21st International Congress on Modelling and Simulation (MODSIM2015), Gold Coast, Queensland, Australia, 2015.

Rodionov, S. N.: A brief overview of the regime shift detection methods, Large-Scale Disturbances (Regime Shifts) and Recovery in Aquatic Ecosystems: Challenges for Management Toward Sustainability. UNESCO-ROSTE/BAS Workshop on Regime Shifts, Varna, Bulgaria, 14-16 June 2005, 2005.
* * *

---

## Author Response (AR1)

**Authors' response**

We would like to thank both reviewers for their positive and thoughtful comments, most of which we have accommodated in some way. It is a large paper, so we are grateful for the effort and attention the reviewers have given.

The revision outlines a structure that better relates theory to plausible physical mechanisms and statistical tests at the beginning of the paper to make it quite clear what we are testing and why. The different uses of linear and linearity in the manuscript are clarified. The more philosophical passages have been removed.

**Response to Reviewer 1**

POINT 1: *My main concern is that it is not clear enough what exactly the hypotheses are that are tested. Sometimes the authors say that what is tested is whether (i) internally and externally forced components of the climate system are independent or not (page 3, page 29); sometimes they say that what is tested is whether (ii) the development of climate variables follows a trend or is step-like (abstract, Section 5). It is not the case that independence of internally and externally forced components of the climate system implies that there is a trend; this is possible, but there could also be independence and at the same time step-like behaviour. Also, it is not the case that dependence of internally and externally forced components implies that there is necessarily step-wise behavior. This could be the case, but there could also be dependence and a trend at the same time. As a result, it remains unclear what exactly is tested: (A) (Only) whether the internally and externally forced components are independent. (B) (Only) whether the climate variables follow a trend or not. (C) Whether the internally and externally forced components are independent AND whether there is a trend. (D) Whether the internally and externally forced components are dependent AND there is step-like behavior of the climate variables. Throughout the paper, the authors need to be clearer what exactly is tested.*

*"but there could also be independence and at the same time step-like behaviour"*

The exact hypotheses have been clarified, making the testing environment much clearer. In particular, how *H1* and *H2* related to $h_{step}$ and $h_{trend}$.

This has resulted in the rewriting of the introduction and an expansion of the methodology section to relate scientific to statistical hypotheses and described the tests in more detail.

*"it is not the case that dependence of internally and externally forced components implies that there is necessarily step-wise behaviour"*

True, and we have made it clear that we are not claiming this and why we are using steps, given the amount of prior knowledge we have about their occurrence.

POINT 2: *Related to this, if what is tested is (C) and (D) (as is often suggested; cf. in particular the hypotheses on page 5), then it is important to see that (C) and (D) are not exhaustive (because there are also the possibilities that there is independence and a step-wise development; or that there is dependence and a trend). The authors want to test an exhaustive set of hypotheses, but (C) and (D) are not exhaustive.*

We are not testing an exhaustive set of either physical or statistical hypotheses and have made this clear; however, understanding the theoretical background and potential mechanisms that informs the testing environment is important. The existing research identifying step changes as a mechanism for warming change on decadal timescales needs to be tested in an environment that is dominated

by an existing paradigm that says the opposite. Because paradigms are partly a sociological construct (though are informed by theory) only a strong case that can cover the theory along with addressing cognitive values has a chance of being accepted.

The framework we are using is a theoretical-mechanistic – statistical induction framework where theory is used to distinguish plausible mechanisms that allow a clear choice to be made between alternative hypotheses. This will be described more clearly. Statistical testing provides the means to do this, so statistical hypotheses need to be developed that match the alternative mechanisms.

POINT 3: Throughout the paper the assumption seems to be that "trend-like" and gradual as opposed to step-wise and non-gradual means that there is a linear relationship (e.g. on page 7). It is unclear why gradual implies that there is a linear relationship. There can be gradual behavior with various kinds of relationships (a quadratic relationship etc).

Here linear is referring to whether the temperature signal follows a secular trend therefore can be defined by a line and refers to the linear transfer relationship between forcing and temperature. Nonlinear response implies the transfer is nonlinear and is modified by nonlinear physical processes. This has been clarified in Section 2 para 2. The usage in the paper is physical and when statistically linear is used is will be specific, e.g., linear trends.

POINT 4: On page 7 the hypothesis states that there is a "(probably monotonic)" trend. The brackets are confusing. Is it now tested that the trend is monotonic or is it allowed that the trend is not monotonic?

Related to the previous point – we have clarified that the underlying signal of atmospheric warming is assumed to be monotonic (while we accept that total global warming is monotonic). Section 2 has been rewritten as per the overall response to both reviewers 1 and 2.

POINT 5: on page 5 six tests are described. It should be clearly stated which tests test which hypotheses (becomes clear later, but should be stated clearly early on).

The tests have been clarified. They all test the hypotheses but Tests 1–4 are mainly probative, Test 5 a mixture and Test 6 is largely error testing to determine whether steps or trend best describe the data.

POINT 6: The beginning of Section 2: here it is argued that the gradualist thesis is derived from induction. Yet, as the paper later argues, the data actually do not support the gradualist thesis and the gradualist thesis rather seems to be often adopted for no empirical reasons (convenience, simplicity). Hence it seems that the gradualist thesis is not justified by induction after all.

It is probably more accurate to say the gradualist thesis is sustained by statistical inference (as a form of induction). The paper has been made clearer in this regard and focuses on severe testing.

Below are some notes but this discussion is omitted from the paper.

We have done some more work around this to inform other papers on the same theme. In the 1970s, various theoretical arguments were put to suggest if a system received a small internal forcing the response would be proportional (e.g., Leith 1973, 1975). Much of the subsequent work was based on vectorizing the forcing-response relationship, which will produce a linear outcome. This was based on statistical physics.

There are two types of induction therefore: one is largely analogical, based on statistical physics, proposed by scholars like Leith. This is now generally accepted as an approximation that only holds if

the residuals are Gaussian (e.g., Palmer, 1999) and they are not in every case. There have been two styles of statistical induction. The first in the 19th century based on, which is truth-centred. The other is modern statistical induction, which is often reduced to a mechanised form based on establishing a sufficiently small probability for the null hypothesis. For some, the trend has again become truth-centred, but this can is only be the case if warming is gradual. Otherwise, it is a statistically-derived approximation of the relationship between forcing and response. This issue is at the core of the paper.

Other cognitive values such as convenience and simplicity are applied, because they are mentioned in the literature frequently, often as an escape clause to bypass some acknowledged but unknown complexity. They are also used as a defence against over-parameterisation and overfitting. Their use is mainly sociological rather than scientific. A full discussion of all of these is not appropriate for this paper, so we will pare it back to the basic issues – we acknowledged these other issues because readers will realise that the story is incomplete but a more comprehensive exploration is not feasible.

This issue has been complicated by the climate wars, where trend analysis has been used as a defence, so if not strictly truth-centred, it had to be defended as scientifically correct. This has hampered the consideration of alternatives. This comment is largely background because most of the philosophical content would be removed in a revision leaving the severe testing component.

*POINT 7: The beginning of Section 2: "The application of linear trend analysis to atmospheric warming is invariably justified as inference to the best explanation". I am puzzled by this sentence. Why is there suddenly a reference to the inference to the best explanation (previously the matter of concern was induction).*

This point has been omitted from the revision.

Technical corrections have been addressed.

**Response to Reviewer 2**

**General comments**

The single paper option has been chosen as recommended by the editor and the introduction and Section 2 rewritten to focus on the statistical testing. However, it is important to preface the statistical tests with the probative conditions as to why those tests were chosen. The relationship between theoretical propositions of independence and interacting externally-forced and internally-generated change have been simplified (as per the response to reviewer 1), the mechanisms reflecting those theoretical propositions and why they were selected and the statistical hypotheses are better described. The most important aspect of this is to test between sustained incremental change as would be manifest in gradual warming and episodic change. For that reason, retaining Page 5 lines 6–19 is important because it explains what the statistical hypotheses represent.

**Specific comments**

Abstract – test numbers bracketed

Introduction – see above

2.3 line 23 – ill-posed inverse problem will be given a plain-language description and term has been removed. This relates to part of test 1: regional stratification – the test description has been revised.

External forcing – linear – the explanation referred to in the response to Reviewer 1's comments has clarified the relationship between physical and statistical linearity.

Decadal scale – the most widespread usage is around timescales, not means. We don't see how this can be confusing. CMIP time series are all annual – we're not sure where the 5-year figure comes from. The only place where five-year averages are used is for estimating total warming over a given time period. We do compare steps within set decades to ECS but that is a special case.

Page 6 – *I do not see why entrainment of heat energy into the various heat reservoirs of the Earth and especially the hydrothermal system need always be nonlinear*

Perhaps this is a short-hand argument, but the transport of heat from the equator to the poles is fundamentally nonlinear at the global scale. Our substantial argument has been moved to the discussion and the linearity/nonlinearity argument much better focussed.

Page 6 – *Lines 2 to 6 outline a number of alternative approaches to determine 'shifts', 'change points', 'step changes', but there is no discussion of the advantages/disadvantages of these different approaches and why they were not used in this study. See also: Drijfhout et al. (2015) and Reid et al. (2016) Reid PC, Hari RE, Beaugrand G et al. (2016) Global impacts of the 1980s regime shift. Global Change Biology, 22, 682-703.*

An objective rule-based version of the bivariate test was the best tool we could use, based on our previous use of the test and the effort that has been put into developing the multi-step rule-based model. This has been discussed in past papers and we will be a little more emphatic about it. This discussion will be held in another paper currently being prepared by Ricketts, who does test some of the alternatives. There is a good summary in Rodionov (2005) and the bivariate test is on a par with the Alexanderssen test (Rodionov does not mention the bivariate test but colleagues at the Australian Bureau of Meteorology tested both in the 1990s when developing homogenization strategies and judged them to have similar performance). The bivariate test has the advantage of being able to use different reference time series. The multi-step procedure was developed to overcome the problems with multiple steps, where the test results do not hold – whereas they will for sequential testing.

This has been clarified in the text

*Page 7/8 – acronyms*

We have put a Table for acronyms in the Supplementary Information (Table S4) – ECS is defined on Page 6.

Page 7 – ECS

Its use is explained in more detail later, as an independent variable against which to measure timeseries components through correlation to determine which carries more signal and which carries more noise. We have slightly expanded the explanation of the tests as also requested by Reviewer 1

Page 9 lines 22 to 28. *A diagrammatic representation of the different terms used for the analyses is needed. A descriptive expansion of what is meant by each of the terms would be helpful. The word 'shift' has been used in a different way in previous papers and a different word would be more appropriate here for this characteristic. Is the text in brackets at the end of the last bullet correct?*

Finding terminology to go with nonlinear change and defining and measuring it is difficult, whereas there is so much language associated with trend analysis and framing around this that we are used to. We retain steps for the bivariate test because that is what it measures (as is the case for the STARS (Rodionov) test). After much consideration, shift was chosen as the most representative measure of visually what can be seen and measured across the gap produced by displaced trends. A diagram will be produced. Note that the term regime is still being debated (e.g., Overland et al., 2008) – a scientific language for nonlinearity needs to be developed.

Some words to this effect and a figure have been added to a new Section 2.3.4 on metrics.

Page 23 Section 5.1. *This section would be better drafted as the conclusions of the paper rather than as a summary of severe testing.*

We would prefer to leave this here and focus on this summary in the discussion – it is a long paper and we see a summary and the conclusions as being slightly different. We also think this summary is needed to help frame the discussion

Page 25 line 2. *Again I do not like the use of the word decadal here. Table 6 does not show that $h_{step}$ is better at a decadal scale the steps are occurring within a year, but may continue at a new level or develop a trend afterwards for more than a decade.*

Can be changed to decadal timescales, but the use of decadal scale to signify timescales of decades is almost ubiquitous in climatology and if decadal means are signified, decadal mean is the term usually used (Google scholar confirms this if "decadal scales" and climate are search terms). We are quite puzzled with these objections to using decadal in this sense. The only exception might be for ocean sediment coring, where sampling horizons can be decades and centuries.

Page 27 line 1-2. *The hiatus is now thought to be due to an increased storage of heat deeper in the ocean and is not a continuing event considering the warming of the last few years. See Reid 2016 and references included.*

We disagree with this interpretation because we view the hiatus as a regime, and the steady state in between step changes a normal part of climate. The discussion has been expanded to describe our views on the role of regimes.

The so-called hiatus and previous mid-century pause (Wally Broecker coined that term) was clearly related to a La Niña phase of the Interdecadal Pacific Oscillation, however, for the 1977 – 1996 period of the El Niño phase, we suggest there were two steps rather than a trend. We are likely to be in the next step change and Peyser et al. (2016) have identified the trigger for this in dynamic changes in sea level in the warm pool, leading to an outburst of heat (They interpret this as variability, but our view is that it is a nonlinear expression of the climate signal).

In the discussion, we reinterpret two recent publications (Peyser et al., 2016; Meehl  et al., 2016) that came out after the initial submission to explain the trigger for the recent warming and that which occurred in 1996–98 (Peyser et al., 2016). Meehl et al. (2016) suggest that the IPO may be switching from negative to positive that they interpret as the resumption of an increased warming trend, largely similar to the comment above.

Page 18 line 4. *Lack of predictability. How can the authors be so definite that this might be due to aerosols?*

Because of the negative correlations associated with decades of negative steps responding to volcanic eruptions and sulphate aerosols of the immediate post WWII period and the positive correlations cancelling each other out – as discussed on page 17 – not sure why this is contentious – text slightly edited to make this more obvious.

Page 14 lines 23-25. *Sea level steps are said to be ubiquitous in local tide gauge time series, Table 3 in Jones et al. 2013, but were not checked or analysed by Jones et al.*

Misreading of the table therein – model sea level not checked – that statement was for observations. Tide gauge records are illustrated in Figure 36 (Fremantle, San Francisco) of that reference. A new Figure, Figure 5 contains a panel of four non-temperature step changes in the revised version to better illustrate Test 5: two tide gauge record (San Francisco, Fremantle), rainfall (northern Australia and south-west Western Australia) and shallow ocean heat content.

Technical corrections – all corrections addressed unless otherwise indicated

Page 2 Line 2. Abstract. *Change to: 'variations that extend over decadal scales of time'. See later comment on use of decadal.*

See responses above

Page 2 Line 13. *'the correlation'*

2.2 Line 21. *First mention of H1 and H2 together. They were used separately in the introduction.*

These have been rewritten as per both reviewers' comments.

Page 6 Line 7. *Start 'For H1: : :.'.on a different line to make it comparable to H2 below. There are no citations to back up the statements made in the H1 section.*

The H1 case is cited in paragraph 1 and the Corti et al and Hasselmann references describe both.

Line 18. *Decadal again. The transfer from one regime to another is evident at an annual level and not decadal.*

See comments above

Line 17. *Should not be numbered 3 or indented.*

No. We think there a three distinct points, rather than two choices. Wording amended to clarify this.

Lines 25 to 30. *This text should be part of a discussion and not here.*

Incorporated into the discussion

Line 33. *At the end it is important to note that regime change is precipitated, but to a new level or a trend.*

Incorporated into the discussion – Peyser et al. and Meehl et al. (2016) allow us to better identify the mechanism.

Line 31 to Page 7 line 4 *repeated below.* – will remove

Page 7 line 12. *–H1 and –H2 mentioned for the first time. Define what they mean in general language.*

Section rewritten and *-H1* and *-H2* have been removed to simplify.

Page 7 line 13 onwards – *Six tests are identified. It is not clear if the first two are the same as the two tests mentioned on page 9 lines 31-32. Please make this section clearer.*

They are the same – have expanded this section to say what the tests do in more detail.

Page 8 line 19/20 *"MYBT is considered reliable". Is this remark necessary without some backup? You could refer to page 13 line 2 in the Supplementary Information*

We are be more direct about why the bivariate test is being used in Section 2.3. Twelve months' development went into the rule-based multi-step component adapting the original test with a great deal of testing to ensure that the results were robust, consistent with known steps and the test could be as reliable as possible with data that contained real trends and lagged autocorrelations (ENSO-like). There is no doubt that redness itself will produce shifts in the data, which is why theory is so important when trying to interpret the results. Some of this is documented in Ricketts (2015), but unfortunately the final paper describing the model has not yet been finalised.

Page 9 line 2. *Put in a heading Data and distinguish between the observed and modelled time series by putting them in different paragraphs. It would have been helpful to leave a line space between each paragraph.*

Done

Page 9 line 13 *Again provide a new sub-heading*

Ok

Page 9 line 32. *Again, does the reference to Test 1 and 2 refer to the first two tests of the six mentioned earlier?*

Yes – made clear

Page 11 *below line 13 put in a heading: Shift/Trend Ratios*

Ok - added

Page 12 lines 1 and 2. *An important result. Missing full stop after warming.*

Ok

Page 12 line 7. *Suggest change to "Annual and seasonal anomalies were investigated". And edit next sentence so not starting with Annual.*

ok

Page 12 line 12. *Why are quarterly anomalies only examined for the satellite temperatures?*

*This needs explaining.*

Sentence added to say that seasonal anomalies were examined to distinguish between 1995 and 1998 step dates. Quarterly (3-monthly) time series were assessed for satellite and two surface temperature records.

Page 12 line 23. *Confirmation of the results from Reid et al. 2016 that the 1987 regime shift is evident at a global scale and yet on the next line it is said to be only evident at a regional scale.*

Have added that these results were mainly due to using different area averages – however, not confident that all the step changes/shifts identified by Reid et al., (2016) should be allocated to 1987/88 (e.g., Australia).

Page 12 line 25 and 26. *An important result. "When all four records are plotted on a common baseline of 1979–1998, the surface and satellite temperatures display similar shifts but different internal trends (Fig. 3)".*

Page 12 lines 333-34. *An important observation. "Unless substantially contaminated by artefacts, these changes do not reflect gradual warming in the atmosphere, but instead may reflect regime-like change controlled from the surface". As is the subsequent comment on heat release from the ocean during El Niño. See commentary in Reid 2016 on this issue.*

We will go into this into the discussion, especially through Peyser et al. (2016)

Page 13 line 5. *Which timescale?* – clarified

Page 13 line 14. *Insert 'out' after carried.* – ok

Page 14 line 18 *An important observation. "indicates that the onset of the warming signal in these broader regions is abrupt (Jones, 2012)".*

Page 14 line 21 *Use year (2016) of hard copy publication for Reid et al. (2015).* – ok

Page 14 lines 23-25. *Sea level steps are said to be ubiquitous in local tide gauge time series, Table 3 in Jones et al. 2013, but were not checked or analysed by Jones et al.*

Misreading of the table therein – model sea level not checked – that statement was for observations. Tide gauge records are illustrated in Figure 36 (Fremantle, San Francisco) of that reference and are updated here.

Page 17 line 4. *Why are 5 year averages used here, the first mention that the data has been treated in this way.*

This is a simple difference using 5-year averages to avoid single-year variations. The IPCC often uses 10-year averages in a non-stationary system for simple differences, we use five to minimise the sampling errors with one year but wanted to keep this interval as short as possible. Because it has been done consistently for an ensemble it will provide consistent results.

Page 18 line 33. *Lable bullet A1 and at top of next page the bullet A2.*

Not sure what A1 and A2 signify – left unchanged

Page 19 lines8-9. *"peaking in the 2080s: : :.." does not fit with the figure 5f. What should be Fig. 5f is a duplicate of Fig. 5d.*

Not sure how that happened – have fixed

Page 20 line 23. Is the first part of Section 5 essential to the paper? Would it be better to label it 'Sensitivity testing'.

Yes, it is essential and is a comprehensive way of testing whether the time series are steplike or trend-like on the timescales of interest. The section has been edited slightly to sharpen it and we are making it clear this is largely error testing.

Page 20 line 25. *Insert 'and' after 'warming'?*
Page 21. Line 30 change to: *'performs the best'*
Page21 line 31-33. *Duplication 'into the' and 'test'. Change to: 'at a global scale when each model is'*
Page 22 line 30. *'21st'*
Page 24 line 9. *Spelling 'are' not 'area'*
Page 24 line 18-19. *Edit sentence beginning: 'Warming is not: : :.'*
Page 24 lines 24-25. *Make sure this statement is backed up by appropriate citations in the results section*

All ok

Page 24 lines 9-10 and 30-32 *repetition. Is this necessary.*

Section edited and this has been removed

Page 25 lines 5 to 10. *Delete 'In summary' and draft as the final paragraph of the conclusions.*

Will move into the conclusions and edit

Page 25 line 7 *in situ in italics. And, 'or as a gradual'.*

Page 25 lines 9-10. *Edit to: 'where climate change and variability interact rather than varying independently.'*

Page 25 line 13 *Discussion. Include a discussion of how the results of Drijfhout et al. (2015) compare to those presented in this paper.*

Drijfhout S, Bathiany S, Beaulieu C et al. (2015) Catalogue of abrupt shifts in Intergovernmental Panel on Climate Change climate models. Proceedings of the National Academy of Sciences, 112, E5777-E5786.

Done

Page 25 line 15 *change 'earlier' to 'before'?*
Page 25 line 17. *'gradualism' and 'as a key tool to understand how'?*
Page 25 line 23 *'to explain climate'*
Page 25 line 24. *Change' 'covering methods' to 'applying procedures'?*
Page 25 line 25. *Delete 'and its application to understanding climate processes'.*
Page 26 line 5. *'analytical'.*
Page 26 line 12 *a priori Italics*
Page 26 lines13-14. *Important observation that needs to be included in the conclusions. 'the processes involved are timescale invariant indicate that the meaning of seamless has not really been thought through'.*
Page 26 line16. *'would likely be'. And change 'considerable' to 'sizeable' as repeated on the next line.*
Page 26 line17. *Change 'under' to 'that have'.*

Page 26 line 19. *First sentence of bullet. Something is missing.*
Page 26 line 20-21. *'physics, understood as being primarily linear and hydrometeorology with its substantial nonlinear behaviour; both remain largely unreconciled.*

All the above removed except for the bullet point on decadal prediction, retained in discussion

Page 27 line 9. *'stated'*

Removed

Page 27 line 20. *Somewhere in the text above it is worth stating that both Cahill and Foster consider that the hiatus was a non-event.*

This passage removed from the discussion

Page 28 lines 7-9. *Delete: 'As we discussed in a related paper where H2 is examined in greater detail' and the reference to Jones and Ricketts, 2016 as this paper is only 'in preparation'. Edit the sentence without the above text except for H2.*

Text removed and discussion rewritten

Page 28 lines 11-18. *An important paragraph. You might also cite Roemmich's recent papers and Reid 2016 to back up this paragraph.*

Cited

Page 28 lines 19- 22 *repeated below on lines 23-26.*

Removed

Page 28 line 31, *The word 'extraordinary' is perhaps a bit too strong.*

Edited

Page 28 line 32. *'to either side'*

Page 29 lines 1-2. *Leave out the sentence: 'Elsewhere : : :..'., but, raise the possibility that we are undergoing another shift at present.*

Substantially rewritten with new research to suggest that we are undergoing another shift at present

Page 29 lines 3-5. *Poor ending to this section. Edit and improve as a statement to round off the discussion.*

Completely rewritten – this point has not been retained.

Page 29 line 13. *See earlier comment on >50 year climate change.*

The context has been made much clearer – we state that it is a complex trend over the long term – this is physically important as it relates to changing boundary conditions. The theoretical background for why this is important has been expanded

Page 29 line 17. *Delete sentence beginning: 'We discuss this : : :..'*

Done

Page 46. *Figure 4. I don't know what the journal policy is for sub-figures, but I prefer the lettering, a, b, c to be in the top left hand corner, inside the enclosing border of each sub-plot. It would also help if*

*the respective sub-plots were labelled: England, Texas and Australia within the enclosing border. Insert at the beginning of the legend: 'Regional temperature change'.*

Page 47. *Same comment as for Figure 4. Label a, b, c, d, e, f in the top left corner of the subplots and in the top right in order: Add in sequence in the top right corner of a: 'observed', of b: simulated, of c: '2.6', of d: '4.5', of e: '6.0' and of f: '8.5'. In the legend add downward blue and upward red as for Figure 1.*

Figures edited as per suggestions

Leith, C.: The design of a statistical-dynamical climate model and statistical constraints on the predictability of climate, 
[revised manuscript text omitted]

---

## Author Response (AR2)

Author's response

Thank you for the comments.

They actually went to the heart of the paper's structure, so required some consideration. We have endeavoured to address them, sometimes giving extra explanation, while trying not to go beyond the reviewed content.

We are grateful to ESD and Copernicus for the opportunity to submit a longer paper because there are not many journals left that will accommodate such length. This is difficult when researchers have work that requires detailed explanation, is controversial or both.

**A. ON THE STATS**
* * *
1. The scientific methodology follows Mayo's account of severe testing. D. Mayo is a philosopher, and perhaps fortunately no philosophical statement is ever perceived as definitive nor uncontroversial. There are certainly zones of frictions between Mayo's discourse and a number of statisticians and this is how ideas evolve, which is perfectly fine. For sure, inference in presence of model discrepancy ("all models are wrong") represents a hard philosophical challenge, one which has no obvious solution. Expressing oneself clearly is part of the challenge. In fact, much of the work done in this article can be justified in fairly simple terms: we challenge competing models as much as we can, with constant concern for the physical meaning of what we are doing. You might want to work harder to track sentences that may appear unnecessarily obscure. E.g.:

The focus here is not so much on models but on theory as it is represented by models. The problem of confirming a theory via testing is difficult, especially given that Popperian falsification was eventually judged to be insufficient, including by Popper himself.

Although Mayo is not the clearest illustrator of her own thoughts, a very useful paper has just been published by Brian Haig that outlines the procedure in simple terms

Haig, B. D.: Tests of statistical significance made sound, Educational and Psychological Measurement, 0013164416667981, 10.1177/0013164416667981, 2016.

Even though it's for psychology, it's broadly relevant across the sciences.

We have twice submitted this work as a pair of straightforward and relatively short papers describing statistical tests illustrating warming as steps, not trends. With one journal, we were attacked on the basis that the test was rubbish, the steps are physically impossible and the answer was that climate follows a segmented trend, where the peaks and troughs did not seem to coincide with known dates associated with any climatological phenomena. The findings were ignored. JR has prepared a manuscript showing that the fitted segmented trend model cannot be sustained.

The two papers were combined and severe testing added in because it fitted our approach. Also, we wanted to bring in other lines of evidence to support what is essentially statistical inference.

All the papers we have seen that interpret warming as a monotonic trend have either assumed physical linearity from the outset to justify their use of linear statistics or have subtracted components of climate variability from the data (e.g., ENSO, PDO, AMO etc). The latter strategy will not be appropriate if these phenomena are part of the dissipative system carrying the forcing signal. Also, if forcing is linearly additive over multidecadal timescales, but follows a stochastic pathway, the linearity will show up in standard statistical tests and the stochasticity will be discarded. From a risk analytic point of view, this is not good practice.

In Mayo's world, and with others working across philosophical/disciplinary areas, these kinds of tests constitute weak, not strong, inference. Where null hypothesis significance testing (NHST) is used to say anything $pH_0 < 0.05$ confirms the theory and $pH_0 > 0.05$ confirms the null, then the work qualifies as behavioural or mechanistic statistics (e.g., Gigerenzer, G., and Marewski, J. N.:

Surrogate Science: The Idol of a Universal Method for Scientific Inference, Journal of Management, 41, 421-440, 10.1177/0149206314547522, 2015) where no-one needs to reason why, just run the test.

The paper applies a nested structure that links theory with mechanistic explanations with statistical inference. The reasoning is that if the statistical results can be expressed with climatologically sound arguments, and articulated with links to the underlying theory, then it presents a more complete view of the science.

We want to get readers to accept the statistical arguments, but we also want them to think about the climate system, the processes in it and how that can be characterised as risk. For example, the radiative physics (dry) community and hydrometeorological (wet) communities do not always work that closely together, partly because it takes a lot of work to understand one area, let alone both. There has been friction between the two groups because the 'wet' group criticise the 'dry' group for being linear, when they know that their system is intrinsically nonlinear. The 'dry' response is that their system undergoes boundary-limited change which is linear and the nonlinearity is important but not part of the signal. Insights that would help both groups to understand climate better are often lost because they dismiss each other's concerns.

This also requires people to think beyond a Newtonian mechanical structure described with classical statistics to a complex systems structure, where the statistical approaches fit the aspect of the system being investigated.

- p. 3, l. 28: " It is applied to the substantive null of model adequacy approach described " -> what is the substantive null of model adequacy ?

- The aim of severe testing is to produce highly probed ... : what is "highly probed" ?

To this end, we have explained the substantive null of model adequacy more clearly on page 4 and outlined the probative nature of testing. Although our philosophers say that proof of theory is not possible (as you allude to above), they do discuss probative testing that develops evidence for a hypothesis.

2. p. 6, it is written that "the null hypothesis for H2 is H1". Is that clear that H2 U H2 form the set of all possibilities (which is what is implied by this sentence). How would a quadratic trend stand with respect to both ?

The wording has been changed. A quadratic trend would apply to H1 and we hope this has been made clear. We have also tried to state that although many permutations are possible for H2, based on existing analyses, we are testing step changes only. The nature of the tests makes this clear later on.

B. ON THE PHYSICS
* * *
3. The points 1,2,3 p. 5 are unsatisfactory. The attempt to explain the existence of steps here as a combination between in situ warming and ocean absorption is awkward. The manuscript would be improved by simply acknowledging that there is a physical possibility of step change. In fact, the 3 paragraphs may simply be withdrawn.

Because the paper is focussed on the presentation of physically distinct alternatives, and the latter part of the paper draws on these, we have taken summaries from the previous discussion and brought them forward to make this clearer. As described above, we previously tried a simple approach based on the stats alone and it is insufficient. We want to establish for the reader that the extra (anthropogenic) longwave radiation will take different pathways under H1 and H2.

4. I can't adhere to the distinction between statistical and physical linearity, which you introduced

in response to previous referees comments. Non-linearity does not imply step changes (in fact, strange attractors may respond smoothly to changes in boundary conditions or forcing, and this is the fundamental assumption of linear response theory applied to complex systems, see, e.g., by Lucarini). Use 'smooth' if this is what you mean (or 'continuous' / 'gradual') and don't substitute 'linear' to these concepts.

We have changed the language to gradual-monotonic vs step-like and nongradual. We focus the linear response on the $\delta T = \lambda \delta F$ relationship, which empirically mirrors the linear response theory and is the source of the pattern scaling approach for climate projections (having been introduced in 1990 by Santer, B. D., Wigley, T. M. L., Schlesinger, M. E., and Mitchell, J. F. B.: Developing climate scenarios from equilibrium GCM results, Max Planck Institut für Meteorologie, Hamburg, 79, 1990). We have been quite clear all along that linearity holds over the long term, an assertion that dates back to Lorenz 1975, if not longer, in his characterisation of intransitive change.

Our results indicate that the dissipative system is behaving nonlinearly while the whole system response to the forcing is linear and additive. This is consistent with the linear stochasticity of Hasselmann – and the possibility that both nonlinear behaviour introduced by Lorenz and linear stochasticity are combining is discussed by Ghil who we think comes closest to what we are observing climatologically.

Lorenz, E. N.: Climate Predictability, in: The Physical Basis of Climate and Climate Modelling, World Meteorological Organisation, Geneva, 132-136, 1975.

Hasselmann, K.: Stochastic climate models part I. Theory, Tellus, 28, 473-485, 1976.

Ghil, M.: A mathematical theory of climate sensitivity or, How to deal with both anthropogenic forcing and natural variability?, in: Climate Change: Multidecadal and Beyond, edited by: Chang, C.-P., Ghil, M., Latif, M., and Wallace, J. M., World Scientific Publishing Company, London, Singapore, 2015.

5. p. 21: the reference to entropy production is, I would say, pretty brave. It is not immediately clear that increased GHG generate increased entropy production. There are also many sources of entropy production on the Earth system and not all are relevant to the dynamics of atmosphere-ocean fluctuations. These statements, without proper framework and citation, might end up being judged as vacuous sophisticated statements, and they may be in the end be detrimental to the manuscript.

Multiple papers by Lucarini, Ghil, Ozawa, R Lorenz, Kleidon have to linked entropy production within the climate system, not within the radiative part, though. Mostly to do with the dissipative process. We think the mention here is fairly neutral, given that it refers to distribution.

Many of these are cited in the paper.

C. Other comments:

p. 6 : the text about the 6 tests need to be rewritten in terms of simple sentences, presenting clearly the hypothesis to be tested. Using questions (with question marks) is one option. The semi-telegraphic used here does not suit.

This has been done – thanks for the suggestion

p. 7 : Introduce a section 2.3.1 just after section header.

Done

p. 7, l. 4 : Change "Paraphrasing Mayo and Spanos (2010) to address the results: with very high probability, Tests 1–6 would have produced a result that accords less well with H2/H1 than does

H1/H2, if H2/H1 were false or incorrect" -> into "Paraphrasing Mayo and Spanos (2010) to address the results: with very high probability, Tests 1–6 would have produced a result that accords less well with H2 than does H1, if H2 were false or incorrect (and conversely)".

Done

- The LOWESS model is introduced without definition. It will not be obvious to everyone what it is.

Done

- Three times in the manuscript (p. 12, 18 and 32) it is written that internal trends are given 'preference' to shifts. The background to this affirmation is too implicit. I understand that we only accept shifts if we can reject trends. If this is indeed what is meant to be said it should be more explicit.

Scaled back – this was actually referring to the incommensurate nature of shifts and trends when considering total change (they add up to roughly 105% over the RCP4.5 MME). If trends are considered in a trend/total steps ratio, then that weights trend preferentially to shifts by about 5%. This effect is not strong but it is there.

- The discussion is pretty unusual. The standard practice is to use the discussion section to focus on critical aspects of the methodology and open outlooks for further investigations. Here the discussion has the air of an essay, mixing considerations on climate macro-physics ("homeostatic effect of ocean", "In situ warming is hardly plausible if the atmosphere has no intrinsic heat memory.") with philosophy of sciences ("Most, if not all assessments take up the basic assumptions of those simpler models, forming a consensus mental picture that operates as the dominant paradigm of how climate changes".).
Admittedly, the text is at places much to the point and it conveys a number of relevant ideas. However much care is needed before formulating judgements that may be interpreted as an outsider's opinion about a broad community. Indeed, the stance is at places patronizing "The assumption that the processes involved are timescale invariant indicate that the meaning of seamless has not really been thought through" : thought trough by who ? While I agree that more is to be done about teaching dynamical system concepts in Earth sciences master degrees, we also ought to be fair to a good number of investigators, present on climate science subjects, and who can apprehend many subtleties of non-autonomous, non-linear dynamical systems theory.

In summary, you definitely have the right to express some more informal, though-provoking considerations within of a research article, but I still believe there is considerable scope to shorten and focus the discussion. The manuscript might end up gaining impact in the end.

We have considered this and split the previous discussion into two parts. The first part expands on mechanisms for step-like warming that essentially uses the results of the study in a mini review to construct more detailed explanation of the mechanisms that might be contributing to step-like warming. This is based on a couple of years' work testing and discarding alternatives. Our view is that if we don't proffer a physically realistic case, the statistical conclusions will be regarded as fantastic – at best, interesting, but not important. The Peyser et al. 2016 paper came out after the initial submission and was exactly the type of phenomenon we needed to describe the trigger and release mechanism – the western Pacific warm pool acting like a 'heat pump' fits in with so much evidence, including palaeoclimatic evidence.

The discussion broadens this into related areas covering current mental models and where some of the differences lie when considering H2 from an H1 viewpoint, a brief discussion of how we think the climatology and theory intersect, trying not to be controversial and some comments on implications for application. These are all points in the previously reviewed discussion, but

reordered. The nested model approach to developing theory to nonlinear climate change currently described in the literature (e.g., Ghil, Lucarini and others) is advancing with less speed than climate change itself. We would like to think the issues we discuss in this paper will provide some focus and some urgency.

With respect to the "meaning of seamless has not be fully thought through" – it is still there but reworded and a short explanation added. RJ was coordinating lead author in AR5 WGII chapter Foundations of Decision Making that had a section on climate services. The supply-driven nature of 'climate forecasting' is of great concern to the adaptation community and the stripping out of variability from projections of mean change on the basis of model skill is not a good way to contribute to strategic risk management and decision support. Weather forecasting is event-based and climate forecasting is trend-based. We think this is a mistake from a decision sciences point of view. Moving towards a natural hazards risk-based approach with a liberal use of scenarios is better. The forecasting systems might be operationally seamless, but they need to be carefully thought through to provide what users need, not what the producers think they should deliver. This isn't a physics problem, but having the physics to better understand it would be terrific.

[revised manuscript text omitted]